# COUNT BRIDGES ENABLE MODELING AND DECONVOLVING TRANSCRIPTOMIC DATA

**Nic Fishman**[1,*]**, Gokul Gowri**[2]**, Tanush Kumar**[1]**, Jiaqi Lu**[1]**, Valentin de Bortoli**[3]**,
Jonathan S. Gootenberg**[4,6] **& Omar Abudayyeh**[5,6]
[1]Harvard University; [2]MIT; [3]CNRS;
[4]Beth Israel Deaconess Medical Center; [5]Brigham and Women's Hospital; [6]Harvard Medical School
[*]Corresponding author: `njwfish@gmail.com`

## ABSTRACT

Many modern biological assays, including RNA sequencing, yield integer-valued counts that reflect the number of molecules detected. These measurements are often not at the desired resolution: while the unit of interest is typically a single cell, many measurement technologies produce counts aggregated over sets of cells. Although recent generative frameworks such as diffusion and flow matching have been extended to non-Euclidean and discrete settings, it remains unclear how best to model integer-valued data or how to systematically deconvolve aggregated observations. We introduce Count Bridges, a stochastic bridge process on the integers that provides an exact, tractable analogue of diffusion-style models for count data, with closed-form conditionals for efficient training and sampling. We extend this framework to enable direct training from aggregated measurements via an Expectation-Maximization-style approach that treats unit-level counts as latent variables. We demonstrate state-of-the-art performance on integer distribution matching benchmarks, comparing against flow matching and discrete flow matching baselines across various metrics. We then apply Count Bridges to two large-scale problems in biology: modeling single-cell gene expression data at the nucleotide resolution, with applications to deconvolving bulk RNA-seq, and resolving multicellular spatial transcriptomic spots into single-cell count profiles. Our methods offer a principled foundation for generative modeling and deconvolution of biological count data across scales and modalities.

## 1 INTRODUCTION

Integer-valued counts are a fundamental product of scientific measurements because of the discrete nature of molecules. Modern biological assays yield massive streams of count data: RNA-seq read counts, fluorescence imaging molecule counts, and mass cytometry ion counts (Klein et al., 2015; Raj et al., 2008; Bendall et al., 2011). However, these measurements are often aggregated over multiple individual units, obscuring the fine-grained patterns underlying these natural phenomena. Transcriptomics technologies exemplify this challenge, with technologies such as Visium capturing 10-50 cells per spot (Ståhl et al., 2016) and bulk RNA-seq aggregating thousands to millions of cells per readout, yielding averages rather than high-resolution details. Deconvolving these aggregates into single-cell profiles is critical for the precise mapping of cellular heterogeneity, cell-cell interactions, and tissue architecture (Moses & Pachter, 2022; Armingol et al., 2021). The challenge is twofold: building generative models that respect the integer nature of counts and extending these models to infer unit-level profiles from aggregated observations.

Recent developments in generative modelling only partially addresss the problem. Discrete diffusion models (Austin et al., 2021; Lou et al., 2023) treat counts as unordered categories through masking or uniform noise. Blackout Diffusion (Santos et al., 2023), the only count-specific approach, uses pure-death processes that cannot transport between arbitrary distributions. The biological deconvolution literature on the other hand focuses on deconvolving cell-type (cluster-level) proportions (Kleshchevnikov et al., 2022; Cable et al., 2022; Li et al., 2023), rather than unit-level count profiles. Thus, there is need for a framework that respects the integer and ordinal structure of counts, enables transport between arbitrary distributions, and can systematically deconvolve aggregated observations.

We introduce Count Bridges: a stochastic bridge process on $\mathbb{Z}^d$ using Poisson birth-death dynamics. This yields closed-form conditionals for exact sampling and extends naturally to deconvolution via an EM algorithm treating unit-level counts as latent. The birth-death mechanism allows transport between arbitrary integer-valued distributions while preserving the ordinal structure, as both increments and decrements respect the natural ordering of counts. We show that Count Bridges outperform existing methods on synthetic benchmark datasets and scale more favorably to high-dimensional settings. We then showcase Count Bridges on two real-world biological applications centered on deconvolution: nucleotide-resolution single-cell RNA-sequence modeling for bulk RNA-seq deconvolution and reference-free spatial transcriptomic deconvolution. The codebase is available here.

## 2 BACKGROUND ON DIFFUSION MODELS

Diffusion models specify a time–indexed family of *bridge kernels* connecting $X_0 \sim p_0$ to a simple source distribution $X_1 \sim p_1$ (often Gaussian). There are two layers of structure: (i) an *unconditional forward process* $(X_t)_{t \in [0,1]}$ with kernels $K_{t|0}(x_t \mid x_0) = \text{Law}(X_t \mid X_0 = x_0)$; (ii) for any $0 \leq s \leq t \leq 1$, a family of *bridge kernels* $K_{s|0,t}(x_s \mid x_0, x_t) = \text{Law}(X_s \mid X_0 = x_0, X_t = x_t)$.

Diffusion models require two consistency properties. First we require a bridge consistency identity.

$$\text{For any } 0 \leq s \leq t \leq u \leq 1, \quad K_{s|u}(x_s \mid x_u) = \int K_{s|t}(x_s \mid x_t) \, K_{t|u}(x_t \mid x_u) \, dx_t. \tag{1}$$

Thus multi-step sampling along any grid $u \to t \to s$ matches the single-step $u \to s$ bridge.

Second, the kernel must have a projective posterior:

$$K_{s|t}(x_s \mid x_t) = \int q_{0|t}(x_0 \mid x_t) \, K_{s|0,t}(x_s \mid x_0, x_t) \, dx_0, \tag{2}$$

where $q_{0|t}(x_0 \mid x_t) = \text{Law}(X_0 \mid X_t = x_t)$. This identity expresses $K_{s|t}$ as a mixture over the posterior of the $p_0$ data. It is essential for denoising: during sampling, each predicted $X_t$ changes the posterior $q_{0|t}$, so the reverse kernels must be projective under this posterior update.

Together, equation 1 and equation 2 lets us define a general diffusion approach. First we train a denoiser $q_\theta$ that approximates the posterior, $\tilde{X}_0 \sim q_\theta(\cdot \mid x_t, t) \approx \text{Law}(X_0 \mid X_t = x_t)$, using tuples $(t, X_t, X_0)$ drawn from the "global" bridge: sample $x_0 \sim p_0$, $x_1 \sim p_1$, $t \sim \text{Unif}[0,1]$ and then $X_t \sim K_{t|0,1}(\cdot \mid x_0, x_1)$.

For sampling, pick a grid $1 = t_K > \cdots > t_0 = 0$, draw $X_1 \sim p_1$, set $X_{t_K} \leftarrow X_1$, sampling

$$\tilde{X}_0^{(k+1)} \sim q_\theta(\cdot \mid X_{t_{k+1}}, t_{k+1}), \qquad X_{t_k} \sim K_{t_k|0,t_{k+1}}(\cdot \mid \tilde{X}_0^{(k+1)}, X_{t_{k+1}}). \tag{3}$$

By our consistency properties, this multi–step procedure is equivalent to sampling directly from the $(0,1)$ bridge, so the model cannot drift out of the training distribution.

### 2.1 DIFFUSION AS A BRIDGE BETWEEN NOISE AND DATA

Let us consider the unconditional $K_{t|0}$ process $(X_t)_{t \in [0,1]}$ of the following form

$$X_t = \alpha(t)X_0 + B_t, \tag{4}$$

where $(B_t)_{t \in [0,1]}$ is a $d$-dimensional Gaussian process with non-decreasing standard deviation $\sigma(t)$, and $\alpha(t)$ a non-increasing function. Note that $\alpha(0) = 1$ and $\sigma(0) = 0$.

We want to define a process that interpolates smoothly between $X_0 \sim p_0$ and $X_1$ given by another distribution as in Peluchetti (2023); Albergo et al. (2023); Delbracio & Milanfar (2023); Liu et al. (2022; 2023). We have the following proposition defining the global and local bridge.

**Proposition 2.1.** *Let $(X_t)_{t \in [0,1]}$ be given by equation 4. For $0 < s < t \leq 1$, consider $(X_s)_{s \in [0,t]}$ conditioned on $X_t = x_t$ and $X_0 = x_0$. Then the conditional law $K_{s|0,t}(\cdot \mid x_0, x_t)$ is Gaussian and can be written*

$$X_s \stackrel{d}{=} \alpha(s)(1 - r(s,t))X_0 + \frac{\alpha(s)}{\alpha(t)}r(s,t)X_t + \sigma(s)(1 - r(s,t))^{1/2}Z, \tag{5}$$

*where $Z \sim \mathcal{N}(0, \text{Id})$ is independent of $(X_0, X_t)$ and $r(s,t) = \frac{\alpha(t)^2 \sigma(s)^2}{\alpha(s)^2 \sigma(t)^2}$. In particular, the family $\{K_{s|0,t}\}_{0 \leq s \leq t \leq 1}$ defined by equation 5 satisfies equations 1 and 2.*

Note that if $X_1 \sim \mathcal{N}(0, \text{Id})$, $\alpha(1) = 0$ and $\sigma(1) = 1$ we have $X_t \stackrel{d}{=} \alpha(t) X_0 + \sigma(t) Z$. Furthermore, our equation 5 recovers the interpolation described in Albergo et al. (2023) with the identification $\alpha(t) \to \alpha(t)(1 - r(t))$, $\frac{\alpha(t)}{\alpha(1)} r(t) \to \beta(t)$ and $\sigma(t)(1 - r(t))^{1/2} \to \gamma_t$.

## 2.2 Sampling the Posterior

In this paradigm the bridge is only the first of two choices that define the model. We also have to choose how to model the posterior $X_0 | X_t, t$. There are two core options: we can use differential equations to model the posterior in the limit of small steps or we can focus more directly on modeling the posterior. In Euclidean space, the former lets us learn a simple conditional expectation, whereas the latter always requires a distribution model.

**Infinitesimal.** Consider a small backward step of size $\delta > 0$. The local bridge between times $t$ and $t - \delta$ is Gaussian, so conditioned on $X_t = x$ we can write to first order in $\delta$

$$X_{t-\delta} \mid X_t = x \approx x - \delta \, b(x, t) + \sqrt{\delta} \, \xi_t, \qquad \xi_t \sim \mathcal{N}\big(0, \Sigma(x, t)\big),$$

where $b$ is the reverse-time drift and $\Sigma$ is the diffusion covariance of the bridge.

The conditional law $X_{t-\delta} \mid X_t$ is Gaussian and can be computed in closed form:

$$b(x, t) = B_1(t) \, x + B_2(t) \, \mathbb{E}[X_0 \mid X_t = x] + b_0(t), \qquad \Sigma(x, t) = \Sigma_0(t).$$

The diffusion covariance depends only on $t$ (from the Brownian increment), and the drift depends on the posterior $\text{Law}(X_0 \mid X_t)$ only through its mean. This justifies learning the mean $q_\theta(x, t) \approx \mathbb{E}[X_0 \mid X_t = x]$ (equivalently, a score or velocity) as in standard diffusion models (Song et al., 2020).

**Distributional.** Following De Bortoli et al. (2025) we can learn the conditional law $q_\theta(\cdot \mid x_t, t) \approx \text{Law}(X_0 \mid X_t = x_t)$, using any distribution learning approach. We can then sample and directly plug into the bridge

$$\tilde{X}_0^{(k+1)} \sim q_\theta(\cdot \mid X_{t_{k+1}}, t_{k+1}), \quad X_{t_k} \sim K_{t_k|0, t_{k+1}}\big(\cdot \mid \tilde{X}_0^{(k+1)}, X_{t_{k+1}}\big).$$

to sample the posterior. The distributional perspective is particularly powerful when the infinitesimal perspective fails to admit a simplification to the conditional expectation, which motivates our use of the distributional approach for Count Bridges (see Sec. 3.2). In categorical discrete settings, all approaches are distributional since they are based on cross-entropy losses, see Campbell et al. (2022); Austin et al. (2021); Shi et al. (2024); Sahoo et al. (2024).

## 3 Count Bridges

### 3.1 An integer bridge between distributions

Mirroring Sec. 2, we seek a bridge for integer-valued data. Instead of a Gaussian process, we use a pair of independent Poisson birth/death processes $(B_t)_{t \in [0,1]}$ and $(D_t)_{t \in [0,1]}$ that increment/decrement the counts. We define an increasing "jump-intensity" function $w : [0, 1] \to \mathbb{R}_{\geq 0}$ with $w(0) = 0$, $w(1) = 1$, and then write the cumulative birth/death intensities as $\Lambda_\pm(t) = \lambda_\pm w(t)$ for some $\lambda_\pm > 0$ so $B_t \sim \text{Poi}(\Lambda_+(t))$ and $D_t \sim \text{Poi}(\Lambda_-(t))$. From here we can define the unconditional kernel $K_{t|0}$:

$$X_t = X_0 + B_t - D_t. \tag{6}$$

Denoting the displacement $d_t = X_t - X_0$, the total number of jumps $N_t = B_t + D_t$, and the slack variable $M_t = \min(B_t, D_t)$. Any two of these variables determine the third:

$$N_t = |d_t| + 2M_t, \qquad B_t = \tfrac{1}{2}(N_t + d_t), \qquad D_t = N_t - B_t. \tag{7}$$

From the $(N_t, B_t)$ perspective, Poisson superposition and thinning imply that, conditional on $(N_t, B_t)$ at time $t$, the earlier counts $(N_s, B_s)$ for $s < t$ can be sampled by a Binomial draw for $N_s$ and a Hypergeometric draw for $B_s$. Switching to $(M_t, d_t)$, a Poisson change of variables yields the slack posterior $M_t \mid d_t$, whose pmf has Bessel form (see Prop. A.6 in App. A). These two ingredients together give a count analogue of Proposition 2.1; the full derivation is in App. A.

**Proposition 3.1.** *Let $(X_t)_{t \in [0,1]}$ be given by equation 6. Now, consider $(X_s)_{s \in [0,t]}$ conditioned by $X_t = x_t$ and $X_0 = x_0$. Then, we have the Poisson Birth-Death bridge $K_{s|0,t}$:*

$$X_s \stackrel{d}{=} X_0 + B_s - D_s, \tag{8}$$

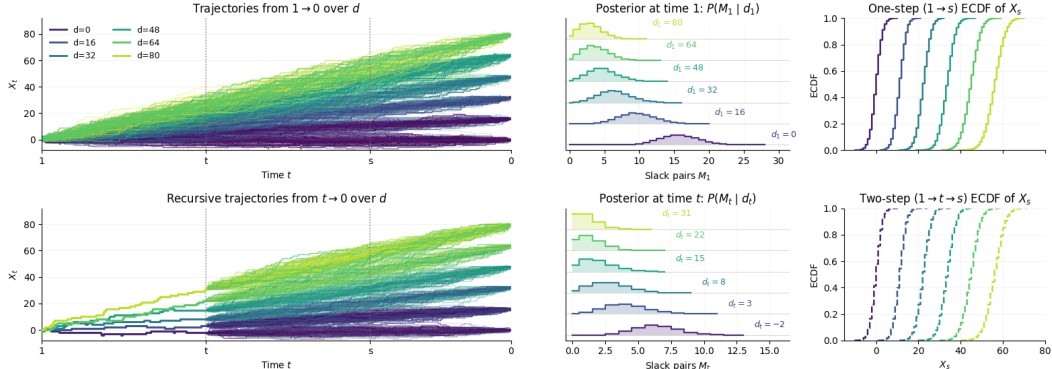

Figure 1: *Left:* Sample paths for several endpoint gaps $d_1$ (top). Fixing the prefix $[0, t]$ resample $(t, 1]$ by the recursive kernel (bottom). *Middle:* Bessel slack posteriors at initial and intermediate times. The slack $M_t$ concentrates near 0 as $|d|$ grows. *Right:* ECDFs of $X_s$ from a one–step kernel $(1 \to s)$ and a two–step kernel $(1 \to t \to s)$ are indistinguishable, confirming composition.

*where we condition on $d_t = X_t - X_0$ and sample $M_t \mid d_t \sim Bes(|d_t|; \Lambda_+(t), \Lambda_-(t))$, changing variables to $N_t$ and $B_t$ to sample $B_s$, and $D_s$ which we can plug into equation 8:*

$$N_s \mid N_t \sim \mathrm{Bin}\left(N_t, \frac{w(s)}{w(t)}\right), \quad B_s \mid (N_t, N_s, B_t) \sim \mathrm{Hyp}(N_t, B_t, N_s), \quad D_s = N_s - B_s. \quad (9)$$

*The family $\{K_{s|0,t}\}_{0 \le s \le t \le 1}$ defined by equation 8 satisfies equations 1 and 2.*

We visualize this process in Fig. 1 where we show the trajectories for the one- and two-step models along with the core composition property that drives bridge models. This setup enables training and sampling from a Count Bridge, see Algorithms 1 and 2. These results leverage our custom CUDA kernel implementing the fast Bessel sampler of Devroye (2002) to enable sampling at scale.

In Fig. 1 we also see that as $d_t$ grows the slack $M_t$ concentrates near zero, so there is no slack. This means that Count Bridges are an instance of the static Schrödinger bridge problem (Léonard, 2013): they solve an entropy-regularized optimal transport. Let $\kappa = \sqrt{\lambda_+ \lambda_-}$ be the jump intensity and $p_{\mathrm{ref}}^\kappa(x_0, x_1) = p_0(x_0) K_{1|0}^\kappa(x_1|x_0)$ be the joint law of $(X_0, X_1)$ induced by the kernel. Over the space of couplings $\mathcal{C}(p_0, p_1) = \{C \text{ on } \mathcal{X} \times \mathcal{X} : C(\cdot, \mathcal{X}) = p_0, \ C(\mathcal{X}, \cdot) = p_1\}$, Count Bridges solve

$$C_\kappa \in \arg \min_{C \in \mathcal{C}(p_0, p_1)} \mathrm{KL}\left(C \,\|\, p_{\mathrm{ref}}^\kappa\right).$$

Letting $\kappa \to \infty$ yields the independent coupling $p_0 \otimes p_1$, but as $\kappa \downarrow 0$ we obtain

$$\mathrm{KL}\left(C \,\|\, p_{\mathrm{ref}}^\kappa\right) \ \approx \ \log\left(\tfrac{2}{\kappa}\right) \mathbb{E}_C |X_1 - X_0| - H(C),$$

so $\kappa \to 0$ recovers discrete OT with cost $|x_1 - x_0|$ (see App. A.2).

This echoes the Gaussian case (Sec. 2) where we define $\sigma = \sigma(1)$ and $p_{\mathrm{ref}}^\sigma$, and as $\sigma \downarrow 0$

$$\mathrm{KL}\left(C \,\|\, p_{\mathrm{ref}}^\sigma\right) \ \approx \ \tfrac{1}{2\sigma^2} \mathbb{E}_C \|X_1 - X_0\|^2 - H(C),$$

so $\sigma \to 0$ recovers quadratic OT, while $\sigma \to \infty$ again gives $p_0 \otimes p_1$ (Shi et al., 2023). Thus the bridge parameters $\kappa$ (count) and $\sigma$ (Gaussian) play the same role as entropy–regularization strengths.

### 3.2 DISTRIBUTIONAL SCORING LOSS FOR THE DENOISER

Training requires a distributional loss due to the discrete nature of the space. As shown by Holderrieth et al. (2024), the ELBO for discrete generators (e.g., jump processes) is distributional and cannot be reduced to expectations over point estimates. This mirrors the need for cross-entropy in discrete diffusion and flow models. We can use cross-entropy with Count Bridges (we test this, see App. D.1), but it has two issues: first, it does not incorporate the lattice structure; second, cross-entropy cannot model the joint of $X_s \mid X_t$ without exponential cost in dimension, so cross entropy is usually

**Require:** dataset $(\mathbf{x}_0, \mathbf{x}_1)$, $w(\cdot)$, $\Lambda_\pm(\cdot)$
1: **for** each minibatch **do**
2:     sample $(x_0, x_1) \sim (\mathbf{x}_0, \mathbf{x}_1)$
3:     $t \sim \mathrm{Unif}[0, 1]$
4:     $d_1 \leftarrow x_1 - x_0$
5:     $M_1 \sim \mathrm{Bes}(|d_1|; \Lambda_+(1), \Lambda_-(1))$
6:     $N_1 \leftarrow |d_1| + 2M_1$
7:     $B_1 \leftarrow \frac{1}{2}(N_1 + d_1)$
8:     $N_t \sim \mathrm{Bin}(N_1, w(t))$
9:     $B_t \sim \mathrm{Hyp}(N_1, B_1, N_t)$
10:     $x_t \leftarrow x_1 - 2(B_1 - B_t) + (N_1 - N_t)$
11:     update $\theta$ on $\mathcal{L}(\theta)$
12: **end for**

Algorithm 1: Training Poisson–BD Bridge

**Require:** $x_{t_K} = x_1$, model $q_\theta$, $w(\cdot)$, $\Lambda_\pm(\cdot)$
1: **for** $k = K, K-1, \dots, 1$ **do**
2:     sample $\hat{x}_0 \sim q_\theta(\cdot \mid x_{t_k}, t_k)$
3:     $d_{t_k} \leftarrow x_{t_k} - \hat{x}_0$
4:     $M_{t_k} \sim \mathrm{Bes}(|d_{t_k}|; \Lambda_+(t_k), \Lambda_-(t_k))$
5:     $N_{t_k} \leftarrow |d_{t_k}| + 2M_{t_k}$
6:     $B_{t_k} \leftarrow \frac{1}{2}(N_{t_k} + d_{t_k})$
7:     $r \leftarrow w(t_{k-1})/w(t_k)$
8:     $N_{t_{k-1}} \sim \mathrm{Bin}(N_{t_k}, r)$
9:     $B_{t_{k-1}} \sim \mathrm{Hyp}(N_{t_k}, B_{t_k}, N_{t_{k-1}})$
10:     $x_{t_{k-1}} \leftarrow x_{t_k} - 2(B_{t_k} - B_{t_{k-1}}) + (N_{t_k} - N_{t_{k-1}})$
11: **end for**
12: **return** $x_{t_0}$

Algorithm 2: Sampling Poisson–BD Bridge

factorized, modeling each coordinate of $X_s \mid X_t$ independently or autoregressively. Specializing to count data we can go beyond cross-entropy by using a proper scoring rule that (i) incorporates the geometry and (ii) enables modeling of the joint.

Formally, let $(X_0, X_t)$ denote a training pair from $K_{t|0,1}$ at time $t \in [0, 1]$, and let $q_\theta(\cdot \mid x_t, t)$ be our denoiser. We train $q_\theta$ using a strictly proper distributional scoring rule (Gneiting & Raftery, 2007; De Bortoli et al., 2025). Fix a negative-type semimetric $\rho$ on $\mathbb{Z}^D$ (all our experiments focus on the $\rho(x, x') = \|x - x'\|_2^\beta$ with $\beta = 1$). For any distribution $p$ and outcome $y$, the energy score is

$$S_\rho(p, y) = \tfrac{1}{2} \mathbb{E}_{X,X' \sim p}[\rho(X, X')] - \mathbb{E}_{X \sim p}[\rho(X, y)] \text{ and } \mathcal{L}(\theta) = \mathbb{E}_{X_0, X_t, t}\Big[S_\rho\big(q_\theta(\cdot \mid X_t, t), X_0\big)\Big]$$

which is strictly proper when $\rho$ is characteristic. Taking $m$ i.i.d. samples $\hat{x}^{(j)} \sim q_\theta(\cdot \mid x_t, t)$ we can use the plugin estimator: $\widehat{S}_\rho = \frac{1}{m(m-1)} \sum_{j \neq j'} \frac{1}{2} \rho(\hat{x}^{(j)}, \hat{x}^{(j')}) - \frac{1}{m} \sum_{j=1}^m \rho(\hat{x}^{(j)}, x_0)$.

## 4 DECONVOLUTION WITH COUNT BRIDGES

We extend Count Bridges to handle unit–level generation when we only observe aggregates. Consider $G$ units in the one-dimensional case where the group-level state at time $t$ is a vector $\mathbf{X}_t \in \mathbb{Z}^G$ with entries $X_{gt}$ for unit $g$ at time $t$. Each entry evolves independently according to the bridge in Section 3. The key challenge: we observe the unit–level endpoint $\mathbf{x}_1$ but only the aggregate at time 0, $a_0 = \sum_{g=1}^G x_{g0} \in \mathbb{Z}$, not the unit–level vector $\mathbf{x}_0$. Our goal is to learn a count bridge $q_\theta(x_0 \mid x_t, t, z)$ that generates unit–level endpoints given start data at time $t = 1$ and side information $z$.

We formulate this as a generalized EM problem, similar to Rozet et al. (2024), where $\mathbf{X}_0$ is latent and $a_0 = \sum_g \mathbf{X}_{g0}$ is observed. Let $A : \mathbb{Z}^G \to \mathbb{Z}$ be a linear aggregate map (e.g., sums across units, block sums). For $(x_t, t, z)$, the denoiser $q_\theta(\cdot \mid x_t, t, z)$ defines an i.i.d. product prior over $\mathbf{X}_0 = (X_{10}, \dots, X_{G0})$. Conditioning on the aggregate yields

$$Q_\theta(\mathbf{X}_0 \mid a_0, x_t, t, z) \propto \Big[ \prod_{g=1}^G q_\theta(X_{g0} \mid x_t, t, z) \Big] \mathbf{1}\{A(\mathbf{X}_0) = a_0\}.$$

In the E-step we will generate "latent" $x_0^{\approx}$ using the model and in the M-step we will use these $x_0^{\approx}$ to train the model at the aggregate level. We summarize the overall procedure in Algorithms 3 and 4.

**E-Step** The ideal E–step would sample from the exact aggregate–conditional law

$$\mathbf{X}_0^\star \sim Q_\theta(\cdot \mid a_0, x_t, t, z).$$

We could then use the sampled $\mathbf{x}_0^\star$ as latent variables to sample $x_t$ between $(\mathbf{x}_0^\star, \mathbf{x}_1)$ using the unit–level kernel $K_{t|0,1}$ from Prop. 3.1.[1] Unfortunately, $Q_\theta$ is generally intractable to sample from,

---

[1]The same method described here can be used with distributional diffusion on continuous space, but we focus on counts since most often when we observe aggregates we believe they are based on discrete underlying data.

**Require:** $(\mathbf{x}_1, a_0, z), w(\cdot), \Lambda_{\pm}(\cdot), q_\theta, \Pi$
1: **for** $k = K, K-1, \ldots, 2$ **do**
2:      Sample $\hat{\mathbf{x}}_{0,t_k} \sim q_\theta(\cdot \mid \mathbf{x}_{t_k}, t_k, z)$
3:      $\tilde{\mathbf{x}}_{0,t_k} \leftarrow \Pi(\hat{\mathbf{x}}_{0,t_k}, a_0, z)$
4:      Update $\mathbf{x}_{t_{k-1}}$ by running the reverse step
5:          using steps 4–10 of Alg. 2, with $\tilde{\mathbf{x}}_{0,t_k}$
6: **end for**
7: $\mathbf{x}_0^{\widetilde{\approx}} \leftarrow$ sample and project $\hat{\mathbf{x}}_{0,t_1}$
8: **return** $\mathbf{x}_0^{\widetilde{\approx}}$

Algorithm 3: Guided Sampling to for $x_0^{\widetilde{\approx}}$

**Require:** $(\mathbf{x}_1, a_0, z), w(\cdot), \Lambda_{\pm}(\cdot), q_\theta, \Pi$
1: **for** each minibatch **do**
2:      **E-step:** Sample latent $\mathbf{x}_0^{\widetilde{\approx}}$ from
3:          $\mathbf{x}_1$ conditional on $a_0$ via Alg. 3
4:      **M-step:** $t \sim \mathrm{Unif}[0,1]$
5:      Sample $\mathbf{x}_t$ via the forward bridge on
6:          $(\mathbf{x}_0^{\widetilde{\approx}}, \mathbf{x}_1)$ using steps 4–10 of Alg. 1
7:      Update $\theta$ using the gradient of $-\mathcal{L}_{\mathrm{agg}}(\theta)$
8: **end for**

Algorithm 4: Training with Aggregate Supervision

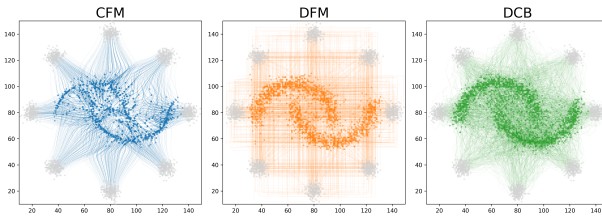

Figure 2: A scaled and rounded variant of the classic 8 gaussian to two moons task. Here we compare the trajectories of continuous flow matching, discrete flow matching, and count bridges. CB achieves the lowest $W_2$, MMD, and EMD, see Table 6.

given just a unit-level model, so we approximate it through the diffusion sampling process itself. Starting from $\mathbf{x}_1$, we run the sampling process as in Algorithm 2, but at each timestep $t_k$ we: (1) predict $\hat{\mathbf{x}}_0 \sim q_\theta(\cdot \mid \mathbf{x}_{t_k}, t_k, z)$, (2) project $\hat{\mathbf{x}}_0$ to satisfy the aggregate constraint (see Sec. 4), yielding $\tilde{\mathbf{x}}_0$, and (3) perform the sampling step using $\tilde{\mathbf{x}}_0$ as the predicted endpoint. This projection–guided diffusion ensures the aggregate constraint is incorporated throughout the denoising trajectory (see Alg. 3). This process produces latent $x_0^{\widetilde{\approx}}$ samples that are consistent with the aggregate constraints, which we can then use in the M-step to train the model.

**M-Step** With these unit-level samples in hand, the M–step runs the bridge process as in Section 3. But instead of computing the loss on the unit-level latents, we compute the loss with respect to the aggregates. Given the ground-truth aggregate $a_0$, we lift the same strictly proper score to aggregates:

$$S_\rho^A(p, a) = \tfrac{1}{2} \mathbb{E}_p\big[\rho(A(\mathbf{X}), A(\mathbf{X}'))\big] - \mathbb{E}_p\big[\rho(A(\mathbf{X}), a)\big] \text{ and } \mathcal{L}_{\mathrm{agg}}(\theta) = \mathbb{E}_{A_0, \mathbf{x}_t, t}\Big[S_\rho^A\big(q_\theta(\cdot \mid \mathbf{X}_t, t, z), A_0\big)\Big]$$

with the plug-in obtained by sampling $\hat{\mathbf{X}}_0^{(j)} \sim q_\theta(\cdot \mid \mathbf{X}_t, t, z)$ and forming $\hat{a}^{(j)} = A(\hat{\mathbf{X}}_0^{(j)})$.

**Approximate Sampling from the conditional distribution** Given a predicted endpoint $\hat{\mathbf{x}}_0$ from our diffusion model and target aggregate $a_0$, we need to sample from the conditional distribution $Q_\theta(\cdot \mid A(\mathbf{X}_0) = a_0)$. While this is intractable, we can derive a principled approximation.

**Proposition 4.1** (First–order aggregate projection). *Let $A(\mathbf{X}_0)$ be the aggregate, and let $p_0$ be the prior law of $\mathbf{X}_0$. Under the regularity conditions in App. B.1, the aggregate–conditional law $Q_\theta(\cdot \mid A_0 = a_0)$ admits a first–order exponential tilt. The corresponding generalized KL projection*

$$\Pi(\mathbf{x}_0) = \arg \min_{\mathbf{y}_0 : A(\mathbf{y}_0) = a_0} D_{\mathrm{KL}}(\mathbf{y}_0 \| \mathbf{x}_0)$$

*gives a kind of first–order approximation to $Q_\theta(\cdot \mid A_0 = a_0)$. For an elementwise sum $A(\mathbf{x}_0) = \sum_g x_{g0}$ this projection is the simple scaling $\Pi(x_0)_g = a_0 x_{g0}/(\sum_{g'} x_{g'0})$.*

The proposition shows that the natural rescaling operation is not ad hoc, but can be justified as a kind of first-order approximation to the true conditional distribution in a large sample regime (see Appendix B.1). When unit-level training data exist, we can learn a projection $\Pi_\psi(\hat{\mathbf{x}}_0, z, a_0)$ that actually enables sampling conditional on the mean. See Sec. 6 where we show how to learn such a projection.

## 5   RELATED WORKS

**Stochastic interpolants.** Our formulation allows us to transport any integer-valued distribution $p_1$ to another integer-valued distribution $p_0$. In the case of Euclidean state space early works such

as (De Bortoli et al., 2021; Vargas et al., 2021; Chen et al., 2021) have shown how to perform such an interpolation leveraging (Entropic) Optimal transport and the concept of Schrödinger Bridges. In more recent works, ignoring the Optimal Transport constraints, several works have proposed to bridge distributions in a more relaxed formulation leveraging the concept of Markov projection, see Peluchetti (2023); Albergo et al. (2023); Delbracio & Milanfar (2023); Liu et al. (2022; 2023) for instance. Those frameworks can be shown to be strictly equivalent to diffusion models in the case where one of the end distribution is a unit Gaussian, see Gao et al. (2025). However, those works are limited to the Euclidean setting, and extension to the integer-valued setting is required.

**Discrete diffusion models.** Recently, with the advent of language diffusion models such as Ye et al. (2025); Song et al. (2025); Sahoo et al. (2024); Shi et al. (2024); Ou et al. (2024a); Arriola et al. (2025); Nie et al. (2024); Zheng et al. (2023), discrete diffusion models have gained considerable traction. Most works rely on discrete equivalents of the original formulation of diffusion models, explicitly or implicitly replacing the continuous Gaussian noising process by a Continuous-Time Markov Chain (CTMC) (Austin et al., 2021; Campbell et al., 2022; Lou et al., 2023; Campbell et al., 2024; Kitouni et al., 2024; Sun et al., 2023). Other approaches include relying on some Euclidean relaxation (Chen et al., 2022) or modelling the space of probability (Avdeyev et al., 2023; Stark et al., 2024). Similarly, flow matching techniques have been extended to cover this paradigm (Gat et al., 2024). Most of these works focus on *categorical* data and therefore consider uninformed forward process such as uniform or masking process. In contrast, in this work, we focus on ordinal data. To the best of our knowledge, the only existing work that also deals with such a process is Blackout Diffusion (Santos et al., 2023), which considers a pure-death process where an image is taken to the all-zero limit, as opposed to an endpoint conditioned bridge. Our approach generalizes this setup in two ways: first, we allow births and deaths at every time, recovering their pure birth construction in the limit as $\kappa \to 0$; second, we generalize the process to a bridge which can transport $X_1$ to $X_0$.

Finally, we highlight that diffusion models have been extended to the very general setting where only an *infinitesimal generator* is available Benton et al. (2024); Holderrieth et al. (2024). While our work can be seen as an instanciation of this general framework, these general frameworks do not give any information regarding the design of the forward process for integer-valued data, the specific parameretization in terms of slack variables and the necessity of the distributional diffusion loss.

**Distributional Diffusion Models.** In De Bortoli et al. (2025); Shen et al. (2025), the authors learn the conditional distribution $p_{0|t}(x_0|x_t)$ through the use of scoring rules, going beyond the classical training framework of diffusion, which approximates the conditional mean $\mathbb{E}[X_0|X_t = x_t]$. The importance of approximating the covariance was already noted by Nichol & Dhariwal (2021) and further analyzed in (Ho et al., 2020; Nichol & Dhariwal, 2021; Bao et al., 2022a;b; Ou et al., 2024b). In a similar flavor (Xiao et al., 2022) uses a GAN to approximate $p_{0|t}(x_0|x_t)$.

**Sequence-to-expression models** An ambitious goal in biology is to predict gene expression from DNA sequence information. There have been several attempts to train deep learning models for sequence-to-expression prediction tasks (Barbadilla-Martínez et al., 2025), including Enformer (Avsec et al., 2021), a state-of-the-art transformer-based DNA sequence model. While powerful, Enformer, like the vast majority of sequence-to-expression models, was trained on bulk gene expression data and is not able to predict single-cell expression profiles, missing the cellular heterogeneity and fine-grained regulatory patterns that shape tissue function.

**Spatial transcriptomic deconvolution** Spatial transcriptomics encompasses a family of recently developed techniques which measure gene expression and spatial location in tissues. The majority of these techniques are not capable of resolving individual cells, instead providing aggregate information over small neighborhoods consisting of on the order of tens of cells (Moses & Pachter, 2022). To address this limitation, a number of deconvolution methods have been developed to infer single-cell level information (Li et al., 2023). The majority of these methods, including cell2location (Kleshchevnikov et al., 2022) and RCTD (Cable et al., 2022), require a paired non-spatially resolved scRNA-seq atlas, and output cluster-level mixture proportions rather than single cell counts. The ideal deconvolution would recover full single-cell count profiles directly from spatial data without requiring external reference atlases. DestVI (Lopez et al., 2022), which outputs count profiles but requires a reference, and STDeconvolve (Miller et al., 2022) which does not require a reference but outputs cluster-level predictions, both take steps toward this goal.

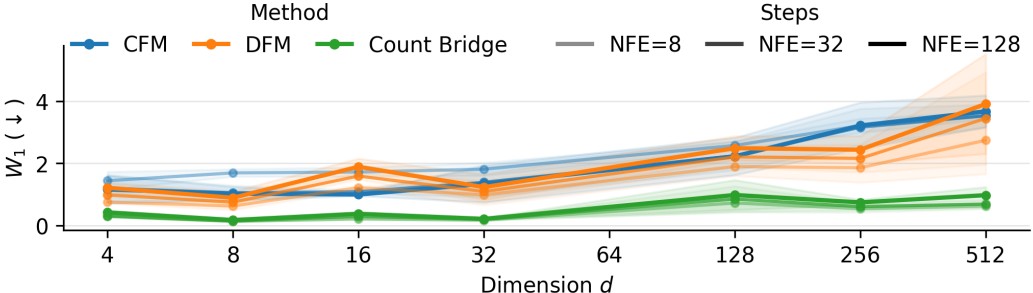

Figure 3: CFM, DFM, and CB on our low-rank mixture of Gaussians transport experiment across dimensions and NFE. See App. D.2 for full details.

## 6 APPLICATIONS

We evaluate with three distributional metrics: the Energy score, the Wasserstein-2 distance, and the MMD (RBF). For deconvolution, we evaluate cell-type proportion predictions using RMSE, the Jensen-Shannon Divergence (JSD), and Spearman correlation following Li et al. (2023). Synthetic tasks have std. errors over 3 training seeds; main applications have std. errors 3 over inference seeds.

### 6.1 SYNTHETIC DISTRIBUTIONS

Here, we benchmark count bridges (CB) against continuous flow matching (CFM) (Lipman et al., 2022) and discrete flow matching (DFM) (Gat et al., 2024) across a range of synthetic experiments.

**Discrete 8-Gaussians to 2-Moons.** We adapt this classic task to the integers. We plot the learned trajectories in Fig 2. Qualitatively CB achieves the best performance. DFM is much more competitive in this experiment than CFM, but DFM trajectories are decoupled from the underlying geometry, whereas CB produces OT-like trajectories similar to CFM. These qualitative evaluations are confirmed quantitatively: CB achieves the best performance across $W_2$, Energy, and MMD (see App. D.1).

**Scaling in Low-Rank Gaussian Mixtures.** To test scalability to higher dimensions, we construct integer-valued datasets with fixed intrinsic dimensionality while ambient dimension $d$ increases in powers of two from 4 to 512. Each dataset is a 5-component Gaussian mixture with latent rank $r = 3$, projected to $\mathbb{Z}^d$. In Fig. 3 see that CB has the best scaling in dimensionality (see App. D.2 for more).

**Deconvolution of Gaussian Mixtures.** We extend the low-rank mixture task to evaluate deconvolution capabilities. In this experiment, each observation is an aggregate constructed by summing a group of $G$ samples. For each group, the $G$ samples are drawn from a group-specific Gaussian mixture whose component weights are sampled from a Dirichlet distribution with concentration parameters $(\alpha_1, \ldots, \alpha_5)$. The labels of the $G$ source components are provided as unit-level side information. We then vary the size of the group $G$ and the extent of variation between groups by changing the concentration parameter $\alpha$ (see Appendix D.3 for details). In Fig. 4 we see performance degrades as groups become more uniform and larger. We explore the theoretical limits to deconvolution in Apps. B.2 and

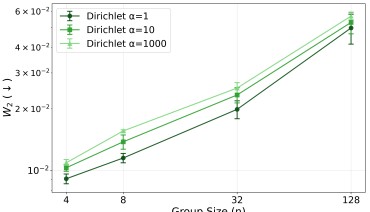

Figure 4: Deconvolution of the low-rank Gaussian mixture across different group sizes and levels of between-group heterogeneity.

B.3, which confirm that deconvolution requires between-group heterogeneity to enable identification, which is inherently lost as groups become large. Despite these limits, we demonstrate practical deconvolution on moderately-sized groups in our spatial transcriptomics application (Section 6.3).

### 6.2 MODELLING GENE EXPRESSION AT SINGLE-CELL AND SINGLE-NUCLEOTIDE RESOLUTION

A central goal in biology is to understand the relationship between DNA sequence and gene expression. Many models relate sequence and expression, the most prominent of which, such as Enformer (Avsec

| Method | Bulk MSE | CT MSE | Comparison | MMD | $W_2$ | Energy |
|---|---|---|---|---|---|---|
| Fine-tuned Enformer | 2.590 | 3.142 | Bulk mean | 0.515 | 0.208 | 56.800 |
| Count Bridge | **0.601** ±0.000 | **1.410** ±0.002 | Count Bridge | **0.446** ±0.000 | **0.182** ±0.001 | **28.583** ±0.003 |

Table 1: Nucleotide-level MSE for bulk and bulked cell-type (CT) specific predictions.

Table 2: Gene expression count profile deconvolution error for bulk RNA sequencing data.

et al., 2021), are Transformer-based models that predict expression from sequence. More recent work has explored fine-tuning Enformer on single-cell data (Hingerl et al., 2024). On the other hand, there is a mature literature on deconvolving bulk RNA-seq (Newman et al., 2019; Wang et al., 2019). These methods operate at the gene (rather than nucleotide) level, leveraging bulk cell-type profiles or single-cell references to deconvolve bulk profiles into cell-type proportions (not count profiles).

We use CBs to jointly model sequence and single-cell expression counts in scRNA-seq data, and to enable nucleotide-level deconvolution of bulk profiles. To validate CBs in this setting, we demonstrate two key results. First, we show that CBs trained on single-cell data produce meaningful count profiles and outperform a fine-tuned Enformer model on sequence-to-expression prediction.

| Method | JSD | RMSE | Spearman |
|---|---|---|---|
| CIBERSORTx | 0.194 | 0.109 | 0.079 |
| MuSiC | 0.313 | 0.140 | 0.186 |
| Count Bridge | **0.113** ±0.001 | **0.073** ±0.000 | **0.267** ±0.005 |

Table 3: Cell-type proportion deconvolution error for nucleotide level bulk RNA sequencing data.

Second, we show that conditioning CBs on bulk profiles enables deconvolution of bulk gene expression into inferred single-cell gene expression profiles. We validate these deconvolved profiles distributionally and show that they achieve state-of-the-art performance relative to cell-type proportion deconvolution models.

**Modeling sequence and single-cell counts** We train CBs on PBMC scRNA-seq counts at nucleotide resolution using $10^6$ cells across $10^3$ donors (Yazar et al., 2022). Each training example corresponds to a nucleotide position in a single cell, and is represented by the noisy count $x_t$ and diffusion time $t$ from the CB forward process, a cell-type embedding, a local genomic context $z$ obtained by encoding the surrounding DNA sequence with Enformer, and i.i.d. noise $\zeta$ for the distributional loss. These features are concatenated and passed through residual multi–head attention blocks and a final softplus head that parameterizes the conditional count distribution $X_0|X_t, t, z$. The model is trained directly on unit-level (single-cell) expression profiles rather than only on aggregated counts. During training we randomly mask cell-type labels so that the model supports both unconditional and cell-type-conditional sampling at test time.

**Learned projection for deconvolution** Since we have unit-level data we can learn a better projection operator than the simple rescaling function in Prop. 4.1. We augment the CB with a small projection module $\Pi_\psi$, an attention block operating on each nucleotide (represented by $z$) across cells in the batch. Given an initial CB prediction $\hat{x}_0$, an observed aggregate $a_0$, and the noisy state $x_t$, the module outputs $\tilde{x}_0 = \Pi_\psi(\hat{x}_0, a_0, \mathbf{x}_t, z)$, we train this using the distributional loss to learn to sample $X_0 \mid A(X_0)=a_0, X_t, t$. To support both unconditional and aggregate-conditioned inference, we apply the projection module only on a random 10% of training examples where $a_0$ is provided.

**Bulk gene expression** We first evaluate the ability of our model to predict expression from sequence, both unconditionally and conditional on cell type. As a baseline, we use an Enformer model fine-tuned directly on the PBMC dataset. We find that Count Bridge predictions outperform fine-tuned Enformer (Table 1, for results by cell type and further details see App. E).

**Deconvolved profiles** We can use this unit-level model for deconvolution tasks: we can condition on an aggregate (bulk profile) to sample single-cell profiles from the model while matching that aggregate. We next evaluate the ability of CBs to deconvolve mixtures of cell types from held-out individuals. We held out 10% of patients from our training set and synthetically bulked these patients. Since we have the ground truth data, we can then evaluate deconvolution quality. We first evaluate the distributional quality of these predictions against the bulk mean, further validating the CB count profiles (Table 2). As a more robust set of baselines, we compare to CIBERSORTx (Newman et al., 2019) and MuSiC (Wang et al., 2019). To facilitate comparison, we aggregate our nucleotide-level predictions into gene counts and assign each of our deconvolved cells to the closest

cell type. CBs achieve better performance on JSD, RMSE, and Spearman correlation while providing nucleotide-level counts (Table 3). In App. E we plot the UMAP for qualitative comparison.

### 6.3 DECONVOLVING SPATIAL TRANSCRIPTOMIC SPOTS INTO SINGLE-CELL COUNTS

Next, we show how CBs can be used to infer single cell gene expression profiles from spot-level aggregates in spatial transcriptomic data. In spatial transcriptomic data generated by Visium (Ståhl et al., 2016), it is common to have access to side information beyond the spot-level count aggregates. In particular, many datasets include images of the cells with a nuclear stain (Palla et al., 2022). CBs provide a natural way to leverage this cell-level side information to deconvolve aggregate count data.

**Modeling spatial aggregates** We train CBs on a MERFISH mouse brain dataset (Vizgen, 2021), which is resolved at the single-cell level, and artificially aggregate neighborhoods of cells to simulate spot-level Visium data. This synthetic dataset gives us access to spot-level aggregates and their corresponding single-cell ground truth, as well as single-cell nuclear images. Following the notation in Sec. 4, the spot-level counts can be treated as aggregates $a_0$, and single-cell images can be treated as unit-level side information $z$. In this application, we never observe single-cell count profiles, only spot-level aggregates and the single-cell images. We leverage a UViT (Bao et al., 2023) extended to incorporate count and noise patches (see App. F). We use a simple source distribution $X_1 \sim \text{Poi}(10)$.

**Cell type proportions** We benchmark CBs against STDeconvolve (Miller et al., 2022), a widely used spatial transcriptomic deconvolution method which is state-of-the-art among reference-free approaches Li et al. (2023) (see Appendix F for comparisons to reference-based methods). STDeconvolve outputs cell type (clus-

| Method | JSD | RMSE | Spearman |
|--------|-----|------|----------|
| STDeconvolve | 0.288 | 0.177 | 0.255 |
| Count Bridge | **0.231** | **0.110** | **0.332** |
| | ±0.002 | ±0.001 | ±0.001 |

Table 4: Cell-type proportion deconvolution error for spatial transcriptomics.

ter identity) proportions for each spot rather than single cell counts. As such, we evalute the quality of deconvolution by comparing the predicted cell type proportions to the true cell type proportions per spot. For CBs, which provide single-cell count profile predictions rather than cell type proportions, we assign each predicted count profile its nearest neighbor cell type in order to compare against STDeconvolve. CBs outperforms STDeconvolve on both the JSD and the RMSE (Table 4).

**Count profiles** We next evaluate the quality of the count profiles inferred by CBs. Here, because STDeconvolve does not provide these predictions, we instead consider a simple baseline: predicting the spot-level mean ($a_0/G$) for each cell. This baseline, while seemingly naive, is actually biologically well-motivated. In spatial transcriptomics, cells within a spot represent local tissue organization where neigh-

| Comparison | MMD | $W_2$ | Energy |
|-----------|-----|-------|--------|
| Spot mean | 0.409 | 0.030 | 41.717 |
| Count Bridge | **0.203** | **0.017** | **8.903** |
| | ±0.000 | ±0.000 | ±0.014 |

Table 5: Gene expression count profile deconvolution error for spatial transcriptomics

boring cells coordinate their functions (Armingol et al., 2021). As such, we expect cells in spatial neighborhoods to have correlated gene expression profiles, making the spot mean a reasonable baseline. Nonetheless, CBs outperform the spot-level mean baseline (see Table 5), showing CBs can learn meaningful unit-level distributions from real-world aggregate data. In App. F we provide a more detailed biological evaluation of the cell types and pathways in our generated data, alongside the UMAP to facilitate qualitative comparison.

## 7 CONCLUSION

Count Bridges offer a tractable, discrete-native alternative to continuous diffusion models, unifying direct count generation with deconvolution from aggregates. We demonstrate the power of Count Bridges for nucleotide-level deconvolution of bulk RNA-seq and spatial transcriptomic deconvolution.

**Limitations** (i) When counts are well-approximated as continuous, Euclidean models may match or exceed performance. (ii) Identifiability in pure deconvolution degrades as group sizes grow or between-group heterogeneity shrinks, so our EM procedure is most reliable at moderate aggregation. (iii) The projection step we use is a first-order surrogate and lacks serious theoretical support.

Despite these caveats, Count Bridges lay the groundwork for rigorous discrete generative modeling and invite future work on a deeper understanding of the projection-guided sampler, sharper identifiability bounds, and generally stronger guarantees for projection-guided EM.

**Ethics Statement.** This study uses publicly released, de-identified single-cell and spatial transcriptomics datasets under their respective licenses; no new human subject data were collected, and institutional review board (IRB) approval was therefore not required. We do not foresee serious ethical implications to Count Bridges beyond the risks already posed by standard diffusion/flow matching models. Our deconvolution methods could possibly pose some additional privacy risks. We used LLMs to help draft portions of the code used in our experiments and to edit portions of this manuscript. All our models are intended for research use only, not clinical use. LLMs were not used in any way significantly outside the current norms of academic research.

**Reproducibility Statement.** We have taken significant steps to ensure that all results presented in this work are reproducible. An anonymous source code repository is provided here, containing complete implementations of the Count Bridge framework, including model architectures, training procedures, projection-based deconvolution, and evaluation pipelines. The appendix includes full mathematical derivations and proofs of all theoretical claims. We also provide descriptions of all data preprocessing steps for synthetic benchmarks, PBMC sequence-to-expression prediction, and spatial transcriptomic aggregation, as well as architectural and hyperparameter specifications. Together, these materials are intended to allow independent researchers to fully reproduce our theoretical and empirical findings.

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

# A  COUNT BRIDGES

## A.1  COUNT BRIDGES

To review the setup, we have $X_0 \sim p_0$, $X_1 \sim p_1$, where $X_0, X_1$ lie on a discrete lattice ($\mathbb{Z}^+$). We derive an unconditional forward process kernel $K_{t|0} = P(X_t|X_0)$ and a family of bridge kernels $K_{s|0,t} = P(X_s|X_t, X_0), 0 \leq s \leq t \leq 1$ such that these kernels satisfy the consistency property described in 1.

In this section, we define the model and the associated kernels formally, and show how the Binomial and Hypergeometric distributions emerge in our framework. We prove they satisfy the consistency property thus establishing their correctness as diffusion kernels for count data.

### A.1.1  SAMPLING PROCESS

**Forward Kernel:** Let $w : [0, 1] \to \mathbb{R}_{\geq 0}$ be an increasing "jump–intensity" shape function with $w(0) = 0$ and $w(1) = 1$. Intuitively, $w(t)$ describes the proportion of "jumps" that have occurred by time $t$ for $t \in [0, 1]$. Further, let $\lambda_+, \lambda_- > 0$ be intensities for the birth and death processes respectively. We then define the cumulative birth (+) / death (-) intensities

$$\Lambda_+(t) = \lambda_+ w(t), \qquad \Lambda_-(t) = \lambda_- w(t),$$

We use these jump intensities to define the independent (non-homogeneous) Birth/Death Poisson processes, $(B_t)_{t \in [0,1]}$ and $(D_t)_{t \in [0,1]}$ respectively, such that:

$$B_t \sim \mathrm{Poi}(\Lambda_+(t)), D_t \sim \mathrm{Poi}(\Lambda_-(t))$$

These Random Variables define counts at $t$, $X_t = X_0 + B_t - D_t$, giving us the unconditional forward kernel $K_{t|0}$.

**Bridge Kernels:** To derive $K_{s|0,t}$, we first define three more random variables that arise through the Poisson process.

Let $d_t$ denote the net displacement at time $t$, i.e:

$$d_t = X_t - X_0 \qquad t \in [0,1]$$

Let $N_t$ denote the total number of jumps that have occurred until time $t$, i.e:

$$N_t = B_t + D_t \qquad t \in [0,1]$$

Finally, we define $M_t$, denoting the number of "slack" jumps that cancel each other out during the process. For example, a birth followed by a death results in no net displacement, but result in 1 "slack" jump. By definition:

$$M_t = min(B_t, D_t) \qquad t \in [0,1]$$

It follows from the definitions that any two of the variables define the third, as shown in equation 7.

While $d_t$ is observed, $N_t$ and $M_t$ are not. For the following lemmas, we first assume that $M_t = m$ is known and fixed before deriving a tractable posterior distribution for $M_t$ in the next section. We now condition on the points $X_0 = x_0$, $X_t = x_t$ and $M_t = m$ to obtain:

$$d_t = x_t - x_0, \qquad N_t = |d_t| + 2m \qquad B_t = \frac{1}{2}(N_t + d_t), \qquad D_t = N_t - B_t.$$

We first characterize two properties of $(N_t)_{t \in (0,1)}$:

**Proposition A.1** (Total jumps process and time-rescaling invariance). $(N_t)_{t \in [0,1]}$ *is a (non-homogeneous) Poisson process with cumulative intensity* $\Lambda(t) = (\lambda_+ + \lambda_-)\, w(t)$. *Moreover, conditional on* $N_1 = n$, *the* $n$ *unordered jump times are i.i.d. with cdf*

$$\mathbb{P}(T \le t \mid N_1 \ge 1) = w(t), \qquad t \in [0,1],$$

*hence* $w$ *is precisely the time–rescaling that makes the jump times uniform on* $[0,1]$.

*Proof.* Note that $N_t = B_t + D_t$ is the sum of two independent Poisson processes, and by Poisson superposition principle, is a Poisson process itself with cumulative intensity equal to the sum of the cumulative intensities of the individual processes. The second statement is a direct result of the Probability Integral Transform. □

To compute $K_{s|0,t} = P(X_s = X_0 + B_s - D_s | X_0, X_t)$, we essentially have to compute $(N_s, B_s)$ or $(B_s, D_s)$, both being equivalent to computing the state $X_s$, given the state $X_t$ and $(N_t, B_t)$. For now, we consider the $(0, 1)$ bridge, i.e $K_{t|0,1}$.

This problem can be visualized in the one-dimensional case as follows: Given $(N_1, B_1)$, view the total $N_1$ jumps as a set of $N_1$ points on $[0, 1]$, each labeled $+1$ denoting birth or $-1$ denoting death, with the density governed by the function $w(\cdot)$. Note that these labels are exchangeable, for both processes have the same intensity function. Thus the goal is to find the number of points to the left of $t$ ($N_t$), and the number of those that are $+1$s ($B_t$).

From this perspective, counting how many of the $N_1$ points fall to the left of $t$ is analogous to sampling coloured balls from an urn with replacement (each jump independently lands before $t$ with probability $w(t)$), giving a Binomial draw. Determining how many of those are births is equivalent to drawing a subset of size $N_t$ from the original population with $B_1$ successes without replacement, i.e a Hypergeometric draw. We formalize this:

**Lemma A.2** (Binomial–Hypergeometric structure of Count Bridges). *Fix* $t \in (0, 1]$ *and condition on* $N_1 = n_1$ *and* $B_1 = b_1$. *Then:*

1. Binomial structure. *Each of the $n_1$ jumps independently falls before time $t$ with probability $w(t)$ (Proposition A.1), hence*

$$N_t \mid N_1 = n_1 \ \sim \ \mathrm{Bin}\big(n_1, w(t)\big).$$

2. Hypergeometric structure. *Since the jump labels are exchangeable, the $n_t$ jumps in $(0, t]$ form a uniformly random subset of the $n_1$ total jumps, hence*

$$B_t \mid (N_1, B_1, N_t) = (n_1, b_1, n_t) \ \sim \ \mathrm{Hyp}\big(n_1, b_1, n_t\big).$$

We can generalize this to arbitrary time points. For $0 < s < t < 1$ define $\gamma = w(s)/w(t)$. Conditional on $(N_t, B_t) = (n_t, b_t)$, the $n_t$ jumps in $(0, t]$ form a non-homogeneous Poisson process with cumulative intensity $\Lambda(\cdot)/\Lambda(t)$, so the $N_s$ jumps in $(0, s]$ satisfy

$$N_s \mid N_t = n_t \ \sim \ \mathrm{Bin}\big(n_t, \gamma\big),$$

and, by the same exchangeability argument applied within $(0, t]$,

$$B_s \mid (N_t, B_t, N_s) = (n_t, b_t, n_s) \ \sim \ \mathrm{Hyp}\big(n_t, b_t, n_s\big), \qquad D_s = N_s - B_s.$$

### A.1.2 COMPOSITION CONDITIONAL ON THE SLACK

In this section we prove two elementary lemmas showing that the Binomial and Hypergeometric distributions have a compositional property, and then use these results to show that the bridge satisfies the consistency equation under a fixed and known slack. In the next sections, we derive the posterior distribution of the slack and show that the result maintains for the general case when we "mix" the bridge over the slack posterior.

Binomial composition arises from the observation that each of the $N \in \mathbb{N}$ objects in a binomial draw are treated completely independent of each other. It states that the following two-step process:

1. Given $N$ balls in an urn, classify each ball as "success" or "failure" independently with probability $\theta$. Let $K$ balls be marked "success".

2. Of those $K$ balls, further classify each ball as "success" or "failure" independently again with probability $\eta$. Let $L$ balls be marked "success".

is equivalent to the one-step process of classifying each of the original $N$ balls as "success" or "failure" with probability $\theta\eta$. We prove this formally in the following lemma.

**Lemma A.3** (Binomial composition). *Let $N \in \mathbb{N}$ and $\theta, \eta \in [0, 1]$. Let $K \sim \mathrm{Bin}(N, \theta)$ and conditional on $K$, let $L \sim \mathrm{Bin}(K, \eta)$. Then $L \sim \mathrm{Bin}(N, \theta\eta)$.*

*Proof.* We begin by writing the joint distribution directly from the model:

$$\mathbb{P}(K = k, \ L = \ell) = \mathbb{P}(K = k)\,\mathbb{P}(L = \ell \mid K = k) = \binom{N}{k}\theta^k(1-\theta)^{N-k}\binom{k}{\ell}\eta^\ell(1-\eta)^{k-\ell}.$$

Apply the standard combinatorial identity

$$\binom{N}{k}\binom{k}{\ell} = \binom{N}{\ell}\binom{N-\ell}{k-\ell},$$

to obtain the factorization

$$\mathbb{P}(K = k, \ L = \ell) = \binom{N}{\ell}(\theta\eta)^\ell(1-\theta\eta)^{N-\ell}\binom{N-\ell}{k-\ell}\left(\tfrac{\theta(1-\eta)}{1-\theta\eta}\right)^{k-\ell}\left(\tfrac{1-\theta}{1-\theta\eta}\right)^{(N-k)}.$$

To get the marginal law of $L$, sum the joint pmf over all $k \geq \ell$.

$$\mathbb{P}(L = \ell) = \sum_{k=\ell}^{N}\binom{N}{\ell}(\theta\eta)^\ell(1-\theta\eta)^{N-\ell}\binom{N-\ell}{k-\ell}\left(\tfrac{\theta(1-\eta)}{1-\theta\eta}\right)^{k-\ell}\left(\tfrac{1-\theta}{1-\theta\eta}\right)^{(N-k)}.$$

Let $m = k - \ell$; then $m$ ranges from $0$ to $N - \ell$, and

$$\mathbb{P}(L = \ell) = \sum_{m=0}^{N-\ell} \binom{N}{\ell} (\theta\eta)^\ell (1 - \theta\eta)^{N-\ell} \binom{N-\ell}{m} \left( \frac{\theta(1-\eta)}{1-\theta\eta} \right)^m \left( \frac{1-\theta}{1-\theta\eta} \right)^{(N-\ell)-m}.$$

Let

$$a = \frac{\theta(1-\eta)}{1-\theta\eta}, \qquad b = \frac{1-\theta}{1-\theta\eta}.$$

and factor out all terms not depending on $m$:

$$\mathbb{P}(L = \ell) = \binom{N}{\ell} (\theta\eta)^\ell (1 - \theta\eta)^{N-\ell} \sum_{m=0}^{N-\ell} \binom{N-\ell}{m} a^m b^{(N-\ell)-m},$$

Since $a + b = 1$, the inner sum is exactly the Binomial theorem,

$$\sum_{m=0}^{N-\ell} \binom{N-\ell}{m} a^m b^{(N-\ell)-m} = (a+b)^{N-\ell} = 1.$$

Thus

$$\mathbb{P}(L = \ell) = \binom{N}{\ell} (\theta\eta)^\ell (1 - \theta\eta)^{N-\ell},$$

which is the pmf of $\mathrm{Bin}(N, \theta\eta)$. $\qquad\square$

We now show hypergeometric composition:

**Lemma A.4** (Hypergeometric composition). *Let $\mathcal{N}$ denote a population of $N \in \mathbb{N}$ balls in an urn, with $M \leq N$ marked "success". Suppose we perform a two-stage sampling without replacement:*

1. *Draw $K \leq N$ balls from the population without replacement and keep them aside. Amongst those $K$ balls, let $B \leq K$ be the number of balls marked "success". I.e, $B \sim \mathrm{Hyp}(N, M, K)$*

2. *From those $K$ balls, draw another $L < K$ balls, and let $A \leq L$ be the number of successes observed. I.e, $A \sim \mathrm{Hyp}(K, B, L)$*

*Then the marginal distribution of $A$ is:*

$$A | L \sim \mathrm{Hyp}(N, M, L)$$

*Proof.* We prove this by computing the probability of observing a given $L$-subset $A$ after the two-step procedure, and show that it equals the pmf of a Hypergeometric draw.

To do so, we first show that all $L$-subsets are equally likely. The problem then simply amounts to calculating the proportion of $L$-subsets with $A$ successes and $L - A$ failures.

Let $\mathcal{K}$ be the subset of $\mathcal{N}$ of size $K$ selected after the first draw, and $\mathcal{L} \subset \mathcal{K}$ be the set of size $L$ selected after the second draw.

Let $S_{\mathcal{L}}$ denote all possible sets of size $K$ that could have formed $\mathcal{L}$. Formally, $S_{\mathcal{L}} = \{U : \mathcal{L} \subset U \subset \mathcal{N}, |U| = K\}$. Note that $S_{\mathcal{L}}$ cannot be empty for any given $\mathcal{L}$. Moreover, under sampling without replacement, each set in $S_{\mathcal{L}}$ is equally likely. Therefore, the probability of selecting any $L$ sized subset $\mathcal{L}$ is:

$$\mathbb{P}(\mathcal{L}) = \sum_{U \in S_{\mathcal{L}}} \frac{1}{\binom{N}{K}} \cdot \frac{1}{\binom{K}{L}}$$

Note that the number of such sets in $S_{\mathcal{L}}$ is equal to the number of ways to select the remaining $K - L$ items (since $L$ is already determined by selecting $\mathcal{L}$) from the remaining $N - L$ balls. Therefore:

$$\mathbb{P}(\mathcal{L}) = \frac{\binom{N-L}{K-L}}{\binom{N}{K}\binom{K}{L}} = \frac{1}{\binom{N}{L}}$$

The right-hand side does not depend on $\mathcal{L}$, so conditional on $L$ every $L$–subset of $\{1, \ldots, N\}$ is equally likely.

There are $\binom{N}{L}$ such subsets, among which exactly

$$\binom{M}{A}\binom{N-M}{L-A}$$

subsets contain $A$ successes and $L - A$ failures. Hence

$$\mathbb{P}(A = a \mid L = \ell) = \frac{\binom{M}{a}\binom{N-M}{\ell-a}}{\binom{N}{\ell}},$$

which is the pmf of the hypergeometric distribution $\mathrm{Hyp}(N, M, L)$. $\qquad\square$

We can now state composition of the birth–death bridge for fixed slack using the above two lemmas. We show transitioning through the fixed-slack bridge kernel to timepoint $s$ from the starting timepoint 1 yields the same state $X_s$ in distribution regardless of the choice of the number of steps taken to reach $s$.

**Theorem A.5** (Bridge composition conditional on slack). *Fix $0 < s < t < 1$ and endpoints $X_0 = x_0$, $X_1 = x_1$, and slack $M_1 = m$ (equivalently, fix $(N_1, B_1)$ via equation 7). Let $w : [0,1] \to [0,1]$ be the time–rescaling function from the Poisson birth–death reference.*

*Define the fixed–slack bridge kernels*

$$K_{a|0,b}^{(m)}(x_a \mid x_0, x_b) = \mathbb{P}\big(X_a = x_a \mid X_0 = x_0, X_b = x_b, M_1 = m\big), \qquad 0 \le a \le b \le 1.$$

*Then the two–stage transition $(1 \to t \to s)$ yields the same law for $(N_s, B_s)$ as the single–stage transition $(1 \to s)$, and the fixed–slack kernels satisfy the bridge–consistency identity*

$$K_{s|0,1}^{(m)}(x_s \mid x_0, x_1) = \sum_{x_t} K_{s|0,t}^{(m)}(x_s \mid x_0, x_t)\, K_{t|0,1}^{(m)}(x_t \mid x_0, x_1). \qquad (10)$$

*Proof.* We show that the two-step transition $(1 \to t \to s)$ is equivalent to the one-step transition $(1 \to s)$ for $(N_s, B_s)$, or equivalently, $X_s(X_s = x_0 + 2B_s - N_s)$.
Conditional on $(N_1, B_1)$ (equivalently on $(x_0, x_1, M_1 = m)$), we have $N_1$ jumps with $B_1$ "births" and $N_1 - B_1$ "deaths".
The Binomial structure of Lemma A.2 yields

$$N_t \mid N_1 \sim \mathrm{Bin}\big(N_1, w(t)\big), \qquad N_s \mid N_t \sim \mathrm{Bin}\big(N_t, w(s)/w(t)\big).$$

Applying Lemma A.3 with $\theta = w(t)$ and $\eta = w(s)/w(t)$ yields

$$N_s \mid N_1 \sim \mathrm{Bin}\big(N_1, w(s)\big),$$

which is equivalent to the direct $(1 \to s)$ transition.

Now conditioning on the number of jumps at $t$, $N_t$, the Hypergeometric structure described in Lemma A.2 yields

$$B_t | N_t \sim \mathrm{Hyp}(N_1, B_1, N_t) \qquad B_s | (N_t, B_t, N_s) \sim \mathrm{Hyp}(N_t, B_t, N_s)$$

Further, drawing $N_s$ using the binomial draw and applying Lemma A.4 yields:

$$B_s | (N_1, B_1, N_s) \sim Hyp(N_1, B_1, N_s)$$

which is equivalent to the direct $(1 \to s)$ transition.

Thus, at fixed $(x_0, x_1, M_1 = m)$, the two–step transition $(1 \to t \to s)$ and the one–step transition $(1 \to s)$ agree in law for $(N_s, B_s)$, and hence for $X_s$. Equivalently, the conditional distributions $K_{s|0,1}^{(m)}(x_s \mid x_0, x_1) = \mathbb{P}(X_s = x_s \mid X_0 = x_0, X_1 = x_1, M_1 = m)$ obey the kernel identity equation 10, which is exactly the bridge consistency with fixed slack. $\qquad\square$

A.1.3  SLACK POSTERIOR

In the previous section, we established that, *conditional on a fixed slack value*, the bridge satisfies exact composition.

Note that $N_t, B_t, D_t, M_t$ are not observed in the sampling process. To obtain an observable Markov bridge, we *resample the slack at each time* from its posterior distribution, conditional only on the *observed $d_t$*:

$$\pi_t(m \mid d_t) = \mathbb{P}(M_t = m \mid d_t = x_t - x_0),$$

Once we sample $M_t$, we can compute $(N_t, B_t)$ and apply the conditional BH transition on $(N_t, B_t)$ to obtain $(N_s, B_s)$, as described in Lemma A.2.

This subsection derives the explicit form of $\pi_t(m \mid d_t)$ and, crucially, proves its closure under the time-rescaling $w(t)$ of the birth-death process.

**Proposition A.6** (Bessel slack posterior). *Under the Poisson birth-death reference described in section A.1.1, fix $t \in (0, 1]$ and let $d_t = X_t - X_0 = B_t - D_t$. Conditional on $d_t$, the pair $(B_t, D_t)$ lies on the lattice*

$$(B_t, D_t) = \begin{cases} (m + d_t, m), & d_t \geq 0, \\ (m, m + |d_t|), & d_t < 0, \end{cases} \qquad m \in \mathbb{N},$$

*and the induced posterior pmf of the slack $M_t = \min(B_t, D_t)$ is*

$$\pi_t(m \mid d_t) = \mathbb{P}(M_t = m \mid d_t) \; \propto \; \frac{(\Lambda_+(t)\Lambda_-(t))^m}{(m + |d_t|)! \, m!}, \qquad m = 0, 1, 2, \dots \tag{11}$$

*with normalizer*

$$Z_t(d_t) = \left(\Lambda_+(t)\Lambda_-(t)\right)^{-|d_t|/2} I_{|d_t|}\left(2\sqrt{\Lambda_+(t)\Lambda_-(t)}\right),$$

*where $I_\nu$ is the modified Bessel function of the first kind.*

*Proof.* We treat $d_t \geq 0$, the other case being symmetric. If $d_t = B_t - D_t \geq 0$ then $B_t = D_t + d_t$, $M_t = \min(B_t, D_t) = D_t$ and thus

$$(B_t, D_t) = (m + d_t, m) \quad \Longleftrightarrow \quad M_t = m.$$

By independence of the Poisson processes, the joint pmf at $(m + d_t, m)$ is

$$\mathbb{P}(B_t = m + d_t, \, D_t = m) = e^{-(\Lambda_+(t) + \Lambda_-(t))} \frac{\Lambda_+(t)^{m+d_t}}{(m + d_t)!} \frac{\Lambda_-(t)^m}{m!}.$$

Applying Bayes' rule and conditioning on $d_t$:

$$\pi_t(m \mid d_t) = \frac{\mathbb{P}(B_t = m + d_t, \, D_t = m)}{\mathbb{P}(d_t)} = \frac{\Lambda_+(t)^{m+d_t} \Lambda_-(t)^m / ((m + d_t)! \, m!)}{\sum_{r=0}^{\infty} \Lambda_+(t)^{r+d_t} \Lambda_-(t)^r / ((r + d_t)! \, r!)}.$$

Factor $\Lambda_+(t)^{d_t}$ from numerator and denominator to obtain

$$\pi_t(m \mid d_t) = \frac{(\Lambda_+(t)\Lambda_-(t))^m}{(m + d_t)! \, m!} \; \bigg/ \; \sum_{r=0}^{\infty} \frac{(\Lambda_+(t)\Lambda_-(t))^r}{(r + d_t)! \, r!},$$

which is exactly equation 11. The denominator is the standard Skellam normalizer, given by the stated Bessel expression. $\square$

The slack variable $M_t$ represents the "hidden" jumps that cancel out (e.g., a birth followed by a death), and the Bessel posterior tells us their exact distribution. Notably, such "hidden" jumps are increasingly unlikely with increasing displacement $d_t$ due to the factorial growth in the denominator, as observed in Figure 1

We now show that the slack posterior is closed under time-rescaling. In the next section, we define the observable bridge as a "mixture" over the slack posterior, and show that it satisfies the bridge consistency equation 1 using the closure property.

**Corollary A.7** (Slack posterior is closed under time-rescaling). *For any $0 < s < t$,*

$$\pi_s(\,\cdot\mid d_s)\ \text{has the same functional form as}\ \pi_t(\,\cdot\mid d_t),$$

*with parameters obtained by replacing $(\Lambda_+(t), \Lambda_-(t))$ by $(\Lambda_+(s), \Lambda_-(s))$. Since $\Lambda_\pm(s) = \lambda_\pm w(s)$, this closure depends only on the rescaling of $w(\cdot)$ and not on any other properties of the process.*

*Proof.* For closure, note that for any $s < t$ we have independent thinning

$$B_s \sim \mathrm{Poi}(\Lambda_+(s)), \qquad D_s \sim \mathrm{Poi}(\Lambda_-(s)), \qquad \Lambda_\pm(s) = \lambda_\pm w(s).$$

Repeating the same conditioning argument with $(B_s, D_s)$ and $d_s$ yields the same functional family as equation 11, with parameters $(\Lambda_+(s), \Lambda_-(s))$ replacing $(\Lambda_+(t), \Lambda_-(t))$. □

### A.1.4   Composition over Slack

We now define the observable bridge as a mixture of the fixed-slack kernels over the slack posterior defined in the previous section.

**Definition A.8** (Observable Bridge). *Let $K_{a|0,b}^{(m)}$ be the fixed-slack bridge kernel,*

$$K_{a|0,b}^{(m)}(x_a \mid x_0, x_b) \;=\; \mathbb{P}\big(X_a = x_a \mid X_0 = x_0, X_b = x_b, M_1 = m\big), \qquad 0 \le a \le b \le 1.$$

*and let $\pi_t(\,\cdot\mid d_t)$ be the Bessel slack posterior described in A.1.3. Then define the observable bridge in (0,1) at terminal time 1 as*

$$K_{t|0,1}(x_t \mid x_0, x_1) \;=\; \sum_{m=0}^{\infty} \pi_1(m \mid d_1)\, K_{t|0,1}^{(m)}(x_t \mid x_0, x_1), \qquad d_1 = x_1 - x_0.$$

*At an intermediate time $0 < t < 1$, define the observable bridge in (0,t) as:*

$$K_{s|0,t}(x_s \mid x_0, x_t) \;=\; \sum_{m=0}^{\infty} \pi_t(m \mid d_t)\, K_{s|0,t}^{(m)}(x_s \mid x_0, x_t), \qquad d_t = x_t - x_0,$$

In other words, the observable bridge is the expectation of the fixed-slack bridges over the slack posterior. Using the fixed-slack bridge consistency shown in A.5 and properties of the slack posterior described in A.1.3:, we prove the general consistency of the observable bridge.

**Theorem A.9** (Consistency of the observable count bridge). *Under the Poisson birth–death reference described in section A.1.1, the observable bridge, defined as in A.8, satisfies the discrete form of equation 1: for all $0 < s < t < 1$ and endpoints $x_0, x_1$,*

$$K_{s|0,1}(x_s \mid x_0, x_1) = \sum_{x_t} K_{s|0,t}(x_s \mid x_0, x_t)\, K_{t|0,1}(x_t \mid x_0, x_1). \tag{12}$$

*Proof.* Fix $x_0, x_1, d_1 = x_1 - x_0$, and an arbitrary bounded test function $\varphi$ on the state space. We prove equation 12 in weak form by computing $\mathbb{E}[\varphi(X_s) \mid X_0 = x_0, X_1 = x_1]$ in two ways:

First, we write the expectation as a mixture over the slack posterior:

$$\mathbb{E}[\varphi(X_s) \mid X_0 = x_0, X_1 = x_1] = \sum_{m=0}^{\infty} \pi_1(m \mid d_1)\, \mathbb{E}[\varphi(X_s) \mid X_0 = x_0, X_1 = x_1, M = m], \tag{13}$$

Note that each fixed-slack bridge satisfies bridge-consistency as described in Theorem A.5:

$$K_{s|0,1}^{(m)}(x_s \mid x_0, x_1) = \sum_{x_t} K_{s|0,t}^{(m)}(x_s \mid x_0, x_t)\, K_{t|0,1}^{(m)}(x_t \mid x_0, x_1).$$

where $K_{s|0,1}^{(m)}(x_s \mid x_0, x_1) = \mathbb{P}(X_s = x_s \mid X_0 = x_0, X_1 = x_1, M = m)$. By taking expectation over the test function $\varphi$, we get:

$$\mathbb{E}[\varphi(X_s) \mid X_0 = x_0, X_1 = x_1, M = m]$$
$$= \sum_{x_t} \mathbb{E}[\varphi(X_s) \mid X_0 = x_0, X_t = x_t, M = m]\, K_{t|0,1}^{(m)}(x_t \mid x_0, x_1).$$

Plug the above into equation 13 and interchange the sums over $x_t$ and $m$:

$$\mathbb{E}[\varphi(X_s) \mid X_0 = x_0, X_1 = x_1]$$
$$= \sum_{x_t} \sum_{m=0}^{\infty} \pi_1(m \mid d_1) K_{t|0,1}^{(m)}(x_t \mid x_0, x_1) \mathbb{E}[\varphi(X_s) \mid X_0 = x_0, X_t = x_t, M = m] \tag{14}$$

Now consider the first two terms inside the sum over $m$. Note that they form of the joint likelihood of $(M, X_t)$ given $(X_0, X_1)$:

$$\pi_1(m \mid d_1) K_{t|0,1}^{(m)}(x_t \mid x_0, x_1) = \mathbb{P}(M = m, X_t = x_t \mid X_0 = x_0, X_1 = x_1)$$

Using the definition of conditional probability, we can "flip" this joint density to condition on the intermediate state $X_t$:

$$\mathbb{P}(M, X_t \mid X_0, X_1) = \mathbb{P}(M \mid X_t, X_0, X_1), \mathbb{P}(X_t \mid X_0, X_1). \tag{15}$$

Once $X_t$ is fixed, the future terminal state $X_1$ provides no additional information about the slack jumps that occurred before time $t$, thereby isolating $M_t$ on $(0, t)$. As a result $\mathbb{P}(M|X_t, X_0, X_1)$ is simply the posterior distribution of $M_t$, following the Bessel form described in Proposition A.6 as a result of the time-rescaling Corrolary A.7. To summarize:

$$\mathbb{P}(M = m \mid X_t = x_t, X_0 = x_0, X_1 = x_1) = \mathbb{P}(M = m \mid X_t = x_t, X_0 = x_0) = \pi_t(m \mid d_t)$$

Substituting this back into equation 15 gives the identity:

$$\pi_1(m \mid d_1), K_{t|0,1}^{(m)}(x_t \mid x_0, x_1) = \pi_t(m \mid d_t), K_{t|0,1}(x_t \mid x_0, x_1). \tag{16}$$

Plugging the above into equation 14:

$$\mathbb{E}[\varphi(X_s) \mid X_0 = x_0, X_1 = x_1]$$
$$= \sum_{x_t} \sum_{m=0}^{\infty} \pi_t(m \mid d_t) K_{t|0,1}(x_t \mid x_0, x_1) \mathbb{E}[\varphi(X_s) \mid X_0 = x_0, X_t = x_t, M = m] \tag{17}$$

By the law of total expectation:

$$\sum_{m=0}^{\infty} \pi_t(m \mid d_t) \mathbb{E}[\varphi(X_s) \mid X_0 = x_0, X_t = x_t, M = m] = \mathbb{E}[\varphi(X_s) \mid X_0 = x_0, X_t = x_t],$$

Plugging this into the above equation:

$$\mathbb{E}[\varphi(X_s) \mid X_0 = x_0, X_1 = x_1] = \sum_{x_t} K_{t|0,1}(x_t \mid x_0, x_1) \mathbb{E}[\varphi(X_s) \mid X_0 = x_0, X_t = x_t]$$

and the definition of $K_{s|0,t}$ then gives

$$\mathbb{E}[\varphi(X_s) \mid X_0 = x_0, X_t = x_t] = \sum_{x_s} \varphi(x_s) K_{s|0,t}(x_s \mid x_0, x_t).$$

Putting everything together,

$$\mathbb{E}[\varphi(X_s) \mid X_0 = x_0, X_1 = x_1] = \sum_{x_t} \left( \sum_{x_s} \varphi(x_s) K_{s|0,t}(x_s \mid x_0, x_t) \right) K_{t|0,1}(x_t \mid x_0, x_1)$$
$$= \sum_{x_s} \varphi(x_s) \left( \sum_{x_t} K_{s|0,t}(x_s \mid x_0, x_t) K_{t|0,1}(x_t \mid x_0, x_1) \right).$$

On the other hand, by the definition of $K_{s|0,1}$,

$$\mathbb{E}[\varphi(X_s) \mid X_0 = x_0, X_1 = x_1] = \sum_{x_s} \varphi(x_s) K_{s|0,1}(x_s \mid x_0, x_1).$$

Since $\varphi$ is arbitrary, the coefficients of $\varphi(x_s)$ must agree for all $x_s$, that is:

$$K_{s|0,1}(x_s \mid x_0, x_1) = \sum_{x_t} K_{s|0,t}(x_s \mid x_0, x_t) \, K_{t|0,1}(x_t \mid x_0, x_1)$$

which yields equation 12. $\qquad\qquad\qquad\qquad\qquad\qquad\qquad\qquad\qquad\qquad\qquad\square$

We have now shown that the observable birth-death bridge satisfies the same bridge consistency property equation 1 as the Gaussian diffusion bridge, confirming the correctness of the bridge as a diffusion kernel.

## A.2 Link with Schrödinger Bridges

We now justify the Schrödinger-bridge interpretation stated in Sec. 3. Recall the unconditional birth–death construction

$$X_t \;=\; X_0 + B_t - D_t,$$

where $(B_t)_{t \in [0,1]}$ and $(D_t)_{t \in [0,1]}$ are independent Poisson processes with cumulative intensities $\Lambda_+(t) = \lambda_+ w(t)$, $\Lambda_-(t) = \lambda_- w(t)$, and $w(0) = 0, w(1) = 1$. Let $\kappa = \sqrt{\lambda_+ \lambda_-}$ denote the (scalar) jump intensity. Then for each $x_0 \in \mathcal{X}$, the forward kernel at time $t = 1$ has Skellam pmf

$$K_{1|0}^{\kappa}(x_1 \mid x_0) = \exp\big[ -\Lambda_+(1) - \Lambda_-(1) \big] \left( \frac{\Lambda_+(1)}{\Lambda_-(1)} \right)^{\frac{x_1 - x_0}{2}} I_{|x_1 - x_0|}(2\kappa),$$

where $I_\nu$ is the modified Bessel function of the first kind.

Let

$$p_{\text{ref}}^{\kappa}(x_0, x_1) \;=\; p_0(x_0) \, K_{1|0}^{\kappa}(x_1 \mid x_0)$$

be the *reference joint law* of $(X_0, X_1)$ induced by the birth–death process and the data prior $p_0$. Over the space of couplings

$$\mathcal{C}(p_0, p_1) = \big\{ C \text{ on } \mathcal{X} \times \mathcal{X} : C(\cdot, \mathcal{X}) = p_0, \; C(\mathcal{X}, \cdot) = p_1 \big\},$$

we consider the entropic projection

$$C_\kappa \in \arg \min_{C \in \mathcal{C}(p_0, p_1)} \mathrm{KL}\big( C \,\|\, p_{\text{ref}}^{\kappa} \big).$$

Fix any $C \in \mathcal{C}(p_0, p_1)$ and write expectations with respect to $C$ as $\mathbb{E}_C[\cdot]$. By definition,

$$\begin{aligned}
\mathrm{KL}\big( C \,\|\, p_{\text{ref}}^{\kappa} \big) &= \mathbb{E}_C\bigg[ \log \frac{dC}{dp_{\text{ref}}^{\kappa}}(X_0, X_1) \bigg] \\
&= -H(C) - \mathbb{E}_C\big[ \log K_{1|0}^{\kappa}(X_1 \mid X_0) \big] + C_\kappa(p_0, p_1),
\end{aligned}$$

where $H(C)$ is the Shannon entropy of $C$ and $C_\kappa(p_0, p_1)$ collects all terms depending only on the marginals (including $\Lambda_\pm(1)$ and $\mathbb{E}_C[X_1 - X_0]$, which are fixed across all couplings with marginals $p_0, p_1$).

Using the explicit Skellam form, we can isolate the Bessel contribution:

$$\mathrm{KL}\big( C \,\|\, p_{\text{ref}}^{\kappa} \big) = C_\kappa(p_0, p_1) - \mathbb{E}_C\big[ \log I_{|X_1 - X_0|}(2\kappa) \big] - H(C).$$

To study the limits $\kappa \to \infty$ and $\kappa \to 0^+$, we use standard asymptotics for $I_\nu$ (NIS, 2025, §10.41(ii)):

$$I_\nu(z) = \frac{e^z}{\sqrt{2\pi z}} \, (1 + o(1)) \quad \text{as } z \to \infty, \qquad I_\nu(z) = \frac{(z/2)^\nu}{\Gamma(\nu+1)} \, (1 + o(1)) \quad \text{as } z \to 0^+.$$

**High-noise limit $\kappa \to \infty$.** For fixed $\nu$, the large-$z$ expansion shows that $\log I_\nu(2\kappa)$ depends on $\nu$ only through $O(1/\kappa)$ terms. Hence, as $\kappa \to \infty$,

$$\mathbb{E}_C\big[ \log I_{|X_1 - X_0|}(2\kappa) \big] = C'_\kappa(p_0, p_1) + o_\kappa(1),$$

where $C'_\kappa(p_0, p_1)$ does not depend on the choice of coupling $C$. It follows that

$$\mathrm{KL}\big( C \,\|\, p_{\text{ref}}^{\kappa} \big) = C - H(C) + o_\kappa(1), \qquad \kappa \to \infty,$$

for some constant $C$ that is independent of $C$. Maximizing $H(C)$ over $\mathcal{C}(p_0, p_1)$ yields the independent coupling $p_0 \otimes p_1$, so in the high-noise limit $C_\kappa \to p_0 \otimes p_1$.

**Low-noise limit** $\kappa \to 0^+$**.** For small $z$, we have

$$\log I_\nu(2\kappa) = \nu \log\left(\tfrac{2}{\kappa}\right) + O(1) \quad \text{as } \kappa \to 0^+,$$

uniformly for integer $\nu$ in any fixed finite range. Applying this with $\nu = |X_1 - X_0|$ gives

$$-\mathbb{E}_C\big[\log I_{|X_1 - X_0|}(2\kappa)\big] = \log\left(\tfrac{2}{\kappa}\right)\mathbb{E}_C|X_1 - X_0| + O(1), \quad \kappa \to 0^+.$$

Substituting into the expression for the KL divergence, we obtain

$$\mathrm{KL}\big(C \,\|\, p_{\mathrm{ref}}^\kappa\big) = C + \log\left(\tfrac{2}{\kappa}\right)\mathbb{E}_C|X_1 - X_0| - H(C) + o_\kappa(1), \qquad \kappa \to 0^+,$$

for some constant $C$ independent of $C$.

Thus, in the low-noise limit the dominant term in $\mathrm{KL}(C \,\|\, p_{\mathrm{ref}}^\kappa)$ is proportional to $\mathbb{E}_C|X_1 - X_0|$; minimizing the KL over $\mathcal{C}(p_0, p_1)$ therefore recovers discrete optimal transport with cost $|x_1 - x_0|$, while the entropy term $-H(C)$ corresponds to the usual entropic regularization. This proves the characterization stated in Sec. 3.

## B  LIFTING COUNT BRIDGES TO AGGREGATES

We make two central assumptions that enable deconvolution.

**Assumption B.1** (Realizability and recoverability). *There exists $\theta^\star$ such that for all $t \in [0, 1]$:*

1. ***Realizability:*** $Q_{\theta^\star}(a_0 \mid \mathbf{x}_t, t, z) = p_0(a_0 \mid \mathbf{x}_t, t, z)$ *almost surely*

2. ***Recoverability:*** *The aggregate-to-unit map has local modulus $\kappa_{loc}(t)$: for $\theta$ near $\theta^\star$,*

$$D_{KL}(p_0(\mathbf{x}_0 \mid \mathbf{x}_t, t, z) \,\|\, q_\theta(\cdot \mid \mathbf{x}_t, t, z)) \le \kappa_{loc}(t) \cdot D_{KL}(p_0(a_0 \mid \mathbf{x}_t, t, z) \,\|\, Q_\theta(\cdot \mid \mathbf{x}_t, t, z))$$

Recoverability means that the aggregate distribution uniquely determines the unit-level distribution—if we know the sum perfectly, we can deduce the summands. This is not always possible. Consider the simplest case: if $X_1, X_2 \sim \mathrm{Poisson}(\lambda_1), \mathrm{Poisson}(\lambda_2)$ are independent, their sum is $\mathrm{Poisson}(\lambda_1 + \lambda_2)$. Observing only the sum, we cannot distinguish $(\lambda_1 = 3, \lambda_2 = 2)$ from $(\lambda_1 = 4, \lambda_2 = 1)$—both yield $\mathrm{Poisson}(5)$. Now if we had side information that identified the "component" each Poisson was drawn from we could identify this, but it illustrates the difficulties we will face here.

Recoverability holds when units have sufficient diversity. Formally, it requires that units are conditionally independent given $(\mathbf{x}_1, z)$ and have distinct factorial cumulant signatures—essentially, different statistical fingerprints that survive aggregation. For count data, this means units must have different parameters (e.g., different Poisson rates or negative binomial dispersions) that are distinguishable through the covariates $z$.

Appendix B.2 provides a detailed discussion. In practice, this means units should have heterogeneous characteristics captured by $z$—for example, different images associated with the transciptomics is spatial single cell data.

Recoverability faces fundamental limits as the number of units $G$ grows. By the central limit theorem, when $G \to \infty$, the standardized aggregate $(A_0 - \mu_G)/\sigma_G$ converges to a Gaussian regardless of the unit-level distributions. The higher-order cumulants that distinguish different unit configurations vanish at rate $O(G^{-(k-2)/2})$ for order $k > 2$, leaving only the mean and variance (Appendix B.3).

This CLT collapse means our method is most powerful for moderate $G$ (tens to hundreds of units) where unit heterogeneity is preserved in aggregates. For very large $G$, additional structure is needed—either parametric constraints (e.g., unit parameters follow a low-dimensional model), multiple aggregate observations under different conditions, or direct observation of some unit-level data.

We illustrate these issues in Fig. 4 in the main text: as the dirichlet concentration increases the group-level mixture weights concentrate. When combined with large group sizes this forces all groups to become identical. This gives a clear empirical sense for the limits of deconvolution.

### B.1 APPROXIMATELY SAMPLING CONDITIONAL ON THE SUM

The most central part of our deconvolution approach is our projection operation. This appendix formalizes the rescaling operator used in Prop. 4.1 and explains precisely in what sense it approximates the aggregate–conditional distribution $Q_\theta(\,\cdot\mid A(\mathbf{X}_0) = a_0)$.

#### B.1.1 SETUP AND TWO ENSEMBLES

Let $\mathbf{X}_0 \in \mathbb{Z}_{\geq 0}^{G \times D}$ be a random count array with (unconstrained) law $P_\theta$, and define the aggregate statistic

$$A_0 \;:=\; A(\mathbf{X}_0) \in \mathbb{Z}_{\geq 0}^D.$$

In the main text, we are interested in the *microcanonical* (hard-constraint) conditional law

$$Q_\theta(\,\cdot\mid A_0 = a_0) \;:=\; P_\theta(\,\cdot\mid A(\mathbf{X}_0) = a_0), \qquad \text{for feasible } a_0 \text{ with } P_\theta(A_0 = a_0) > 0,$$

which is supported on the affine constraint set $\{\mathbf{x}_0 : A(\mathbf{x}_0) = a_0\}$.

A key point is that $Q_\theta(\,\cdot\mid A_0 = a_0)$ is not, in general, an exponential tilt of $P_\theta$. Exponential tilts instead arise naturally from the canonical (moment-constrained) relaxation, which replaces the hard constraint $A_0 = a_0$ by $\mathbb{E}[A_0] = a_0$.

Throughout this subsection, we work with count arrays $\mathbf{X}_0 \in \mathbb{Z}_{\geq 0}^{G \times D}$, so the aggregate $A_0 = A(\mathbf{X}_0) \in \mathbb{Z}_{\geq 0}^D$ is lattice-valued. Hence for any *feasible* $a_0$ with $P_\theta(A_0 = a_0) > 0$, the microcanonical conditional $P_\theta(\,\cdot\mid A_0 = a_0)$ is an ordinary conditional distribution (no measure-zero issue). When defining the deterministic rescaling projection $\Pi$ below, we relax the projection domain to $\mathbb{R}_{\geq 0}^{G \times D}$ to obtain a closed form.

#### B.1.2 CANONICAL EXPONENTIAL TILT AND FIRST-ORDER EXPANSION

**Assumption B.2** (Cramér regularity for the aggregate). *The log-mgf*

$$\Lambda(\lambda) \;:=\; \log \mathbb{E}_{P_\theta}\Big[ \exp\{\lambda^\top A_0\} \Big]$$

*is finite on an open neighborhood of $\lambda = 0$, twice continuously differentiable there, and has positive-definite covariance*

$$\Sigma \;:=\; \nabla^2 \Lambda(0) \;=\; \mathrm{Cov}_{P_\theta}(A_0) \;\succ\; 0.$$

*Moreover, $\nabla^2\Lambda$ is locally Lipschitz in a neighborhood of $0$.*

**Definition B.3** (Canonical moment-constrained tilt). *For $\lambda$ in the regularity neighborhood, define the tilted law $P_{\theta,\lambda}$ on $\mathbf{X}_0$ by*

$$\frac{dP_{\theta,\lambda}}{dP_\theta}(\mathbf{x}_0) \;=\; \exp\{\lambda^\top A(\mathbf{x}_0) - \Lambda(\lambda)\}.$$

*If $a_0$ is sufficiently close to $\mu := \mathbb{E}_{P_\theta}[A_0] = \nabla\Lambda(0)$, the canonical parameter $\hat\lambda$ is the unique solution of*

$$\nabla\Lambda(\hat\lambda) \;=\; a_0,$$

*and $P_{\theta,\hat\lambda}$ is the unique KL-minimizer of $D_{\mathrm{KL}}(\,\cdot\,\|P_\theta)$ over distributions with $\mathbb{E}[A_0] = a_0$.*

**Lemma B.4** (First-order expansion of the canonical parameter). *Under Assumption B.2, let $\delta := a_0 - \mu$. Then for $\|\delta\|$ sufficiently small,*

$$\hat\lambda \;=\; \Sigma^{-1}\delta \;+\; O(\|\delta\|^2).$$

*Proof.* Taylor expand $\nabla\Lambda(\lambda)$ at $0$: $\nabla\Lambda(\lambda) = \mu + \Sigma\lambda + R(\lambda)$ with $\|R(\lambda)\| \leq \frac{L}{2}\|\lambda\|^2$. Solving $\mu + \Sigma\hat\lambda + R(\hat\lambda) = \mu + \delta$ yields $\hat\lambda = \Sigma^{-1}(\delta - R(\hat\lambda))$ and the stated bound. $\square$

#### B.1.3 HOW THE MICROCANONICAL CONDITIONAL RELATES TO THE CANONICAL TILT

We record an identity and a standard approximation principle.

**Lemma B.5** (Tilting does not change the hard conditional). *For any $\lambda$ in the regularity neighborhood and any feasible $a_0$,*

$$P_\theta(\,\cdot\,\mid A_0 = a_0) \;=\; P_{\theta,\lambda}(\,\cdot\,\mid A_0 = a_0).$$

*Proof.* On the event $\{A_0 = a_0\}$, the Radon–Nikodym factor $\exp\{\lambda^\top A_0 - \Lambda(\lambda)\}$ is constant, hence cancels in the conditional density ratio. $\qquad\square$

Lemma B.5 lets us choose the most convenient tilt for analysis of the conditional. The canonical choice $\lambda = \hat\lambda$ is particularly useful because under $P_{\theta,\hat\lambda}$, the aggregate is centered at the constraint: $\mathbb{E}_{P_{\theta,\hat\lambda}}[A_0] = a_0$.

To connect the hard conditional to the tilt, one needs an additional large-system regime (e.g. many summed contributions, or large aggregate counts) ensuring that conditioning on $A_0 = a_0$ only induces a small correction beyond matching the mean. A classical form is the Gibbs conditioning principle.

Consider a sequence of problems indexed by $n$ with independent random arrays $\mathbf{X}_0^{(n)} = (X_{10}^{(n)}, \ldots, X_{n0}^{(n)})$ under $P_\theta^{(n)}$ and aggregate $A_0^{(n)} = \sum_{g=1}^n X_{g0}^{(n)}$ (elementwise). Assume a uniform Cramér condition and a local limit theorem for $A_0^{(n)}$ under the canonical tilt $P_{\theta,\hat\lambda_n}^{(n)}$ (where $\hat\lambda_n$ solves $\nabla\Lambda_n(\hat\lambda_n) = a_0^{(n)}$ and $\Lambda_n(\lambda) = \log\mathbb{E}[\exp\{\lambda^\top A_0^{(n)}\}]$). Then, for any fixed $m$ and any bounded measurable $f$ depending only on $(X_{10}^{(n)}, \ldots, X_{m0}^{(n)})$,

$$\mathbb{E}_{Q_\theta^{(n)}(\cdot\,|A_0^{(n)}=a_0^{(n)})}[f] \;-\; \mathbb{E}_{P_{\theta,\hat\lambda_n}^{(n)}}[f] \;\longrightarrow\; 0 \qquad (n \to \infty),$$

where $Q_\theta^{(n)}(\,\cdot\,\mid A_0^{(n)} = a_0^{(n)})$ denotes the microcanonical conditional.

Essentially, for local observables in a large-sum regime, the conditional behaves like the canonical tilt with parameter $\hat\lambda$, and Lemma B.4 gives the first-order expansion $\hat\lambda = \Sigma^{-1}\delta + O(\|\delta\|^2)$. This is the precise meaning of the phrase "admits a first-order exponential tilt" in Prop. 4.1. This first-order argument requires a large-$n$ justification, which conflicts with our identifiability perspective below, but we provide it as a useful perspective on the projection operation here since it may serve as a blueprint for thinking about reasonable projections in other settings.

### B.1.4 THE RESCALING MAP AS AN ENTROPIC PROJECTION

The operator $\Pi$ in Prop. 4.1 is a projection of a *point* onto the hard constraint set. Since points are not distributions, we use the standard *generalized KL* (a Bregman divergence on $\mathbb{R}_{\geq 0}^{G \times D}$):

$$D_{\mathrm{KL}}(\mathbf{y}\|\mathbf{x}) \;:=\; \sum_{g=1}^G \sum_{d=1}^D \left( y_g^{(d)} \log \frac{y_g^{(d)}}{x_g^{(d)}} - y_g^{(d)} + x_g^{(d)} \right),$$

with the conventions $0\log 0 = 0$ and $y\log(y/0) = +\infty$ for $y > 0$.

**Lemma B.6** (Closed form of the projection for elementwise sums). *Let $A(\mathbf{x}_0) = \sum_g x_{g0}$ (elementwise) and assume $\sum_g x_{g0}^{(d)} > 0$ for each $d$ with $a_0^{(d)} > 0$. Then the minimizer*

$$\Pi(\mathbf{x}_0) = \arg\min_{\mathbf{y}_0:\,A(\mathbf{y}_0)=a_0} D_{\mathrm{KL}}(\mathbf{y}_0\|\mathbf{x}_0)$$

*exists, is unique, and is given by the coordinatewise scaling*

$$\Pi(\mathbf{x}_0)_g^{(d)} \;=\; x_{g0}^{(d)} \cdot \frac{a_0^{(d)}}{\sum_{g'} x_{g'0}^{(d)}}, \qquad d = 1, \ldots, D.$$

*Proof.* The objective is strictly convex in $\mathbf{y}_0$ on $\{\mathbf{y}_0 \geq 0\}$, and the constraints are affine, so the minimizer is unique. Form the Lagrangian with multipliers $\nu^{(d)}$ for the $D$ constraints:

$$\mathcal{L}(\mathbf{y}, \nu) = \sum_{g,d} \left( y_g^{(d)} \log \frac{y_g^{(d)}}{x_g^{(d)}} - y_g^{(d)} + x_g^{(d)} \right) + \sum_d \nu^{(d)} \left( \sum_g y_g^{(d)} - a_0^{(d)} \right).$$

Stationarity gives $\partial \mathcal{L}/\partial y_g^{(d)} = \log(y_g^{(d)}/x_g^{(d)}) + \nu^{(d)} = 0$, hence $y_g^{(d)} = x_g^{(d)} e^{-\nu^{(d)}}$. Imposing $\sum_g y_g^{(d)} = a_0^{(d)}$ yields $e^{-\nu^{(d)}} = a_0^{(d)}/\sum_g x_g^{(d)}$, proving the formula. $\qquad\square$

Since we are then operating on count matrices here we additionally introduce a randomized rounding algorithm which enables us to exactly sample from a count-valued point on this projected set.

### B.2 Conditions for Recoverability

We give conditions under which aggregate supervision can, in principle, identify (unit-level) count distributions from aggregate observations. Throughout we use the notation of Sec. 4: $\mathbf{X}_0 = (X_{10}, \ldots, X_{G0}) \in \mathbb{Z}_{\geq 0}^G$, $A_0 := A(\mathbf{X}_0)$, and in the sum case $A(\mathbf{x}_0) = \sum_{g=1}^G x_{g0}$. We fix $(\mathbf{x}_t, t, z)$ and suppress this conditioning in the notation when convenient.

#### B.2.1 Factorial cumulant framework

For a nonnegative integer-valued random variable $X$, define its probability generating function (pgf) $G_X(s) = \mathbb{E}[s^X]$ for $s \in [0, 1]$, and write the factorial moment generating function (FMGF) as

$$M_X(t) = \mathbb{E}[(1+t)^X] = G_X(1+t), \qquad t \in (-1, 0] \text{ (and possibly for small } t > 0).$$

Whenever $M_X(t)$ is finite on a neighborhood of $t = 0$, define the factorial cumulant generating function (FCGF)

$$C_X(t) = \log M_X(t) = \sum_{k \geq 1} \frac{\kappa_k(X)}{k!} t^k,$$

where $\kappa_k(X) = C_X^{(k)}(0)$ are the factorial cumulants. The pgf $G_X$ (hence $C_X$) determines the law of $X$ uniquely.

**Lemma B.7** (Additivity under (conditional) independence). *If $X$ and $Y$ are independent (or conditionally independent given a $\sigma$-field $\mathcal{C}$), then*

$$C_{X+Y}(t) = C_X(t) + C_Y(t) \qquad \big(\text{or } C_{X+Y}(t \mid \mathcal{C}) = C_X(t \mid \mathcal{C}) + C_Y(t \mid \mathcal{C})\big),$$

*on any interval where the FCGFs are finite.*

#### B.2.2 Additive identifiability of factorial-cumulant signatures

Recovering unit-level distributions from the law of a sum is *not automatic*: distinct unit configurations can yield the same aggregate law (e.g., sums of independent Poissons depend only on the total rate). To state correct conditions, we separate a *structural identifiability* property from regularity.

**Definition B.8** ($G$-additive identifiability). *Let $\{F(\cdot; \psi) : \psi \in \Psi\}$ be a family of real-analytic functions on a neighborhood of $t = 0$. We say the family is $G$-additively identifiable if for any parameters $\psi_1, \ldots, \psi_G$ and $\psi_1', \ldots, \psi_G'$,*

$$\sum_{g=1}^G F(t; \psi_g) \equiv \sum_{g=1}^G F(t; \psi_g') \quad \text{(as analytic functions near 0)}$$
$$\implies \{\psi_1, \ldots, \psi_G\} \text{ and } \{\psi_1', \ldots, \psi_G'\} \text{ coincide as multisets.}$$

**Lemma B.9** (A simple sufficient condition). *If the family $\{F(\cdot; \psi) : \psi \in \Psi\}$ is (finitely) linearly independent as analytic functions, i.e., any finite linear combination $\sum_{j=1}^m c_j F(\cdot; \psi_j) \equiv 0$ with distinct $\psi_j$ forces $c_j = 0$, then it is $G$-additively identifiable for every $G$.*

*Proof.* If $\sum_{g=1}^G F(\cdot; \psi_g) \equiv \sum_{g=1}^G F(\cdot; \psi_g')$, bring all terms to one side and group identical parameters. This yields a finite linear combination $\sum_{j=1}^m c_j F(\cdot; \tilde\psi_j) \equiv 0$ with integer coefficients $c_j \in \mathbb{Z}$. Linear independence forces all $c_j = 0$, hence matching multiplicities, i.e., the multisets coincide. $\quad\square$

For $X \sim \mathrm{Poi}(\lambda)$ one has $C_X(t) = \lambda t$, so the family is *not* additively identifiable: $\lambda_1 t + \lambda_2 t = (\lambda_1 + \lambda_2)t$ shows only the total rate is recoverable from the sum. Thus recoverability requires families whose FCGFs contain richer (nonlinear) signatures than a single linear term.

### B.2.3 RECOVERABILITY FROM AGGREGATE LAWS

We now give a clean identifiability statement for the *sum* aggregate under conditional independence. This captures the mathematical core of when deconvolution from aggregates is possible.

**Theorem B.10** (Recoverability up to permutation from the sum). *Fix $(\mathbf{x}_t, t, z)$ and suppose that conditional on this information the units $X_{10}, \ldots, X_{G0}$ are independent and*

$$X_{g0} \mid (\mathbf{x}_t, t, z) \sim P_{\psi_g}, \qquad g = 1, \ldots, G,$$

*where each $P_\psi$ is a count distribution whose FCGF equals $F(\cdot; \psi)$ on a neighborhood of 0. Assume $\{F(\cdot; \psi) : \psi \in \Psi\}$ is $G$-additively identifiable (Def. B.8). Then the conditional law of the sum*

$$A_0 = \sum_{g=1}^{G} X_{g0}$$

*uniquely determines the multiset $\{\psi_1, \ldots, \psi_G\}$ (hence the multiset of unit laws $\{P_{\psi_g}\}$), up to permutation of units.*

*Proof.* By Lemma B.7,

$$C_{A_0}(t \mid \mathbf{x}_t, t, z) = \sum_{g=1}^{G} C_{X_{g0}}(t \mid \mathbf{x}_t, t, z) = \sum_{g=1}^{G} F(t; \psi_g).$$

Equality in distribution of $A_0$ implies equality of its pgf (hence FCGF) on an interval, hence equality of the analytic function $C_{A_0}(\cdot)$. By $G$-additive identifiability, the multiset of parameters is unique. □

**How covariates $z$ make learning possible.** Theorem B.10 is the strongest statement one can make from a single sum: because $A_0$ is symmetric in the units, one can in general identify unit distributions only up to permutation. Side information $z$ enables *learnability* once it imposes additional structure that breaks this symmetry across groups. Two sufficient structures are:

1. **Finite labeled types (composition varies across groups).** Suppose each unit has an observed type label $z_g \in \{1, \ldots, K\}$ and $X_{g0} \mid z_g = k \sim P_{\psi_k}$, so all units of a given type share parameters. Then for a group with type counts $(n_1, \ldots, n_K)$,

   $$C_{A_0}(t \mid n_1, \ldots, n_K) = \sum_{k=1}^{K} n_k \, F(t; \psi_k).$$

   Across multiple groups whose compositions $(n_1, \ldots, n_K)$ vary sufficiently (a full-rank design), the shared parameters $(\psi_1, \ldots, \psi_K)$ become identifiable even though within any single group the sum is symmetric.

2. **Low-dimensional parameterization by covariates.** More generally, assume $\psi_g = \psi_\eta(z_g)$ for a *shared* low-dimensional parameter $\eta$ (e.g., a regression model, a neural net, or another identifiable function class). Then the aggregate FCGF $\sum_g F(\cdot; \psi_\eta(z_g))$ varies with $(z_1, \ldots, z_G)$; if the covariates span the function class and the induced map $\eta \mapsto \text{Law}(A_0 \mid z)$ is injective (locally, in our setting), then $\eta$ is learnable from aggregate supervision.

### B.3 LARGE-$G$ LIMITS: GAUSSIAN COLLAPSE OF AGGREGATE INFORMATION

We explain why learning unit-level structure from aggregates becomes statistically ill-conditioned as the number of units $G$ grows.

Fix $(\mathbf{x}_t, t, z)$. Let $\mathbf{X}_0 = (X_{10}, \ldots, X_{G0}) \in \mathbb{Z}_{\geq 0}^G$ be conditionally independent given $(\mathbf{x}_t, t, z)$, with

$$X_{g0} \mid (\mathbf{x}_t, t, z) \sim P_g(\cdot \mid \mathbf{x}_t, t, z), \qquad g = 1, \ldots, G,$$

(e.g., $P_g(\cdot \mid \mathbf{x}_t, t, z) = q_\theta(\cdot \mid \mathbf{x}_t, t, z_g)$ under our model). Let the aggregate random variable be

$$A_0 := A(\mathbf{X}_0) \in \mathbb{Z}, \qquad \text{and in particular for the sum case } A(\mathbf{x}_0) = \sum_{g=1}^{G} x_{g0}.$$

Write

$$\mu_G := \mathbb{E}[A_0 \mid \mathbf{x}_t, t, z], \qquad \sigma_G^2 := \mathrm{Var}(A_0 \mid \mathbf{x}_t, t, z) = \sum_{g=1}^{G} \mathrm{Var}(X_{g0} \mid \mathbf{x}_t, t, z).$$

Assume $\sigma_G^2 \to \infty$ and a Lindeberg (or Lyapunov) condition holds conditional on $(\mathbf{x}_t, t, z)$. Then

$$\frac{A_0 - \mu_G}{\sigma_G} \Rightarrow \mathcal{N}(0, 1) \qquad (G \to \infty).$$

Moreover, if $\sum_{g=1}^{G} \mathbb{E}[|X_{g0} - \mathbb{E}X_{g0}|^3 \mid \mathbf{x}_t, t, z] < \infty$, a Berry–Esseen bound yields

$$\sup_{x \in \mathbb{R}} \left| \Pr\left( \frac{A_0 - \mu_G}{\sigma_G} \le x \,\middle|\, \mathbf{x}_t, t, z \right) - \Phi(x) \right| \le C \frac{\sum_{g=1}^{G} \mathbb{E}[|X_{g0} - \mathbb{E}X_{g0}|^3 \mid \mathbf{x}_t, t, z]}{\sigma_G^3},$$

for a universal constant $C$.

Let $\kappa_k(\cdot)$ denote cumulants (ordinary cumulants, or factorial cumulants for count data). Under conditional independence, cumulants add:

$$\kappa_k(A_0 \mid \mathbf{x}_t, t, z) = \sum_{g=1}^{G} \kappa_k(X_{g0} \mid \mathbf{x}_t, t, z).$$

If $\sup_g |\kappa_k(X_{g0} \mid \mathbf{x}_t, t, z)| < \infty$ and $\sigma_G^2 = \Theta(G)$, then for all $k \ge 3$,

$$\frac{\kappa_k(A_0 \mid \mathbf{x}_t, t, z)}{\sigma_G^k} = O\big(G^{1-k/2}\big) = O\big(G^{-(k-2)/2}\big) \xrightarrow[G \to \infty]{} 0.$$

Thus, after centering and scaling, the aggregate law is asymptotically Gaussian and is effectively determined by $(\mu_G, \sigma_G^2)$; information about unit heterogeneity enters only through standardized cumulants of size $G^{-(k-2)/2}$.

**Implication for deconvolution from aggregates.** Aggregation is a many-to-one map: there are typically many distinct collections of unit-level laws $\{P_g\}_{g=1}^{G}$ that induce the same (or nearly the same) aggregate law of $A_0$. The CLT strengthens this ambiguity for large $G$: once $A_0$ is close to Gaussian, distinguishing different unit-level configurations using aggregate observations alone requires resolving the small $G^{-(k-2)/2}$ corrections beyond mean and variance. Equivalently, any local "recoverability modulus" that upper-bounds unit-level error by aggregate-level error must deteriorate with $G$ unless additional structure is imposed (e.g., multiple independent aggregates, parametric tying/low-dimensional structure across units, or partial unit-level supervision).

## C PROJECTION AND ROUNDING ALGORITHMS

**Group rescaling.** Algorithm 5 rescales item-level nonnegative values $\{x_b\}_{b=1}^{B}$ so that each group $G_g$ attains a prescribed aggregate $C_g$. The procedure is linear-time in the number of items and linear in memory in the number of groups:

$$T = O(B), \qquad M_{\mathrm{extra}} = O(G),$$

and is applied in parallel to each feature dimension.

---

**Algorithm 5** Group Rescaling to Match Aggregates

---

**Require:** item values $x_b \geq 0$ for $b = 1, \ldots, B$; groups $G_1, \ldots, G_G$; targets $C_g \geq 0$
**Ensure:** $y_b \geq 0$ with $\sum_{b \in G_g} y_b = C_g$ for all $g$

  1: **for** $g = 1, \ldots, G$ **do**
  2:      $S_g \leftarrow \sum_{b \in G_g} x_b$
  3:      **if** $S_g > 0$ **then**
  4:         **for** $b \in G_g$ **do**
  5:            $y_b \leftarrow x_b \, C_g / S_g$                                ▷ proportional rescaling
  6:         **end for**
  7:      **else**
  8:         **for** $b \in G_g$ **do**
  9:            $y_b \leftarrow C_g / |G_g|$                                    ▷ uniform split
10:         **end for**
11:      **end if**
12: **end for**
13: **return** $\{y_b\}$

---

**Randomized rounding.** Algorithm 6 independently rounds a real value to the nearest integers, preserving the value in expectation. It runs in constant time and memory per entry, so applying it over $B$ items has

$$T = O(B), \qquad M_{\text{extra}} = O(1).$$

---

**Algorithm 6** Randomized Rounding (scalar form)

---

**Require:** real value $x \geq 0$
**Ensure:** integer $y \in \mathbb{Z}_{\geq 0}$

  1: $a \leftarrow \lfloor x \rfloor$
  2: $r \leftarrow x - a$
  3: sample $U \sim \text{Unif}[0, 1]$
  4: **if** $U < r$ **then**
  5:      $y \leftarrow a + 1$
  6: **else**
  7:      $y \leftarrow a$
  8: **end if**
  9: **return** $y$

---

**Groupwise exact rounding.** Algorithm 7 converts rescaled real values $\{x_b\}$ to integers $\{y_b\}$ while *exactly* preserving each group sum: $\sum_{b \in G_g} y_b = C_g$. Each $y_b$ differs from $x_b$ by at most 1. The only expensive step is weighted sampling without replacement inside each group. With Gumbel–Top-$k$ sampling the worst-case complexity is

$$T = O\left( \sum_{g=1}^{G} |G_g| \log |G_g| \right), \qquad M_{\text{extra}} = O(B),$$

and the algorithm is again applied independently over coordinates.

---

**Algorithm 7** Groupwise Randomized Rounding with Exact Aggregates (scalar form)

---

**Require:** rescaled $x_b \geq 0$, groups $G_1, \ldots, G_G$, integer targets $C_g$
**Ensure:** integers $y_b \geq 0$ with $\sum_{b \in G_g} y_b = C_g$ and $|y_b - x_b| \leq 1$

  1: **for** $g = 1, \ldots, G$ **do**                                     ▷ work inside group $G_g$
  2:     **for** $b \in G_g$ **do**
  3:         $a_b \leftarrow \lfloor x_b \rfloor$
  4:         $r_b \leftarrow x_b - a_b$
  5:     **end for**
  6:     $A_g \leftarrow \sum_{b \in G_g} a_b$
  7:     $S_g \leftarrow C_g - A_g$                                   ▷ # of increments required
  8:     **if** $S_g = 0$ **then**
  9:         **for** $b \in G_g$ **do**
10:             $y_b \leftarrow a_b$
11:         **end for**
12:     **else**
13:         sample a subset $S \subseteq G_g$ of size $S_g$
14:         **without replacement** with weights $\propto r_b$
15:         **for** $b \in G_g$ **do**
16:             **if** $b \in S$ **then**
17:                 $y_b \leftarrow a_b + 1$
18:             **else**
19:                 $y_b \leftarrow a_b$
20:             **end if**
21:         **end for**
22:     **end if**
23: **end for**
24: **return** $\{y_b\}$

---

## D   SYNTHETIC DISTRIBUTIONS

All synthetic tasks use the same base architecture with a 4-layer MLP with 128 dimensional hidden layers. We scale the inputs and outputs in dimension, so for example the DFM and CE-CB have $d \times 256$ dimensional outputs (since we clip all datasets to use a range of 256 to make for easy tokenization). The energy score models take inputs in $d + \text{noise\_dim}$ and we use $\text{noise\_dim} = 100$ throughout. We ran all experiments with Adam using both $lr = 1e - 3, 2e - 4$ and present results for the best performing learning rate for each method. We use a cosine warmup for the learning rate for 100 steps. For all experiments we use gradient norm clipping to size 1, batch size 256, and train for 500 epochs. For the energy score models, we use exponential model averaging, which is crucial to good performance. Full details are available in the codebase.

For the flow matching we use $\sigma = 0.1$ following best practices (we tested larger $\sigma$ but saw large degradations in performance). For the bessel sampler we use $\sqrt{\Lambda_+ \Lambda_-} = 32$.

### D.1   DISCRETE 8-GAUSSIANS TO 2-MOONS

#### D.1.1   DATASET

For qualitative evaluation, we adapt the classic continuous "8-Gaussians to 2-Moons" task into a fully discrete, integer-valued setting suitable for count-based flow matching. Each dataset consists of 50,000 paired samples $(x_0, x_1) \in \mathbb{Z}^2$, constructed as follows.

**Source distribution** ($x_0$). We generate samples from the standard two-moons dataset in $\mathbb{R}^2$ using `make_moons` with noise level `noise = 0.1`. The moons are shifted to be approximately centered at the origin by subtracting $(0.5, 0.25)$.

**Target distribution** ($x_1$). We construct an 8-component Gaussian mixture arranged evenly on a circle of radius 2.0 in $\mathbb{R}^2$. Each component has isotropic Gaussian noise with variance matching

`noise = 0.1`. A sample is generated by first selecting one of the 8 components uniformly at random, then drawing from the corresponding Gaussian.

**Integerization.** Both source and target samples are mapped to the integer lattice by

$$x \mapsto \text{round}(\text{clip}(x \cdot \texttt{scale} + \texttt{offset}, \texttt{min\_value}, \texttt{value\_range} - 1)),$$

with parameters $\texttt{scale} = 30.0$, $\texttt{offset} = 80.0$, $\texttt{min\_value} = 0$, and $\texttt{value\_range} = 196$. This procedure ensures that all outputs fall in the discrete vocabulary $\{0, 1, \ldots, 195\}^2$, but the scales are chosen so that essentially no values are actually clipped.

### D.1.2 RESULTS

We present a visualization of the learned trajectories in Fig. 2 and the full details in Table 6. Count bridges achieve uniformly the best performance using the distributional losses, that is the cross entropy or energy scores with the energy score uniformly best.

Table 6: Discrete Moons Results: Noise → Two Moons

| Method | MMD | $W_2$ | Energy |
|---|---|---|---|
| CFM | $0.065 \pm 0.019$ | $0.049 \pm 0.008$ | $0.874 \pm 0.246$ |
| DFM | $0.010 \pm 0.002$ | $0.010 \pm 0.002$ | $0.035 \pm 0.014$ |
| Count Bridge (CE) | $0.0065 \pm 0.0023$ | $0.0080 \pm 0.0009$ | $0.026 \pm 0.004$ |
| Count Bridge (ES) | $\mathbf{0.0044 \pm 0.0018}$ | $\mathbf{0.0052 \pm 0.0007}$ | $\mathbf{0.0098 \pm 0.0029}$ |
| Count Bridge (MSE) | $0.030 \pm 0.000$ | $0.033 \pm 0.001$ | $0.366 \pm 0.015$ |

We also run the Count Bridge across different noise levels, here we actually find that our default of $\lambda_+ = \lambda_- = 32$ is not optimized, so all results can be considered lower bounds on our performance.

Table 7: Count Bridge (Energy Score) Results Across Different $\lambda_+ = \lambda_-$ Values

| $\lambda_+ = \lambda_-$ | MMD | $W_2$ | Energy |
|---|---|---|---|
| 0 | $\mathbf{0.0038 \pm 0.0012}$ | $0.0046 \pm 0.0002$ | $\mathbf{0.0075 \pm 0.0011}$ |
| 8 | $0.0039 \pm 0.0015$ | $\mathbf{0.0045 \pm 0.0006}$ | $0.0080 \pm 0.0022$ |
| 16 | $0.0049 \pm 0.0003$ | $0.0049 \pm 0.0003$ | $0.0095 \pm 0.0004$ |
| 32 | $0.0052 \pm 0.0024$ | $0.0055 \pm 0.0015$ | $0.011 \pm 0.006$ |
| 256 | $0.0064 \pm 0.0020$ | $0.0063 \pm 0.0009$ | $0.015 \pm 0.004$ |

### D.1.3 NOISE TO 2-MOONS FOR DIFFUSION COMPARISONS

Here we compare against a standard Gaussian DDIM model Song et al. (2020) and Discrete Diffusion as in Shi et al. (2024). Since these models go from noise to a target distribution, we cannot do the 8-Gaussians to 2-Moons task, so we simply target the 2-Moons. This makes the task substantially easier. For Count Bridges we use the same results from the previous table (the more difficult task, but with comparable scores at the endpoint).

Table 8: Discrete Moons Results: Eight Gaussians → Two Moons

| Method | MMD | $W_2$ | Energy |
|---|---|---|---|
| Count Bridge (ES) | $\mathbf{0.0044 \pm 0.0018}$ | $\mathbf{0.0052 \pm 0.0007}$ | $\mathbf{0.0098 \pm 0.0029}$ |
| Gaussian Diffusion | $0.024 \pm 0.009$ | $0.017 \pm 0.006$ | $0.118 \pm 0.064$ |
| Discrete Diffusion | $0.017 \pm 0.010$ | $0.013 \pm 0.006$ | $0.072 \pm 0.055$ |

### D.2 Low-Rank Gaussian Mixture

#### D.2.1 Dataset

For synthetic evaluation, we use a pre-sampled integer-valued Gaussian mixture dataset that scales with the ambient dimension $d$ while fixing the latent rank at $r = 3$. Each dataset consists of 50,000 paired samples $(x_0, x_1) \in \mathbb{Z}^d$ generated according to the following procedure:

**Mixture construction.** We define a $k = 5$ component Gaussian mixture in latent space $\mathbb{R}^r$ with $r = 3$ (we hold these parameters constant as we scale in $d$):

- **Means.** Component means are drawn from $\mathcal{N}(0, \sigma^2 I)$ with scale $\sigma = \texttt{mean\_scale}/\sqrt{r}$, and shifted to lie near the center of the integer range. We set $\texttt{mean\_scale} = 20.0$.

- **Covariances.** Each covariance is constructed by sampling eigenvalues from an exponential distribution with scale $\texttt{cov\_scale} = 10.0$, clamped below $\texttt{min\_eigenvalue} = 0.1$, and conjugating by a random orthogonal matrix.

- **Mixture weights.** Weights are drawn from a $\text{Dirichlet}(1, \dots, 1)$ prior, yielding a random simplex vector.

**Projection to $\mathbb{R}^d$.** Latent samples $z \in \mathbb{R}^3$ are mapped to the ambient space via a random projection matrix $P \in \mathbb{R}^{d \times r}$ with entries scaled by $\texttt{projection\_scale}/\sqrt{r}$, where $\texttt{projection\_scale} = 1.0$. To avoid degeneracy, isotropic Gaussian noise $\epsilon \sim \mathcal{N}(0, \texttt{noise\_scale}^2 I_d)$ with $\texttt{noise\_scale} = 1.0$ is added after projection:

$$y = Pz + \epsilon, \qquad z \sim \text{MoG}_r, \ \epsilon \sim \mathcal{N}(0, I_d).$$

**Integerization.** Projected samples $y \in \mathbb{R}^d$ are rounded to the nearest integer and reflected into the bounded range $[\texttt{min\_value}, \texttt{value\_range} - 1] = [0, 255]$ to ensure validity of DFM.

**Scaling in $d$.** The intrinsic latent structure is fixed at $r = 3$, while the output dimension $d$ is varied across experiments (e.g. $d = 5, 16, 32, 128, 256, 512$). This construction produces datasets with constant intrinsic complexity but increasing ambient dimension, providing a natural test of how models scale in $d$.

#### D.2.2 Results

The central scaling results are presented visually in Fig. 3. Here we also present the full experimental details in Table 9.

### D.3 Deconvolution Gaussian Mixture Dataset

We extend the low-rank Gaussian mixture task (Appendix D.2) to evaluate deconvolution capabilities under controlled conditions. Each observation is formed by aggregating a group of $G$ unit-level samples into a single count vector.

**Group construction.** For each group, component proportions are drawn from a Dirichlet distribution with concentration parameter $\alpha$, yielding group-specific mixture weights. The $G$ unit-level samples are then drawn independently from the corresponding mixture components. Both the aggregated sum $X_0 \in \mathbb{Z}^d$ and the individual unit-level labels $z \in \{0, 1\}^{G \times k}$ are retained, enabling evaluation of methods under both aggregate-only and aggregate+unit supervision.

**Experimental variation.** We vary two factors that control the difficulty of deconvolution:

- **Group size:** $G \in \{4, 8, 32, 128\}$, which determines how many unit-level samples are aggregated. Larger groups yield more uniform averages and less information about component heterogeneity.

- **Dirichlet concentration:** $\alpha \in \{1, 10, 1000\}$, which controls variability in group-specific mixture weights. Small $\alpha$ values produce heterogeneous groups (informative for deconvolution), while large $\alpha$ values yield nearly uniform group proportions (uninformative).

Table 9: Performance Comparison Across Dimensions and Methods

| Dim | Method | NFE | MMD | $W_2$ | EMD |
|---|---|---|---|---|---|
| 4 | CFM | 8 | $0.027 \pm 0.014$ | $0.019 \pm 0.002$ | $0.716 \pm 0.420$ |
| | | 32 | $0.026 \pm 0.017$ | $0.015 \pm 0.004$ | $0.584 \pm 0.460$ |
| | | 128 | $0.026 \pm 0.018$ | $0.015 \pm 0.004$ | $0.565 \pm 0.469$ |
| | DFM | 8 | $0.025 \pm 0.004$ | $0.011 \pm 0.001$ | $0.245 \pm 0.053$ |
| | | 32 | $0.035 \pm 0.003$ | $0.014 \pm 0.001$ | $0.458 \pm 0.078$ |
| | | 128 | $0.046 \pm 0.005$ | $0.018 \pm 0.002$ | $0.759 \pm 0.144$ |
| | Count Bridge | 8 | $\mathbf{0.0053 \pm 0.0007}$ | $\mathbf{0.0040 \pm 0.0004}$ | $\mathbf{0.020 \pm 0.004}$ |
| | | 32 | $\mathbf{0.0054 \pm 0.0010}$ | $\mathbf{0.0042 \pm 0.0002}$ | $\mathbf{0.023 \pm 0.005}$ |
| | | 128 | $\mathbf{0.0098 \pm 0.0008}$ | $\mathbf{0.0058 \pm 0.0003}$ | $\mathbf{0.055 \pm 0.008}$ |
| 8 | CFM | 8 | $0.041 \pm 0.013$ | $0.025 \pm 0.004$ | $1.05 \pm 0.18$ |
| | | 32 | $0.039 \pm 0.012$ | $0.014 \pm 0.004$ | $0.456 \pm 0.147$ |
| | | 128 | $0.040 \pm 0.010$ | $0.014 \pm 0.003$ | $0.421 \pm 0.128$ |
| | DFM | 8 | $0.026 \pm 0.007$ | $0.011 \pm 0.001$ | $0.204 \pm 0.067$ |
| | | 32 | $0.034 \pm 0.009$ | $0.011 \pm 0.003$ | $0.317 \pm 0.142$ |
| | | 128 | $0.042 \pm 0.011$ | $0.012 \pm 0.003$ | $0.497 \pm 0.228$ |
| | Count Bridge | 8 | $\mathbf{0.0036 \pm 0.0007}$ | $\mathbf{0.0023 \pm 0.0001}$ | $\mathbf{0.0068 \pm 0.0024}$ |
| | | 32 | $\mathbf{0.0038 \pm 0.0011}$ | $\mathbf{0.0026 \pm 0.0006}$ | $\mathbf{0.0077 \pm 0.0028}$ |
| | | 128 | $\mathbf{0.0050 \pm 0.0015}$ | $\mathbf{0.0029 \pm 0.0003}$ | $\mathbf{0.012 \pm 0.002}$ |
| 16 | CFM | 8 | $0.066 \pm 0.011$ | $0.028 \pm 0.001$ | $2.08 \pm 0.54$ |
| | | 32 | $0.053 \pm 0.011$ | $0.017 \pm 0.001$ | $0.788 \pm 0.211$ |
| | | 128 | $0.052 \pm 0.011$ | $0.015 \pm 0.001$ | $0.647 \pm 0.163$ |
| | DFM | 8 | $0.078 \pm 0.001$ | $0.017 \pm 0.000$ | $1.20 \pm 0.06$ |
| | | 32 | $0.100 \pm 0.005$ | $0.022 \pm 0.002$ | $1.92 \pm 0.28$ |
| | | 128 | $0.118 \pm 0.017$ | $0.025 \pm 0.004$ | $2.72 \pm 0.86$ |
| | Count Bridge | 8 | $\mathbf{0.0067 \pm 0.0014}$ | $\mathbf{0.0035 \pm 0.0003}$ | $\mathbf{0.025 \pm 0.007}$ |
| | | 32 | $\mathbf{0.011 \pm 0.001}$ | $\mathbf{0.0045 \pm 0.0004}$ | $\mathbf{0.048 \pm 0.007}$ |
| | | 128 | $\mathbf{0.017 \pm 0.001}$ | $\mathbf{0.0057 \pm 0.0004}$ | $\mathbf{0.090 \pm 0.013}$ |
| 32 | CFM | 8 | $0.145 \pm 0.024$ | $0.030 \pm 0.001$ | $4.63 \pm 1.28$ |
| | | 32 | $0.131 \pm 0.043$ | $0.026 \pm 0.005$ | $3.14 \pm 1.72$ |
| | | 128 | $0.131 \pm 0.052$ | $0.022 \pm 0.006$ | $3.09 \pm 1.97$ |
| | DFM | 8 | $0.079 \pm 0.027$ | $0.016 \pm 0.008$ | $1.23 \pm 0.76$ |
| | | 32 | $0.089 \pm 0.023$ | $0.017 \pm 0.006$ | $1.55 \pm 0.85$ |
| | | 128 | $0.100 \pm 0.027$ | $0.018 \pm 0.007$ | $1.99 \pm 1.10$ |
| | Count Bridge | 8 | $\mathbf{0.0083 \pm 0.0008}$ | $\mathbf{0.0024 \pm 0.0003}$ | $\mathbf{0.021 \pm 0.002}$ |
| | | 32 | $\mathbf{0.010 \pm 0.002}$ | $\mathbf{0.0031 \pm 0.0006}$ | $\mathbf{0.029 \pm 0.008}$ |
| | | 128 | $\mathbf{0.010 \pm 0.002}$ | $\mathbf{0.0034 \pm 0.0007}$ | $\mathbf{0.034 \pm 0.007}$ |
| 64 | CFM | 8 | $0.296 \pm 0.077$ | $0.042 \pm 0.008$ | $13.43 \pm 6.74$ |
| | | 32 | $0.313 \pm 0.099$ | $0.046 \pm 0.014$ | $16.07 \pm 9.93$ |
| | | 128 | $0.326 \pm 0.107$ | $0.049 \pm 0.017$ | $17.94 \pm 11.55$ |
| | DFM | 8 | $0.105 \pm 0.033$ | $0.022 \pm 0.007$ | $1.71 \pm 1.07$ |
| | | 32 | $0.126 \pm 0.039$ | $0.020 \pm 0.005$ | $2.57 \pm 1.58$ |
| | | 128 | $0.147 \pm 0.048$ | $0.022 \pm 0.006$ | $3.59 \pm 2.20$ |
| | Count Bridge | 8 | $\mathbf{0.020 \pm 0.004}$ | $\mathbf{0.0051 \pm 0.0005}$ | $\mathbf{0.072 \pm 0.019}$ |
| | | 32 | $\mathbf{0.027 \pm 0.002}$ | $\mathbf{0.0061 \pm 0.0002}$ | $\mathbf{0.112 \pm 0.014}$ |
| | | 128 | $\mathbf{0.029 \pm 0.001}$ | $\mathbf{0.0065 \pm 0.0005}$ | $\mathbf{0.138 \pm 0.018}$ |
| 128 | CFM | 8 | $0.335 \pm 0.009$ | $0.038 \pm 0.003$ | $12.89 \pm 1.09$ |
| | | 32 | $0.276 \pm 0.034$ | $0.033 \pm 0.008$ | $10.59 \pm 3.51$ |
| | | 128 | $0.260 \pm 0.048$ | $0.032 \pm 0.008$ | $10.55 \pm 4.74$ |
| | DFM | 8 | $0.205 \pm 0.036$ | $0.042 \pm 0.005$ | $6.66 \pm 2.47$ |
| | | 32 | $0.236 \pm 0.031$ | $0.043 \pm 0.005$ | $9.06 \pm 2.63$ |
| | | 128 | $0.259 \pm 0.028$ | $0.050 \pm 0.011$ | $10.90 \pm 2.92$ |
| | Count Bridge | 8 | $\mathbf{0.128 \pm 0.066}$ | $\mathbf{0.014 \pm 0.006}$ | $\mathbf{3.30 \pm 2.34}$ |
| | | 32 | $\mathbf{0.140 \pm 0.075}$ | $\mathbf{0.014 \pm 0.006}$ | $\mathbf{3.89 \pm 2.97}$ |
| | | 128 | $\mathbf{0.151 \pm 0.082}$ | $\mathbf{0.016 \pm 0.007}$ | $\mathbf{4.55 \pm 3.65}$ |
| 256 | CFM | 8 | $0.461 \pm 0.007$ | $0.049 \pm 0.007$ | $28.45 \pm 4.63$ |
| | | 32 | $0.402 \pm 0.049$ | $0.047 \pm 0.006$ | $28.95 \pm 10.71$ |
| | | 128 | $0.390 \pm 0.069$ | $0.045 \pm 0.012$ | $30.57 \pm 13.24$ |
| | DFM | 8 | $0.216 \pm 0.049$ | $0.029 \pm 0.008$ | $9.75 \pm 5.07$ |
| | | 32 | $0.228 \pm 0.045$ | $0.033 \pm 0.008$ | $12.63 \pm 4.81$ |
| | | 128 | $0.255 \pm 0.044$ | $0.039 \pm 0.009$ | $15.80 \pm 4.96$ |
| | Count Bridge | 8 | $\mathbf{0.087 \pm 0.045}$ | $\mathbf{0.0093 \pm 0.0022}$ | $\mathbf{1.58 \pm 1.28}$ |
| | | 32 | $\mathbf{0.092 \pm 0.044}$ | $\mathbf{0.0099 \pm 0.0021}$ | $\mathbf{1.68 \pm 1.30}$ |
| | | 128 | $\mathbf{0.105 \pm 0.039}$ | $\mathbf{0.012 \pm 0.001}$ | $\mathbf{1.98 \pm 1.26}$ |
| 512 | CFM | 8 | $0.569 \pm 0.055$ | $0.051 \pm 0.006$ | $49.32 \pm 7.46$ |
| | | 32 | $0.471 \pm 0.046$ | $0.049 \pm 0.005$ | $50.69 \pm 11.35$ |
| | | 128 | $0.438 \pm 0.034$ | $0.048 \pm 0.005$ | $53.70 \pm 14.23$ |
| | DFM | 8 | $0.261 \pm 0.069$ | $0.042 \pm 0.015$ | $30.49 \pm 21.88$ |
| | | 32 | $0.288 \pm 0.099$ | $0.050 \pm 0.019$ | $44.53 \pm 29.76$ |
| | | 128 | $0.319 \pm 0.112$ | $0.058 \pm 0.022$ | $55.03 \pm 35.35$ |
| | Count Bridge | 8 | $\mathbf{0.081 \pm 0.028}$ | $\mathbf{0.010 \pm 0.002}$ | $\mathbf{1.46 \pm 0.73}$ |
| | | 32 | $\mathbf{0.085 \pm 0.026}$ | $\mathbf{0.010 \pm 0.001}$ | $\mathbf{1.79 \pm 0.89}$ |
| | | 128 | $\mathbf{0.113 \pm 0.029}$ | $\mathbf{0.016 \pm 0.003}$ | $\mathbf{3.53 \pm 2.27}$ |

**Dataset parameters.** We fix the ambient dimension at $d = 4$, latent rank at $r = 3$, number of mixture components $k = 5$, and use the same mixture parameterization as in the low-rank dataset (means scaled by 20.0, covariances scaled by 10.0 with minimum eigenvalue 0.1, projection scale 1.0, isotropic noise 1.0, and integerization into $[0, 255]$). Each dataset contains 5,000 groups, drawn from a base pool of 50,000 pre-sampled mixture samples.

**Results.** As shown in Fig. 4, deconvolution performance degrades as groups become larger and more uniform. This matches the theoretical results in Appendices B.2 and B.3, which establish that deconvolution requires between-group heterogeneity for identification, a property that is inherently lost as $G$ grows. We present detailed results across metrics in Tables 10, 11, 12.

Table 10: Deconvolution Performance: $W_2$ vs Group Size and Dirichlet Concentration

| Group Size (n) | $\alpha = 1$ | $\alpha = 10$ | $\alpha = 1000$ |
|---|---|---|---|
| 4 | $0.0091 \pm 0.0005$ | $0.010 \pm 0.000$ | $0.011 \pm 0.000$ |
| 8 | $0.011 \pm 0.001$ | $0.014 \pm 0.001$ | $0.016 \pm 0.000$ |
| 32 | $0.020 \pm 0.002$ | $0.023 \pm 0.002$ | $0.025 \pm 0.002$ |
| 128 | $0.050 \pm 0.008$ | $0.053 \pm 0.006$ | $0.057 \pm 0.002$ |

Table 11: Deconvolution Performance: EMD vs Group Size and Dirichlet Concentration

| Group Size (n) | $\alpha = 1$ | $\alpha = 10$ | $\alpha = 1000$ |
|---|---|---|---|
| 4 | $0.130 \pm 0.006$ | $0.195 \pm 0.025$ | $0.207 \pm 0.040$ |
| 8 | $0.286 \pm 0.023$ | $0.363 \pm 0.037$ | $0.483 \pm 0.010$ |
| 32 | $0.530 \pm 0.051$ | $0.657 \pm 0.095$ | $0.921 \pm 0.222$ |
| 128 | $2.24 \pm 0.58$ | $2.22 \pm 0.54$ | $2.63 \pm 0.14$ |

Table 12: Deconvolution Performance: MMD vs Group Size and Dirichlet Concentration

| Group Size (n) | $\alpha = 1$ | $\alpha = 10$ | $\alpha = 1000$ |
|---|---|---|---|
| 4 | $0.011 \pm 0.001$ | $0.011 \pm 0.002$ | $0.012 \pm 0.004$ |
| 8 | $0.016 \pm 0.001$ | $0.017 \pm 0.002$ | $0.021 \pm 0.001$ |
| 32 | $0.014 \pm 0.003$ | $0.020 \pm 0.003$ | $0.025 \pm 0.010$ |
| 128 | $0.037 \pm 0.005$ | $0.045 \pm 0.007$ | $0.044 \pm 0.001$ |

To investigate the importance of different rounding approaches in our deconvolution implementation we run an ablation study where we substitute our preferred exact rounding approach for two alternatives: first a simple deterministic rounding, and second a randomized rounding (where we simply add zero or one with probability of the decimal value). Deterministic rounding can lead to arbitrarily incorrect results, randomized rounding preserves the expectation, and our exact approach will ensure we exactly match the target aggregate value. We find that although our exact approach is superior across three replicates of the $n = 128, \alpha = 1$ setting the results are very close particularly for the randomized rounding. We believe this could justify substituting the randomized approach since it is simpler, although in different regimes this may matter more or less.

Table 13: Performance comparison of different rounding methods with standard errors.

| Method | MMD | $W_2$ | EMD |
|---|---|---|---|
| exact | $\mathbf{0.037 \pm 0.005}$ | $\mathbf{0.050 \pm 0.008}$ | $\mathbf{2.24 \pm 0.58}$ |
| randomized | $0.038 \pm 0.000$ | $0.051 \pm 0.007$ | $2.33 \pm 0.43$ |
| round | $0.038 \pm 0.003$ | $0.052 \pm 0.007$ | $2.32 \pm 0.44$ |

# E NUCLEOTIDE-LEVEL GENE EXPRESSION MODELLING

## E.1 DATASET

**Nucleotide-level data preprocessing** We use the Onek1k peripheral blood mononuclear cells (PBMC) 10X 3' scRNA-seq dataset, originally collected by Yazar et al. (2022). For our analysis, as we are interested in nucleotide-level counts rather than the gene-level counts provided with the initial publication, we use the preprocessed reads made available by Hingerl et al. (2024). The reads are

aligned to the hg38 human reference genome. The resulting BAM files are filtered to include only high quality, UMI-deduplicated reads. The cell type annotations were used as provided in the original dataset.

**Gene-level data preprocessing**  We construct the gene-level count matrices directly from our single-cell coverage matrices rather than following the typical single-cell gene expression preprocessing pipeline. In particular, for each annotated gene and each cell, we take the max count over the nucleotide-level coverage matrix as the count for the gene.

## E.2    ARCHITECTURE, TRAINING, AND INFERENCE

### E.2.1    INPUTS AND EMBEDDINGS

We model nucleotide–level counts on a fixed window of length $L=896$. For each example we form

$$x_t \in \mathbb{Z}_{\geq 0}^L, \quad t \in (0,1], \quad z \sim \mathcal{N}(0, I_{d_z}), \quad c \in \{1, \dots, C\}, \quad \text{seq} \in \{\text{A}, \text{C}, \text{G}, \text{T}, \text{N}\}^L.$$

Sequence context is embedded with a frozen Enformer encoder (EleutherAI checkpoint), yielding per–position embeddings

$$E(\text{seq}) \in \mathbb{R}^{L \times d_E}, \qquad d_E = 3072.$$

We tile the scalars across positions and concatenate

$$H^{(0)} \;=\; \left[\, E(\text{seq}) \,\|\, x_t \,\|\, t \,\|\, z \,\|\, \text{emb}(c) \,\right] \;\in\; \mathbb{R}^{L \times (d_E + 1 + 1 + d_z + d_c)},$$

with $d_z=100$ and $d_c=14$. A two–layer SELU MLP projects to the model width $d$:

$$X^{(0)} \;=\; \phi\!\big(W_2\, \phi(W_1 H^{(0)})\big) \in \mathbb{R}^{L \times d}.$$

### E.2.2    LOCAL ATTENTION BACKBONE

We apply $N_{\text{attn}}$ residual self–attention blocks (PyTorch `MultiheadAttention`, batch–first) with LayerNorm:

$$\tilde{X}^{(\ell)} \;=\; \text{MHA}\big(X^{(\ell)}, X^{(\ell)}, X^{(\ell)}\big), \qquad X^{(\ell+1)} \;=\; \text{LN}\big(\tilde{X}^{(\ell)} + X^{(0)}\big),$$

where the residual skip uses the pre–block $X^{(0)}$ as in the implementation.[2] We use $N_{\text{attn}}=2$ layers, $d=\text{hidden\_dim}$, and $h=4$ heads. A linear projection followed by `softplus` produces a nonnegative per–position prediction

$$\hat{x}_0 \;=\; \text{softplus}\big(W_{\text{out}} X^{(N_{\text{attn}})}\big) \in \mathbb{R}_{\geq 0}^L.$$

This parameterizes the conditional law $q_\theta(\,\cdot\,|\, x_t, t, z, c, \text{seq})$ used inside the count–bridge reverse kernel (Sec. 3).

### E.2.3    LEARNED PROJECTION MODULE $\Pi_\psi$

When an aggregate constraint $a_0 = \sum_{i=1}^L x_{0,i}$ is observed, we refine $\hat{x}_0$ with a lightweight attention projector that operates across *positions*. We form

$$Y^{(0)} \;=\; \left[\, \hat{x}_0 \,\|\, x_t \,\|\, a_0 \,\|\, X^{(N_{\text{attn}})} \,\right] \in \mathbb{R}^{L \times (1+1+1+d)}.$$

A two–layer SELU MLP lifts to width $d$, then $N_{\text{proj}}=2$ self–attention layers (sequence–first API) with residual+LayerNorm are applied:

$$\tilde{Y}^{(m)} = \text{MHA}(Y^{(m)}, Y^{(m)}, Y^{(m)}), \quad Y^{(m+1)} = \text{LN}\big(\tilde{Y}^{(m)} + Y^{(0)}\big).$$

A linear head produces an additive correction which we re–softplus for nonnegativity:

$$\tilde{x}_0 \;=\; \text{softplus}\big(W_{\text{proj}} Y^{(N_{\text{proj}})}\big) \;+\; \hat{x}_0.$$

At inference, when $a_0$ is present we use $\tilde{x}_0$ as the endpoint in the reverse step; otherwise we use $\hat{x}_0$.

---

[2]Code uses a "global" residual $X \leftarrow X + X^{(0)}$ within each block. We retained this because it stabilized training with $L=896$.

### E.2.4 TRAINING OBJECTIVES AND SCHEDULES

**Distributional loss (energy score).** We train $q_\theta$ with the energy score $S_\rho$ on the conditional $X_0 \mid X_t = x_t$ (Sec. 3.2). For each example we draw $m$ i.i.d. samples $\hat{x}_0^{(j)} \sim q_\theta(\cdot \mid x_t, t, z, c, \text{seq})$ via ancestral decoding of the per–position parameterization and estimate the $U$–statistic version of $S_\rho$ with $\rho(x, x') = \|x - x'\|_2^\beta$ ($\beta \in (0, 2)$). We used $m = 2$ in practice.

**Aggregate–aware training.** With probability $p_{\text{agg}} = 0.1$ we attach an aggregate $a_0$ and route the forward pass through $\Pi_\psi$ to obtain $\tilde{x}_0$, then compute the same energy score. This jointly trains $\Pi_\psi$ to approximate sampling from the mean–conditional $X_0 \mid A(X_0) = a_0, X_t, t$ while preserving the exact reverse transition of the count bridge.

**Cell–type masking.** To support both conditional and unconditional generation, we randomly mask the cell–type embedding with probability $p_{\text{mask}}$ (set to zero vector). We used $p_{\text{mask}} = 0.1$.

### E.2.5 OPTIMIZATION AND HYPERPARAMETERS

We use Adam, learning rate $\{2 \times 10^{-4}$, cosine warmup for 100 steps, EMA with 0.999, batch size 128, gradient clipping at 1.0. See configs for exact architecture specification.

### E.2.6 SAMPLING

At test time we follow Alg. 2: starting from $x_1$ we iterate $t_k \downarrow 0$. At each step we sample $\tilde{X}_0 \sim q_\theta(\cdot \mid x_{t_k}, t_k, z, c, \text{seq})$; if an aggregate is provided we replace with $\tilde{x}_0 = \Pi_\psi(\hat{x}_0, a_0, x_{t_k})$. We then apply the exact binomial–hypergeometric reverse kernel (Prop. 3.1) to obtain $x_{t_{k-1}}$. This guarantees that trajectories remain within the discrete support while leveraging the learned distributional posterior. We use three function evaluations for all results in this application.

### E.3 ADDITIONAL RESULTS

In Tab. E.3 we provide results for gene expression prediction performance, broken down by cell type.

| Cell type | Baseline MSE | CB MSE |
| --- | --- | --- |
| CD4 ET | 3.596 | **1.402** |
| NK | 0.415 | **0.364** |
| CD4 NC | 3.382 | **1.304** |
| CD8 S100B | 2.619 | **1.002** |
| CD8 ET | 1.065 | **0.540** |
| B IN | 2.556 | **1.091** |
| CD8 NC | 3.381 | **1.311** |
| B Mem | 6.742 | **3.416** |
| NK R | 1.624 | **0.781** |
| Mono NC | 1.485 | **0.752** |
| Mono C | 1.253 | **0.676** |
| DC | 9.302 | **4.475** |
| Plasma | 10763.906 | **10696.934** |
| CD4 SOX4 | 3.428 | **1.323** |

We also analyze the unit-level profiles of the deconvolved transciptomes. We aggregate the nucleotide-level transcriptomes up to the gene level by computing the maximum count over the gene profile, enabling us to generate a (standard cell by gene) count matrix from our deconvolved profile. We then plot the UMAP of the held-out ground truth alongside the deconvolved transcriptomic profiles, and find that the cells are well-mixed (Fig. 5), suggesting that generated expression distributions match the true distributions.

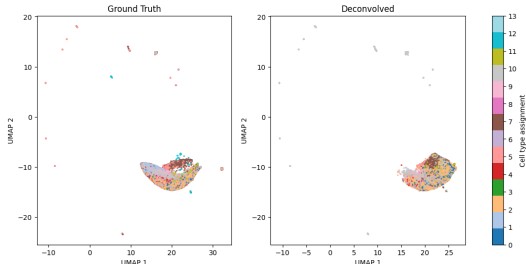

Figure 5: UMAP of true single-cell and CB-deconvolved nucleotide-level expression, aggregated to the gene level and hued by cell type (as assigned by Yazar et al. (2022)).

## F  SPATIAL TRANSCRIPTOMIC DECONVOLUTION

### F.1  DATASET

**Preprocessing**  We used the publicly available mouse brain MERFISH dataset from Vizgen (2021). We subset the data to a particular slice (slice 1, replicate 2). We used the transcript puncta and nuclear segmentation masks as provided with the dataset. For gene expression, we used the raw transcript counts without applying standard single-cell preprocessing pipelines. For each cell, we resized the DAPI image to 256x256 pixels by padding.

**Aggregation**  To simulate a Visium-style spatial transcriptomics dataset, we aggregated the single-cell MERFISH data. A grid of spots was defined with a center-to-center distance of $100\mu m$ and a spot radius of $55\mu m$. The gene expression profile for each simulated spot was then generated by summing the transcript counts of all identified cells whose nuclei fell within the circular bounds of that spot.

### F.2  ARCHITECTURE, TRAINING, AND SAMPLING

#### F.2.1  INPUTS AND TOKENIZATION

We model spot-level *counts* while conditioning on *image* context and diffusion *noise/time* tokens. Each training example provides

$$x_t^{\text{cnt}} \in \mathbb{Z}_{\geq 0}^{D_c}, \quad I_t \in \mathbb{R}^{C \times H \times W}, \quad t \in (0,1], \quad \varepsilon \in \mathbb{R}^{d_\varepsilon}, \quad y \in \{1, \ldots, C_y\} \text{ (optional)}.$$

Images are patchified by a ViT-style embedder (PatchEmbed) into

$$X^{\text{img}} \in \mathbb{R}^{B \times N_{\text{img}} \times d}, \quad N_{\text{img}} = (H/P)(W/P),$$

while counts are converted into a small set of *count patches* using a learned projector (CountPatchEmbedding):

$$X^{\text{cnt}} = \text{reshape}\Big(\text{MLP}(x_t^{\text{cnt}}), [B, N_{\text{cnt}}, d]\Big) \; + \; E_{\text{cnt}},$$

with $N_{\text{cnt}}$ learned "pseudo-patches" and $E_{\text{cnt}}$ learnable positional embeddings.

We form auxiliary tokens for time, noise, and (optionally) class:

$$\underbrace{\tau}_{\text{time}} = \text{TimeMLP}\big(\text{timestep\_emb}(t)\big) \in \mathbb{R}^{B \times 1 \times d}, \quad \underbrace{\eta}_{\text{noise}} = \text{NoiseMLP}\big(W_\varepsilon \varepsilon\big) \in \mathbb{R}^{B \times 1 \times d},$$

and, if labels are used, $\ell = \text{Emb}(y) \in \mathbb{R}^{B \times 1 \times d}$. Concatenating all tokens,

$$X^{(0)} = \big[\, \ell \,;\, \eta \,;\, \tau \,;\, X^{\text{img}} \,;\, X^{\text{cnt}} \,\big] \; + \; E_{\text{pos}} \in \mathbb{R}^{B \times (N_{\text{img}} + N_{\text{cnt}} + \text{extras}) \times d},$$

with a single learned positional table $E_{\text{pos}}$ covering all tokens.

#### F.2.2  U-VIT BACKBONE (FUSION AND DECODING)

We process $X^{(0)}$ with a U-Net–style ViT:

$$\underbrace{X^{(1)}, \ldots, X^{(L/2)}}_{\text{encoder (save skips)}} \xrightarrow{\text{mid Block}} \underbrace{\tilde{X}^{(L/2)}, \ldots, \tilde{X}^{(L)}}_{\text{decoder (with skips)}},$$

where each `Block` is a standard MHA +MLP transformer block with LayerNorm, and decoder blocks attend over skip connections. A final LayerNorm yields $X^{\text{out}} \in \mathbb{R}^{B \times (N_{\text{img}} + N_{\text{cnt}} + \text{extras}) \times d}$.

We then drop the auxiliary tokens and split modalities:

$$X_{\text{out}}^{\text{img}} = X^{\text{out}}[:, \text{img range}, :], \qquad X_{\text{out}}^{\text{cnt}} = X^{\text{out}}[:, \text{cnt range}, :].$$

**Count decoder.** Count patches are decoded back to a vector via a small MLP head with nonnegativity enforced by `Softplus`:

$$\hat{x}_0 = \text{Softplus}\Big(\text{MLP}\big(\text{flatten}(X_{\text{out}}^{\text{cnt}})\big)\Big) \in \mathbb{R}^{B \times D_c}.$$

This parameterizes $q_\theta(\cdot \mid x_t^{\text{cnt}}, I_t, t, \varepsilon, y)$ for the distributional loss and the reverse count-bridge step.

### F.2.3 TRAINING OBJECTIVE AND USAGE

We train the model to predict the distribution of $X_0$ (counts) given multimodal context under the bridge $(X_t)$:

$$\mathcal{L}(\theta) = -\mathbb{E}_{t,(X_0,X_t)}\Big[ S_\rho\big(q_\theta(\cdot \mid X_t^{\text{cnt}}, I_t, t, \varepsilon, y), X_0^{\text{cnt}}\big) \Big],$$

using the energy score $S_\rho$ with $\rho(x,x') = \|x-x'\|_2^\beta$ ($\beta \in (0,2)$) and the standard unbiased $U$-statistic estimator with $m$ samples from $q_\theta$. Time and noise tokens implement the distributional diffusion conditioning; label tokens (if present) enable class-conditional modeling. During reverse sampling we draw $\tilde{X}_0 \sim q_\theta(\cdot \mid x_t^{\text{cnt}}, I_t, t, \varepsilon, y)$ and update $x_{t-\Delta}$ using the exact binomial–hypergeometric count-bridge kernel (Prop. 3.1).

### F.2.4 IMPLEMENTATION SPECIFICS

- **Patchification.** `PatchEmbed` uses patch size $P$ on $I_t$ (channels $C$), producing $N_{\text{img}} = (H/P)(W/P)$ tokens of width $d$. `CountPatchEmbedding` projects $D_c$-dimensional counts to $N_{\text{cnt}}$ tokens of width $d$ with learned positional embeddings.

- **Auxiliary tokens.** Time token: `timestep_embedding` followed by a linear or MLP projector (`time_dim` controls concatenated components); noise token: linear to $d$ then a 2-layer SiLU MLP; label token: lookup embedding if used. All tokens share a single learned positional table.

- **Backbone.** Depth $L$ with $L/2$ encoder and $L/2$ decoder blocks; each block uses $d$-dimensional embeddings, $h$ heads, MLP ratio $r$, LayerNorm, and (optionally) gradient checkpointing. Decoder blocks accept the matching encoder skip.

- **Heads.** Count head: 2-layer GELU MLP over the concatenated count tokens, ending with `Softplus`. Image head exists but is ignored for the loss.

- **No weight decay.** We exclude token positional tables and count-positional embeddings from weight decay, following ViT practice.

### F.2.5 OPTIMIZATION AND HYPERPARAMETERS

We use Adam, learning rate $\{2\times10^{-4}$, cosine warmup for 100 steps, EMA with 0.999, batch size 128, gradient clipping at 1.0. See configs for exact architecture specification.

### F.2.6 SAMPLING

At test time we follow Alg. 3 using the aggregates to ensure our sampled $\hat{x}_0$ exactly match the target sum at each intermediate time. We then apply the exact binomial–hypergeometric reverse kernel (Prop. 3.1) to obtain $x_{t_{k-1}}$.

### F.3 ADDITIONAL RESULTS

**Comparison with reference-based methods** Count Bridges and STDeconvolve are both reference-free methods: that is, they require only aggregate level data, and do not need a reference dataset of unit-level measurements. Many deconvolution methods, including RCTD (Cable et al., 2022) require a single-cell (non-spatial) reference dataset. These methods benefit from unit-level observations, and

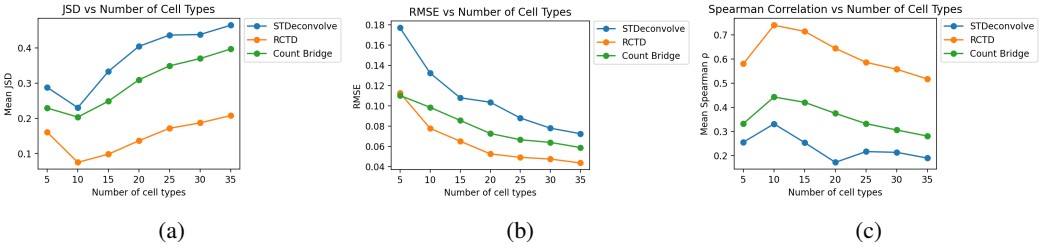

Figure 6: Performance of RCTD, STDeconvolve and Count Bridge on MERFISH deconvolution across number of cell types using (a) Jensen Shannon Divergence (JSD), (B) RMSE and (C) Spearman Correlation

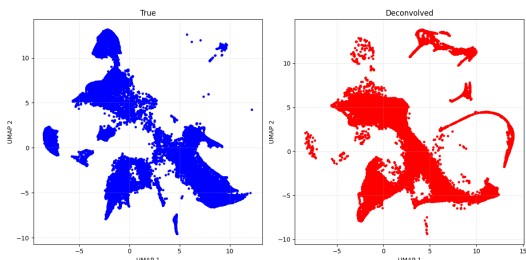

Figure 7: UMAP of true vs deconvolved cell profiles using the Count Bridge.

as such solve a more constrained problem – but require additional reference data which may not be available in all settings.

Using the MERFISH benchmarking setup we also evaluate RCTD, and tabulate the results in Tab. 14. For evaluation, we use Jensen shannon Divergence (JSD) and RMSE metrics as described in Li et al. (2023). We find that Count Bridges perform similarly to RCTD (with a higher JSD but lower RMSE), despite the fact that Count Bridges do not have access to a reference dataset.

| Method | JSD | RMSE | Spearman |
|---|---|---|---|
| STDeconvolve | 0.288 | 0.177 | 0.255 |
| RCTD | 0.161 | 0.113 | 0.580 |
| Count Bridge | 0.229 | 0.110 | 0.332 |

Table 14: Cell-type deconvolution error for spatial transcriptomic data. Note that RCTD requires a single-cell reference dataset for deconvolution.

In Fig. 6, we show the performance of spatial deconvolution methods across varying numbers of cell types. These results evaluate only the recovery of cell type proportions, and do not evaluate full count profiles. Note that RCTD has access to the single-cell level reference data, while STDeconvolve and Count Bridges are fit entirely using aggregate-level data and do not have access to single cell counts.

**Inspection of unit-level data generated by Count Bridges**    In the previous section, we have shown through quantitative metrics that Count Bridges outperform alternatives for reconstructing unit-level gene expression vectors from spot-level aggregates. We next aim to evaluate the extent to which reconstructed gene expression profiles are biologically meaningful. We do this by performing conventional single-cell transcriptomic analysis on the synthetic unit-level expression vectors generated by Count Bridges.

First, we plot the UMAP of the raw expression profiles of the true and generated data in Fig. 7. In this UMAP, we see that the true and deconvolved synthetic data are largely mixed, suggesting that the generated counts are realistic and match the distribution of the true counts.

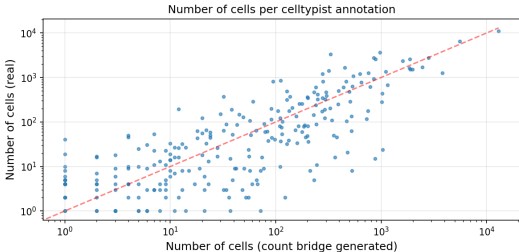

Figure 8: Abundance of putative cell types (automatically annotated by celltypist) in synthetic unit-level data generated by Count Bridges through deconvolution, vs. abundance of putative cell types in true unit-level data.

Next, we perform cell type annotation for the generated counts. We preprocess by normalizing counts to $10^4$ per cell (row-normalizing), followed by a log-transform, then annotate cell types using celltypist (Domínguez Conde et al., 2022). From this process, we identify 268 putative cell types in the synthetic unit-level data. The same pipeline, when applied to the real single-cell level measurements, identifies 278 putative cell types. And as shown in Figure 8, the cell type abundances inferred by Count Bridges (without access to unit level data) closely align with the cell type abundances observed in the true unit level data.

**Deconvolution of a 10X Visium dataset**   We next evaluate the deconvolution of spots in a real world 10X Visium dataset profiling the mouse brain. As ground truth is unavailable in this setting, we validate model predictions by assessing the extent to which the synthetic deconvolved data reflects known biology.

We use the 10X Visium fluorescence dataset distributed by Palla et al. (2022), which profiles a coronal section of a mouse brain. To demonstrate the generalization capabilities of Count Bridges, we apply the model trained on MERFISH data directly to this Visium dataset without retraining.

In order to correct batch effects and align dimensionality between datasets, we employ a moment-matching procedure. For each gene in the MERFISH data, we compute the mean and variance of expression and identify the gene in the 10X Visium data that most closely matches these moments. We then map the 10X Visium count matrices to the MERFISH feature space by subsetting to these matched genes.

We deconvolve the 10X Visium data using Count Bridges with a spot-level mean constraint. To evaluate prediction quality, we use a standard single-cell analysis pipeline (as described above) and determine putative cell type annotations using Celltypist (Domínguez Conde et al., 2022). Celltypist identifies 146 putative cell types, suggesting that the synthetic unit-level data recapitulates a significant degree of cell-to-cell variation. Furthermore, the recovered cell types are biologically consistent: the most abundant identified cell type is the oligodendrocyte, which matches the most abundant annotation in the MERFISH mouse brain dataset described above.

