# OpenReview forum: "Count Bridges enable Modeling and Deconvolving Transcriptomic Data"
_ICLR.cc/2026/Conference — ICLR 2026 Poster_

### Official Review · Reviewer_TMxD · 2025-10-27

**Soundness:** 3
**Presentation:** 2
**Contribution:** 3
**Rating:** 4
**Confidence:** 2

**Summary:**

The paper introduces Count Bridges, a generative framework for modeling and deconvolving integer-valued count data. The core of Count Bridges is a stochastic bridge process on the integers, defined by a Poisson birth-death mechanism. The model is trained using a distributional scoring loss, which is necessary to respect the discrete geometry of the data, as opposed to simpler point estimates used in continuous diffusion. For deconvolution, the authors extend this framework via an Expectation-Maximization (EM) algorithm that treats unit-level counts as latent variables: the E-step uses a projection-guided sampling process to impute unit-level data consistent with an observed aggregate sum, while the M-step trains the model by evaluating its performance at the aggregate level. The authors demonstrate the utility of their method on two large-scale biological applications: predicting single-cell gene expression from DNA sequence (outperforming a fine-tuned Enformer model) and deconvolving both bulk RNA-seq and spatial transcriptomic spots into single-cell profiles, benchmarking against methods like CIBERSORTx and STDeconvolve.

**Strengths:**

Mathematically, I find the paper very sound and compelling. A lot of papers have been released that train diffusion-type models on categorical data, but little research has extended these concepts to counts, which are the main type of modality in transcriptomics. The method appears correct and elegant, and potentially serves as a relevant first step for follow-up research in the field. I also really appreciate the connection between the model and the EM algorithm, which I find very original and effective to guide generation towards a certain aggregated profile.

**Weaknesses:**

Overall, my main points of criticism are in the readability and experimental evidence. I think the paper could be slightly more rigorous in presenting some concepts, and the biological experiments are not as comprehensive as they could be. Nonetheless, I hold a good opinion about the paper, and I am happy to discuss it during the rebuttal phase.

**Text.**

- L126-136: I recommend being a bit more thorough in defining the terms. For example, define $A$ as a covariance and not *noise*, make the sentence in L132-135 more clear (what is the subject?).

- I would also define the concept of ±. When inspecting the math, it is clear that it represents the addition and removal of counts following the Poisson process, where both adding and removing counts have their own time-resolved rate parameter. However, you can make it explicit from the text. Specifically, the introduction to the model is a bit abrupt and, to me, could benefit from a bit more natural language. I would explain that the counterpart to the continuous formulation of bridges is a birth/death-based Poisson process, with subsequent introduction to the individual terms. This could potentially improve the flow.

- A similar abrupt introduction is the connection with Schrödinger Bridges. The $\kappa$ parameter and $\pi$ notation for the process are previously undefined. I believe this breaks the flow of the read. Also, the concept of iteration, does it refer to an iterative proportional fitting? I find this connection quite unclear.

- For the distributional scoring loss, I recommend potentially using another symbol than $w$, as it is used to the success probability of $N_t$ before. A similar concept holds for $A$, it is used both as a covariance for the diffusion term and as an aggregation function. Also $\Pi$ is interchangeably used as the correcting projection for endpoint $x_0$ and the count bridge joint defined in 175. I personally feel all these aspects hinder the ease of reading a little bit.

- Typo L 159: $(X_s)_{t\in[0,1]}$, I think $t$ should be replaced by $s$.

- L258: Is the notation $A(\textbf{X}_{g0})$ correct? $A$ is defined as an aggregation, but here it is applied to a single entry.

**Content.**

- It's not clear to me how the bulk deconvolution task works. How do you match single cells to bulk profiles? Do you build a synthetic dataset? How do you match endpoints to train the bridge? I feel that all these details should be briefly introduced in the main; otherwise, the results are hard to interpret. Also, deconvolution is a very established approach with other methods like MUSiC [1]. I am not suggesting an exhaustive benchmark, but maybe a more comprehensive comparison would be appropriate.

- In general, I feel the experiment presentation is a bit disorganized. The metrics are not introduced, the captions are very short, and the benchmarks are very restricted. It would have been interesting to see some predictions of important genes performed by the model on the spatial slides, or to plot some 2D embeddings to validate that the generated cells are realistic.

[1] Wang, X., et al. (2019). MuSiC: bulk tissue cell type deconvolution with multi-subject single-cell expression references. Nature Communications.

**Questions:**

- What type of metric $\rho$ do you use in the count space?

- How does the experiment for RNA prediction from DNA work? What I understand is that, for every base pair on the DNA context, you condition noise-based generation of counts with the genomic context obtained via Enformer. Is it true? How do you run Enformer then, and how are the two settings comparable?

- You mention that you are using images to condition the bridge parameterization in the spot experiment. Did you try ablating the conditioning on the images and evaluating the performance after this? I find it unexpected that including image information serves as a performance boost for this task.

---

> ### Author Response · Authors · 2025-11-21
>
> We thank the reviewer for their thoughtful assessment. We are glad you found the framework compelling. We appreciate the detailed feedback on readability and notation, which we have used to significantly tighten the manuscript. We have additionally deepened our analysis in the biological experiments.
>
> > Deconvolution is a very established approach with other methods like MUSiC...
>
> We thank the reviewer for this suggestion. We have added MuSiC (Wang et al., 2019) as a baseline for our bulk deconvolution task. Count Bridges significantly outperform MuSiC on both RMSE and JSD metrics.
>
> > It's not clear to me how the bulk deconvolution task works...
>
> We apologize for the lack of clarity here. In the current manuscript we set aside the issues with truly matching bulk and single cell data for our bulk experiment. We think this is an interesting and important area for future work, but here we are most interested in presenting a proof-of-concept. Since ground-truth single-cell data is not available for physical bulk samples, we start with nucleotide-level single cell data. In our case this data comes from 1000 pateints.
>
> We train on cell-level single cell data. We additionally train the learned projection module $\Pi_\psi$. To train the model, on 10% of training batches, we make the target aggregate available and use the projection to project the cells toward that target. We then just use the cell-level energy score objective on these projected cells. This enables us to learn a projection operator that better respects the conditional distribution we are trying to approximate. We have clarified this further in the main text.
>
> For evaluation we hold out 10% of patients to form the evaluation set. We synthetically bulk these patients and then deconvolve them at the nucleotide level using our model by conditioning on the patient level mean. Since we have the ground truth data we can then compare our deconvolved samples to the ground truth. We have clarified this procedure in Section 6.2.
>
> > It would have been interesting... to plot some 2D embeddings to validate that the generated cells are realistic.
>
> We agree. We have added visualizations of the recovered single-cell profiles to the Appendix. These plots show that the single cells generated by Count Bridge recapitulate the manifold structure of the ground truth data, validating that the model learns biological structure.
>
> > In general, I feel the experiment presentation is a bit disorganized...
>
> We apologize for these presentation issues. We have added a clearer introduction of the metrics we use to evaluate at the start of the applications section. We have also significantly edited the applications to make as clear as possible how we are training models and what their capabilities are.
>
>
> > What type of metric $\rho$ do you use in the count space?
>
> We use the squared Euclidean distance: $\rho(x, y) = ||x - y||_2^2$. This corresponds to the Energy Distance, which is a strictly proper scoring rule for distributions with finite first moments. We have clarified this in the main text.
>
> > How does the experiment for RNA prediction from DNA work?
>
> This is how we train the Count Bridge in this setting. The base Enformer model is fine-tuned for bulk prediction on the same data using the same prediction head architecture as we use for the Count Bridges. We fine tune on the bulk objective. The two models are only comparable on the bulk task since our fine-tuned Enformer model (like the original Enformer model) is only capable of bulk predictions.
>
> > Did you try ablating conditioning on images?
>
> For pure deconvolution, our framework requires that we condition on unit-level side information. As such, the images (or some other cell-level information) are required. In the spatial transcriptomic setting, it is unclear what cell-level information is available aside from the images themselves. We appreciate that from a biology perspective the DAPI image is not particularly information rich, but it does at minimum contain key information about cell size which the model can use for deconvolution.
>
> > Symbol Collisions
>
> We apologize for the symbol collisions and abrupt definitions. We have performed a thorough editorial pass to address your "Text" list:
> 1.  Notation: We have renamed the distributional scoring metric parameter to avoid conflict with probability $p$. We now explicitly distinguish between $\Sigma$ (covariance) and our aggregation operator.
> 2.  Birth-Death Intro: We have rewritten the introduction of the Poisson process (Section 3.1) to be less abrupt, explicitly defining the rate parameters $\lambda_\pm$ and the jump intensity before presenting the kernel.
> 3.  Schrödinger Bridges: We have smoothed the transition to the theoretical section, defining $\kappa$ earlier and clarifying generally significantly clarifying this section which was very dense and difficult to parse.

---

> > ### Comment · Reviewer_TMxD · 2025-11-23
> >
> > I commend the authors for their rebuttals. The paper is now improved in clarity and rigor.
> >
> > Positives:
> > * Experiments and notations are clearer.
> > * Metrics are more digestible.
> >
> > Additional points:
> > * There is a slight overuse of $\rho$ (semimetric and weight ratio across generation time). I would make the distinction clearer.
> > * In the experiments, the results in Fig.7 are ok but not great. It seems the model is struggling with the underrepresentation of some rare cell types.
> > * I still find section 6.2 a bit convoluted, but I do acknowledge that this may be a limit in my prior knowledge of the DNA models. If I understand correctly, you produce a series of expression candidates for a certain nucleotide and perform the following comparisons:
> >
> > 1. Tab. 1: Compare the empirical average expression with the actual one in the data.
> > 2. Tab. 2: Compare to which the generated gene expression approximates real single-cell expression compared to the bulk average.
> >
> > Then, how is the bulk average compared to single-cell expression via distribution metrics? Wasn't there a somewhat stronger baseline (e.g., a Flow Matching model conditioned on bulk information to generate decovolved single cell profiles)? What's the variability of the metrics? Are the results statistically significant across generations? For clarity, I am not necessarily asking to run additional methods, just a general clarification on how the comparisons are structured.
> >
> > * Now that I better understand the comparison with Enformer I wonder: Is it fair to compare Enformer's regression task with your generation approach, when the latter is guided by the number of bulk counts it has to generate?
> >
> > * I recommend adding a few words about MUSiC to the main text.
> >
> > In general, I am leaning towards keeping my score or increasing it to a low-confidence 6, as this paper would be great with some restructuring for better clarity and experimental section fine-tuning. Looking forward to a follow-up discussion with the authors. Thank you!

---

> > > ### Author Response · Authors · 2025-11-24
> > >
> > > ## Discussion of nucleotide-level deconvolution
> > >
> > > We thank the reviewer for their interest in the nucleotide-level sequence model. Modelling and deconvolving single-cell transcriptomes at the nucleotide level is an ambitious long-term goal. We believe the evaluations across Tables 1-3 demonstrate that Count Bridges serve as a meaningful proof-of-concept toward this broader project since they outperform reasonable baselines at both the bulk nucleotide level (compared to Enformer) and deconvolution tasks (comparing to state-of-the-art approaches for bulk deconvolution). **We have tried to further clarify these points in the application section.**
> > >
> > > Specifically, Count bridges:
> > >
> > > - Without conditioning on bulk profiles:
> > >   - Are comparable (in fact better than) fine-tuned Enformer for bulk prediction and for cell-type-specific mean prediction, unconditional on any bulk profile.
> > >
> > > - While conditioning on bulk profiles:
> > >   - Significantly outperform simple baselines like the bulk mean on distributional prediction metrics.
> > >   - Outperform existing approaches for cell type deconvolution while providing actual full count profiles at single nucleotide resolution.
> > >
> > > We believe these results are not a result of the quantitative metrics. In App. E., Fig. 5, we present the UMAP (colored by cell type) for the true and devolved cells for the held-out patients. Note that this UMAP is computed from nucleotide-level count matrices. It is clear that Count Bridges are reasonably performant at recapitulating the distribution of real cells.
> > >
> > > We will now respond to each question in a bit more detail:
> > >
> > > > Is it fair to compare Enformer's regression task with your generation approach, when the latter is guided by the number of bulk counts it has to generate?
> > >
> > > For the comparison with Enformer (bulk prediction), we do not condition on the bulk counts; we run unconditional inference so that we can easily compare the two models. If we conditioned on the counts in the bulk comparison, for example, the Count Bridge would achieve zero error. We believe it is interesting that the Enformer approach is worse than our approach; likely it is due to (1) the base Enformer model not handling the variability from modeling both cell-type conditional and unconditional profiles, and (2) the Enformer model being trained across patients, which additionally increases the variance of the gradients in the MSE.
> > >
> > > > How is the bulk average compared to single-cell expression via distribution metrics?
> > >
> > > We compute the distributional metrics, treating each cell as though it had the same profile, the mean profile. Especially in high-dimensional settings, this is a reasonably strong baseline.
> > >
> > > > Wasn't there a somewhat stronger baseline?
> > >
> > > Given the results in our manuscript indicating flow matching lags significantly CB in similar settings, we believe this would in fact be a relatively weak baseline, especially compared to the cell-type proportion deconvolution baselines, which are existing, state-of-the-art methods for deconvolution. As these models are resource-intensive to train, we chose not to implement such a baseline. If the reviewer thinks this addition would significantly strengthen the paper, we can prepare this baseline. However, it may not be trained to parity with our main model during the discussion period, due to time and resource constraints.
> > >
> > > > In the experiments, the results in Fig.7 are ok but not great.
> > >
> > > Yes, both models likely struggle with this cell type since it is simply not seen frequently in training and since its transcriptomic profile is quite distinct. We believe that continued training would likely improve the performance.
> > >
> > > > [A]dd a few words about MUSiC to the main text.
> > >
> > > We have added a note on MuSiC in the main text; we apologize for leaving this out previously.
> > >
> > > ## Other points
> > >
> > > > What's the variability of the metrics?
> > >
> > > Our evaluations use a large number of cells, and these average over these metrics. So the variance between inference runs is quite small; for example, we present the replicates for spot deconvolution in our response to K7ei:
> > >
> > > | Comparison            | Energy Distance | Wasserstein Distance | MMD (RBF) |
> > > |-----------------------|------------------|-----------------------|-----------|
> > > | Pred 1 vs True        | 8.891            | 0.017                 | 0.203     |
> > > | Pred 1 vs Spot Mean   | 42.903           | 0.034                 | 0.419     |
> > > | Pred 2 vs True        | 8.919            | 0.017                 | 0.203     |
> > > | Pred 2 vs Spot Mean   | 42.767           | 0.034                 | 0.418     |
> > > | Pred 3 vs True        | 8.898            | 0.017                 | 0.204     |
> > > | Pred 3 vs Spot Mean   | 42.856           | 0.034                 | 0.419     |
> > > | True vs Spot Mean     | 41.717           | 0.030                 | 0.409     |
> > >
> > > The results across runs are extremely similar qualitatively.
> > >
> > > > There is a slight overuse of $\rho$
> > >
> > > We have swapped in $r$ for the ratio; we thank the reviewer for pointing this out.

---

> > > > ### Comment · Reviewer_TMxD · 2025-11-25
> > > >
> > > > Dear authors,
> > > >
> > > > Thanks for your clarifications. I think the text is now much clearer regarding the single-nucleotide experiment.
> > > >
> > > > I still like the paper and feel like moving to a 6 (borderline accept) as most of my points have been addressed. The main reason for not going higher is that I see a bit of a discrepancy between the quality of the results section and the theory section. The latter I really liked, the former I believe could use some bettering (e.g., adding some error bars to tables, consistency in the breakdown into paragraphs).
> > > >
> > > > I hope the authors found the rebuttal useful and wish them the best of luck with the remainder of the revisions!

---

> ### Author Response · Authors · 2025-11-25
>
> We thank the reviewer for their continued attention to our paper and for their suggestions, which we agree have significantly improved the manuscript. We have made additional revisions:
>
> 1. We have now added standard errors sampled over inference runs to Tables 1-5 for the main applications. We comment on this at the beginning of the applications section.
> 2. We agree that the presentation of the applications was a weakness of the manuscript at the time of submission. We believe that Section 6.2 has benefited tremendously from our back-and-forth, and we have now carried this structure through our Section 6.3 to present a more unified and clearer applications section.
>
> We very much appreciate the reviewer's engagement, time spent helping us improve the manuscript, and kind words about our theory section. We would like to bring the applications section to parity with the theory section. If there are specific remaining adjustments (formatting, clarifications, or additional results or baselines) that would help the reviewer advocate more strongly for the paper, we are happy to implement them!

---

> > ### Comment · Reviewer_TMxD · 2025-11-25
> >
> > Dear authors,
> >
> > Thanks for your additional revisions. It might sound nitpicking, but I would make sure that the number of decimal digits is consistent within the tables, as this does not seem to undermine the significance of the improvement.
> >
> > At the moment, I do think the paper is in better shape, and I appreciate the careful textual edits. I reread the sections, and I find everything better structured and understandable. I will increase my score to 8.
> >
> > Once again, I wish you good luck with the rest of the rebuttals and final decisions!

---

### Official Review · Reviewer_F1ZA · 2025-10-29

**Soundness:** 3
**Presentation:** 2
**Contribution:** 3
**Rating:** 6
**Confidence:** 2

**Summary:**

This paper introduces Count Bridges, a stochastic bridge framework for modeling and deconvolving integer-valued count data, particularly motivated by RNA-seq and spatial transcriptomics. The proposed model is based on a Poisson birth-death process on the integers, yielding closed-form conditionals that enable exact sampling and tractable training. The framework supports both generative modeling of count distributions and aggregate-to-unit deconvolution via an Expectation Maximization procedure treating unit-level counts as latent variables.
The authors demonstrate applications to synthetic benchmarks and two biological tasks: deconvolving bulk RNA-seq data and inferring single-cell profiles from spatial transcriptomic spots. The method achieves strong empirical performance, outperforming baselines such as flow matching, discrete flow matching, CIBERSORTx, and STDeconvolve.

**Strengths:**

- Novelty and conceptual clarity: The introduction of a bridge-based model tailored to integer-valued data is original and addresses an underexplored area between diffusion-style generative models and biological count modeling. The stochastic birth–death formulation provides a principled way to interpolate between discrete distributions.

- Mathematical rigor: The theoretical development is clear and well-supported, including proofs and connections to Schrödinger bridges and optimal transport. The paper demonstrates careful thought in defining the discrete bridge process and the distributional training objective.

- Relevance to biology: Extending generative modeling to integer-valued transcriptomic data and demonstrating utility in deconvolution tasks is timely and important for large-scale single-cell and spatial genomics.

- Empirical results: The experiments convincingly show both generative and deconvolution performance, with clear benchmarks and biological relevance. The model’s ability to outperform reference-free and even reference-based baselines is notable.

**Weaknesses:**

1. Presentation and flow

While mathematically solid, the readability could be improved. Some definitions (e.g., lines 130–140 in the original text) appear abruptly, and notation (such as $A(X_0), Π𝜓$) could be introduced more gently. Reordering to first build intuition before formalism might help readers from outside the generative modeling community.

2. Clarity of deconvolution setup

The biological deconvolution experiments are appealing but could use more methodological clarity. For example:

- How exactly are unit-level latents initialized or sampled during the EM procedure?

- How are aggregates simulated from real datasets (e.g., in MERFISH) and how sensitive are results to the aggregation scheme?

- What metrics are used to evaluate count-level accuracy beyond JSD and RMSE?

3. Comparative baselines

While the model outperforms selected methods, the comparison to continuous diffusion models (e.g., Gaussian-based or Poisson-approximated variants) could be expanded, since this would better position Count Bridges as a discrete alternative rather than a purely new category.

4. Experimental depth

The evaluation focuses mainly on summary-level results. Including visualizations of reconstructed single-cell profiles, or ablations on the projection module $Π𝜓$, would strengthen the empirical case.

5. Theoretical framing

The connection between Count Bridges and Schrödinger bridges is very interesting but only briefly touched upon. Expanding this connection, perhaps through an interpretation in terms of entropy-regularized optimal transport, would make the theoretical section more accessible and conceptually richer.

**Questions:**

See weaknesses.

---

> ### Author Response · Authors · 2025-11-21
>
> We thank the reviewer for finding our work to be "novel," "mathematically rigorous," and "timely" for biological applications. We appreciate the constructive feedback on readability and have used it to significantly improve the manuscript.
>
>
> > The comparison to continuous diffusion models could be expanded...
>
> Most standard diffusion models are designed for generation (Noise $\to$ Data) rather than bridging two arbitrary distributions (Source $\to$ Target), which is the focus of our deconvolution tasks. To assess this we have added a baseline against a standard Gaussian diffusion model (Song, 2020) and masked denoising discrete diffusion model (Shi, 2024) on a new task where the goal is simply to model two moons as a target, without needing to start at any particular initial distribution. We use the same Count Bridge result as in the 8-Gaussians to 2-Moons variant, so the noise is 8-Gaussians. Count Bridges remain superior in this regime.
>
>
> **Discrete Moons Results: Eight Gaussians → Two Moons**
>
> | Method              | MMD                      | \(W_2\)                  | Energy                     |
> |---------------------|--------------------------|---------------------------|-----------------------------|
> | Count Bridge    | **0.0044 ± 0.0018**      | **0.0052 ± 0.0007**       | **0.0098 ± 0.0029**         |
> | Gaussian Diffusion  | 0.024 ± 0.009            | 0.017 ± 0.006             | 0.118 ± 0.064               |
> | Discrete Diffusion  | 0.017 ± 0.010            | 0.013 ± 0.006             | 0.072 ± 0.055               |
>
> > The evaluation focuses mainly on summary-level results
>
> We have now included a number of single-cell analyses to the appendix including visualizations of the single cell profiles. In Section F.3, we show that when standard single cell analysis pipelines are applied to synthetic unit-level gene expression generated by Count Bridges, the identified cell types and cell type distributions closely recapitulate those seen in the true unit level data.
>
>
> > Presentation and flow
>
> We apologize for this lack of clarity. We have generally restructured the core methods section (Section 3) to hopefully be much more straightforward to follow.
>
> > How exactly are unit-level latents initialized...
>
> The unit-level latents are sampled purely from the Count Bridge using guidance to ensure they match the aggregate mean. This means that in the first epoch, we are using a completely untrained Count Bridge to generate the unit-level latents that we then train on. This nonetheless leads to very high-quality results.
>
> > Clarity of deconvolution setup
>
> We apologize for this lack of clarity, we have expanded our explanation of these experiments in the main text and added a deeper analysis of our results in the appendices.
>
> > The connection between Count Bridges and Schrödinger bridges...
>
> We have rewritten the theoretical discussion to make the connection to Entropy-Regularized OT explicit.
>
> In the original text, we connected our parameter $\kappa$ to the diffusion coefficient $\sigma$ in Gaussian SBs. We have now clarified that Count Bridges solves a Schrodinger bridge problem, and have generally cleaned up the exposition here. We believe this clearer framing significantly enriches the theoretical contribution.
>
> > How are aggregates simulated from real datasets...
>
> We discuss how we simulate the aggregates for the MERFISH experiment in Append F.1:
>
> To simulate a Visium-style spatial transcriptomics dataset, we aggregated the single-cell MERFISH data. A grid of spots was defined with a center-to-center distance of $100 \mu m$ and a spot radius of $55\mu m$. The gene expression profile for each simulated spot was then generated by summing the transcript counts of all identified cells whose nuclei fell within the circular bounds of that spot.
>
> These sizes are chosen to match the standard Visium spacing. The results will be sensitive to the aggregation scheme to the extent we aggregate more cells (see the discussion of how learnability scales in the number of cells in Sec. 6 and in particular Fig. 4). Our aggregation scheme here is conservative in this regard: the median group has 15-25 cells whereas in actual Visium data we observe 10-15 cells per spot.
>
> > What metrics are used to evaluate count-level accuracy...
>
> The JSD and RMSE are metrics we use to assess the accuracy of the clusters we infer from our estimated single-cell counts. To estimate the count accuracy we additionally computed the MMD, $W_2$ and Energy distance as we did in Table 4 (Table 5 in the updated manuscript). We have added these evaluations to the bulk data as well in Table 2.
>
> **References**
>
> Shi, J., Han, K., Wang, Z., Doucet, A. and Titsias, M., 2024. Simplified and generalized masked diffusion for discrete data. Advances in neural information processing systems, 37, pp.103131-103167.
>
> Song, J., Meng, C. and Ermon, S., 2020. Denoising diffusion implicit models. arXiv preprint arXiv:2010.02502.

---

### Official Review · Reviewer_xPQ5 · 2025-10-30

**Soundness:** 3
**Presentation:** 2
**Contribution:** 3
**Rating:** 6
**Confidence:** 2

**Summary:**

This paper introduces Count Bridges, a novel stochastic bridge framework for modeling integer-valued count data, extending diffusion-style generative modeling to discrete domains such as single-cell RNA-seq. The method formulates a birth–death bridge process that connects two count distributions through continuous-time dynamics, allowing for closed-form conditional laws and exact likelihood estimation. The authors further propose an EM-style learning scheme that enables deconvolution of aggregated transcriptomic data—e.g., recovering single-cell distributions from bulk or spatial measurements. Experiments on synthetic and real datasets show that Count Bridges outperform existing discrete flow and diffusion models in both modeling fidelity and biological interpretability.

**Strengths:**

Count Bridges constructs a birth–death bridge with a closed-form conditional distribution, enabling precise likelihood estimation for integer data. The EM-style extension for transcriptomics holds practical value: it provides an actionable workflow for deconvoluting single-cell distributions from aggregated observations derived from bulk or spatial sequencing. Experimentally, the method demonstrates robust improvements over relevant discrete baselines across synthetic and real-world data tasks presented in the paper. The paper is thorough in both mathematical derivations and implementation details, ensuring the work combines theoretical rigor with practical applicability.

**Weaknesses:**

- While being compared against CFM, DFM, and some biological baselines, a direct comparison with other recent count - specific or general discrete diffusion models (beyond Blackout Diffusion) on the proposed tasks could provide a more complete picture.
- The computational complexity of Count Bridges versus discrete diffusion or flow models is not reported; scalability to large transcriptomic datasets is unclear.
- While deconvolution results are promising, the paper provides limited biological case studies (e.g., linking recovered cell states to known pathways).

**Questions:**

- Many single-cell datasets are zero-inflated. Does Count Bridges handle this naturally via the birth–death process, or is special preprocessing required?
- How does runtime and memory scale with dimensionality (number of genes) and group size in aggregate deconvolution?
- Can the learned bridge latent representations be linked to biological factors (e.g., cell states, differentiation trajectories)?

---

> ### Author Response · Authors · 2025-11-21
>
> We thank the reviewer for the constructive feedback and for recognizing the theoretical rigor and practical value of Count Bridges. We appreciate the opportunity to clarify the scalability and biological interpretability of our method.
>
> > The computational complexity of Count Bridges...
>
> Sampling the bridge just requires sampling the Bessel, Binomial and Hypergeometric draws. The numpy/cupy implementations of the Binomail and Hypergeometric are both $O(1)$ in time and memory. We write a custom CUDA kernel implementing Devroye (2001) which is also $O(1)$ in time and memory per draw. So the computational complexity is no more than the complexity of Gaussian samples. Since discrete flow matching requires fully expanding the categorical it actually requires $O(K)$ time and memory the Count Bridge enjoys a distinct advantage, particularly when counts can become large.
>
> Additionally both of our applications involve scaling Count Bridges to large transcritomic datasets involving over a million cells and the standard transcriptomic count distribution. We even model this at the nucleotide level, demonstrating extremely high dimensional capabilities.
>
> The key difference in computational complexity is actually at training time, where since we use a distributional objective our computational cost in both time and memory scales quadratically in $m$, the number of Monte Carlo samples we use to estimate the distributional loss. In the main text we use $m=32$ for the synthetic experiments and $m=6$ for the applications. This cost is only at training time, at inference it is $O(1)$.
>
> > How does runtime and memory scale...
>
> We have added an App. C that documents all of the projection and rounding algorithms we use for deconvolution along with time and (extra) memory complexity. These are the only algorithms that change the run time. All algorithms are linear in time in input size and effectively constant in memory, except our procedure for exactly matching the rescaled counts while rounding. This procedure has runtime $O(|G| \log |G|)$, so just a little worse than linear, and linear runtime in group size.
>
> > A direct comparison with general discrete diffusion models...
>
> All of the tasks we consider in the manuscript are about starting from a particular source distribution and going to a particular target, so diffusion models cannot be immediately used for these tasks (this is why we benchmark against flow matching).  To assess this we have added a baseline against a standard Gaussian diffusion model (Song, 2020) and masked denoising discrete diffusion model (Shi, 2024) on a new task where the goal is simply to model two moons as a target, without needing to start at any particular initial distribution. We use the same Count Bridge result as in the 8-Gaussians to 2-Moons variant, so the noise is 8-Gaussians. Count Bridges remain superior in this regime.
>
> **Discrete Moons Results: Eight Gaussians → Two Moons**
>
> | Method              | MMD                      | \(W_2\)                  | Energy                     |
> |---------------------|--------------------------|---------------------------|-----------------------------|
> | Count Bridge    | **0.0044 ± 0.0018**      | **0.0052 ± 0.0007**       | **0.0098 ± 0.0029**         |
> | Gaussian Diffusion  | 0.024 ± 0.009            | 0.017 ± 0.006             | 0.118 ± 0.064               |
> | Discrete Diffusion  | 0.017 ± 0.010            | 0.013 ± 0.006             | 0.072 ± 0.055               |
>
>
> > Many single-cell datasets are zero-inflated...
>
> No special processing is required to handle zero-inflated data, this is handled naturally via the count bridge. In fact the complexity of single cell data is a primary motivation for developing count bridges.
>
> > limited biological case studies...
> > link to biological factors?
>
> In the appendix (Section F.3) we have added a set of analyses exploring the recovered cell states in the spatial transcriptomic deconvolution problem and their biological significance. Notably, we find that nearly all cell types (as defined by CellTypist) present in the real unit-level data can be identified from spot-level data alone by using Count Bridges. Furthermore, the abundances of the cell types are recapitulated by the synthetic unit-level data generated by Count Bridges. This suggests that the latent $x_i$ learned by Count Bridges are biologically significant and reflect true cell states. In addition, we have added a validation on Visium data in response to Reviewer S8Cu.
>
> **References**
>
> Shi, J., Han, K., Wang, Z., Doucet, A. and Titsias, M., 2024. Simplified and generalized masked diffusion for discrete data. Advances in neural information processing systems, 37, pp.103131-103167.
>
> Song, J., Meng, C. and Ermon, S., 2020. Denoising diffusion implicit models. arXiv preprint arXiv:2010.02502.

---

### Official Review · Reviewer_S8Cu · 2025-10-31

**Soundness:** 2
**Presentation:** 3
**Contribution:** 3
**Rating:** 4
**Confidence:** 2

**Summary:**

This paper proposes Count Bridge, an integer-native diffusion framework for count data that replaces Gaussian additive noise with two Poisson processes (birth/death). This yields sequential ±1 jump dynamics that preserve integer structure. The denoiser is distributional and trained with a strictly proper energy score suited to count-space geometry. The authors derive closed-form intermediate conditionals (Binomial / Hypergeometric / Bessel components) and implement principled local-bridge sampling. They extend Count Bridge to deconvolution by treating unit-level X0 as latent and using a generalized EM scheme: a projection-guided diffusion E-step (the projection is applied at each reverse timestep to guide local-bridge sampling under the aggregate constraint) and an M-step that trains the model via an aggregate-level energy score loss. Empirical results on synthetic data and several real deconvolution tasks (single cell -nucelotide expression, bulk RNA data deconvolution, spatial spot deconvolution) show competitive performance versus baselines such as CFM and DFM or Enformer, CIBERSORTx, STDeconvolve.

**Strengths:**

1.	Introduces an integer-native diffusion process (birth–death Poisson kernel) that naturally preserves counts.
2.	Derives analytic intermediate conditionals (Binomial / Hypergeometric / Bessel), which support principled local-bridge sampling.
3.	Practical deconvolution pipeline: Gives a workable EM-style approach (projection-guided diffusion + aggregate-level loss) to infer unit-level counts from aggregate observations.
4.	Broad evaluation: Tests on synthetic and multiple real-world deconvolution tasks, with comparisons to relevant baselines.

**Weaknesses:**

1. Identifiability and aggregation scale. Pure aggregate supervision is intrinsically ill-posed; performance can degrade as group size increases or between-unit heterogeneity decreases. Please provide quantitative sensitivity analyses showing performance vs. group size and vs. within-group heterogeneity (e.g., varying variance of unit distributions), and explain the practical limits (aggregation scales)  where the proposed EM is reliable.
2. For the nucleotide-level gene expression modeling task, the authors compare to Count Bridge, which reportedly improves on the fin-tuned Enformer’s direct sequence to cell type-specific expression. I observe that the MSE for the 'plasma' cell type in Appendix E is several orders of magnitude higher than for other cell types. This is concerning and requires clarification.
3. Please include an ablation over projection and discretization choices (KL rescale + multinomial, rounding/min‑distance, learned Πψ, etc.) and report sensitivity of downstream metrics.

**Questions:**

1.	Enformer-like models are typically valued not just for prediction accuracy but as tools to study sequence-to-expression regulatory mechanisms (e.g., variant effect prediction, attribution maps, motif/TF signal localization). The manuscript reports that Count Bridge outperforms a fine-tuned Enformer on cell-type-specific expression prediction in single-cell data. A key question for me is whether Count Bridge can support the same kinds of downstream regulatory analyses that make it useful in genomics.
2.	In the spatial transcriptomic deconvolution task, I strongly encourage the authors to validate their spatial deconvolution results on genuine spatial transcriptomics data. Specifically, please evaluate the method on at least one sequencing-based platform (e.g., 10x Visium) rather than only on spots synthesized from MERFISH single-cell imaging.

The reviewer wrote the review. LLM was employed only to correct grammatical errors.

---

> ### Author Response · Authors · 2025-11-21
> **Response (I)**
>
> We thank the reviewer for their thoughtful feedback and for recognizing the novelty of our integer-native diffusion framework and the practicality of the deconvolution pipeline. We appreciate the opportunity to clarify the identifiability analysis and the scope of the biological applications.
>
> > Identifiability and aggregation scale.
>
> We thank the reviewer for this comment. We agree it is an extremely central question. We actually performed exactly this analysis in the original submission (Figure 4), but we apologize if its prominence or explanation was insufficient.
>
> In Fig. 4 we plot the $W_2$ distance between the true and inferred unit-level distribution across varying group sizes ($4,8,32,128$) and varying levels of heterogeneity. We parameterize the heterogeneity through a Dirichlet distribution with parameter $\alpha$ over the mixture components. By varying $\alpha$ we can vary the heterogeneity acrross groups. As we let $\alpha \rightarrow \infty$ every group is from the same distribution, and all variance in the observed group-level proportions will come from sampling (e.g., in small groups there is heterogeneity, but as $n\rightarrow\infty$ every group would have the same composition). We find that the more heterogeneous groups are easier to learn from at every $n$, and that as $n$ increases, it becomes exponentially harder to learn. We discuss this in detail in App. B.3.
>
> In general, we think this justifies moderate pure-deconvolution, in up to groups of size up to ~32. One important dimension here is group size heterogeneity. If we mix moderate/large groups with unit-level data that has the potential to let us learn bridges capable of leveraging group-level data while breaking the "curse" of aggregation.
>
> Additionally, it is worth noting that with enough unit-level data, we can just train a unit-level bridge and deconvolve groups of arbitrary size with reasonable results (as in our bulk analysis).
>
> > Please include an ablation over projection and discretization choices
>
> We have added this analysis to the appendix, alongside an appendix documenting precisely what our projection algorithms are. Our ablation includes three methods:
> 1. Rescale and round (failing to preserve expectation)
> 2. Rescale and randomized round (preserve expectation but will not exactly match)
> 3. Rescale and randomized exact match (our preferred method, match exact conditional sum on the integers after rescaling)
>
> We ran each of these in the EM algorithm for the $n=128, \alpha=1$ case. We find that there are small gains from the randomized exact matching across all three distributional metrics, but the randomized (non-exact) approahc is very close to equally performant. Since this method is so much simpler, this maybe justify using it in place of the exact method.
>
> **Table: Performance comparison of different rounding methods with standard errors**
>
> | Method        | MMD                     | W₂                      | EMD                  |
> |---------------|--------------------------|--------------------------|-----------------------|
> | `exact`       | **0.037 ± 0.005**        | **0.050 ± 0.008**        | **2.24 ± 0.58**       |
> | `randomized`  | 0.038 ± 0.000            | 0.051 ± 0.007            | 2.33 ± 0.43           |
> | `round`       | 0.038 ± 0.003            | 0.052 ± 0.007            | 2.32 ± 0.44           |
>
> > Validate their spatial deconvolution results on genuine spatial transcriptomics data...
>
> We agree that real-world validation is crucial. We have now applied our trained spatial model to deconvolve a 10x Visium dataset. While ground truth is unavailable for Visium (unlike the synthetic MERFISH spots).
>
> We deconvolve the 10X Visium data using Count Bridges with a spot-level mean constraint. To evaluate prediction quality, we use a standard single-cell analysis pipeline (as described above) and determine putative cell type annotations using Celltypist \citep{Dominguez-Conde2022-cg}. Celltypist identifies 146 putative cell types, suggesting that the synthetic unit-level data recapitulates a significant degree of cell-to-cell variation. Furthermore, the recovered cell types are biologically consistent: the most abundant identified cell type is the oligodendrocyte, which matches the most abundant annotation in the MERFISH mouse brain dataset described above.

---

> ### Author Response · Authors · 2025-11-21
> **Response (II)**
>
> > Enformer-like models...
>
> We agree that this is an exciting application of Enformer-like models, but careful analysis of gene regulatory mechanisms is beyond the scope of our work here. The primary goal of our work is provide a framework for deconvolution rather than bulk prediction alone. Count Bridges are complementary to Enformer, potentially enabling the identification of genomic regions which contribute most to single cell variability.
>
> > For the nucleotide-level...
>
> Thank you for catching this. The 'plasma' label refers to plasmacytoid dendritic cells (pDCs), which are extremely rare and transcriptionally distinct [1]. The high MSE arises because this cell type is characterized by extreme sparsity mixed with very specific high counts. Crucially, both Count Bridge and the fine-tuned Enformer baseline perform poorly here. The error is driven by the difficulty of the specific cell type distribution rather than a defect unique to Count Bridge. We have clarified this in the appendix.
>
> *References*
>
> 1. Hao, Yuhan, et al. "Integrated analysis of multimodal single-cell data." Cell 184.13 (2021): 3573-3587.

---

### Official Review · Reviewer_K7ei · 2025-10-31

**Soundness:** 4
**Presentation:** 3
**Contribution:** 4
**Rating:** 8
**Confidence:** 4

**Summary:**

This submission proposes Count Bridges, a novel approach to finding a probability path between end-marginal distributions on integer values. This is partly achieved by modeling what resembles the diffusion term in an SDE as a difference of independent Poisson processes.

Using an EM algorithm, the method is extended to the setting where the observations obtained from one of the marginals is an aggregate of samples, which then is used to solve important problems in biology. There are multiple real and synthetic data experiments in the paper.

**Strengths:**

This is one of the best papers I have read this year. The only reason I did not reward the submission with the highest presentation score is because some concepts were not explained/taken for granted (I discuss this in Weaknesses).

The originality and quality of the paper is top-class, especially considering the very clever use of the EM algorithm to solve the deconvolution problem. Furthermore, modeling probability bridges directly on the space of count data is of great importance in genomics and transcriptomics. Needless to say, Count Bridges can have a substantial impact on the field of computational biology *and* the method is novel, non-trivial and elegant.

Although the results are missing error bars and the quantitative analysis of the results could be more thorough, Count Bridges are evaluated on multiple important biological tasks, outperforming for example a fine-tuned version of the DeepMind produced Enformer.

Count Bridges will surely be an appreciated addition to the class of generative models by the ML community.

**Weaknesses:**

### Distinction to SBs
Why should Count Bridges not be considered as an instance of Schrödinger Bridges? SBs are not constrained to continuous measures, and there is no statement about why CBs are not SBs (there is instead a note on the connection between SBs and CBs in line 172).

I think this requires some clarity to avoid the risk of artificially distancing CBs from SBs in order to promote novelty.

### Clarity
Surprisingly, I was mainly concerned with the clarity in Sec. 3. I found that key concepts were not defined.

* I found it quite frustrating that $\Lambda_{-}$,  $\lambda_{-}$,  $\Lambda_{+}$ and  $\lambda_{+}$ were not defined. The definitions/models for these should be in the main text.

* line 200: "We employ the standard U-statistic estimator (Gneiting & Raftery, 2007; De Bortoli et al., 2025)." This is not informative: do you mean that you use the weighting scheme as in De Bortoli et al. (2025)? They do not refer to their scheme as standard or a U-statistic **estimator**, nor do they reference Gneiting and Raftery so I could not figure out what the above sentence implied.

*  "Let $P^{\kappa}_\text{ref}$ be the joint induced by the birth-death kernel." What is this kernel? It has not been defined at this point?

* The distributions and their parameters pop up in Proposition 3.1 without explanation of where and how they were derived. I found this information in the appendix, but the reader needs to be guided there.

### Related work
This is a minor, but I think DestVI might be worth mentioning in the context of ST deconvolution methods. Otherwise I believe the literature review is very comprehensive. Please double check how citations are delivered here, for instance the references between lines 309-310 after (CTMC) should be in parentheses.

### Experiments
Are the results averaged over multiple runs? Or are these single runs? For a stochastic algorithm like Count Bridges I would expect to see error bars and reported averages.

### Typos
* line 45: adresss
* line 170: "CUDA kernel implementing (Devroye, 2002)". Either it should be **implementation** or described what in Devroye 2002 has been implemented.
* line 173: Léonard (2013) should be in parentheses.
* JDS and RMSE are not defined properly.
* line 360: The sentence "STDeconvolve Miller et al. (2022)" is on the loose.
* In the Applications section there is inconsistencies in how CBs are referenced: count bridges (line 366), CB, Count bridges and Count Bridges.
* The introduced abbreviation CB is only used in Sec. 6.1?

**Questions:**

* How do you choose $\Lambda_{-}$,  $\lambda_{-}$,  $\Lambda_{+}$ and  $\lambda_{+}$?
* Could CBs be used to find a bridge between pairs of bulk data distributions?
* Could CBs be used to find a bridge between pairs of unit-level data distributions? Where the unit-level counts are inferred using your EM algorithm?

---

> ### Author Response · Authors · 2025-11-21
> **Response (I)**
>
> We thank the reviewer for their encouraging feedback and for describing our work as "novel, non-trivial and elegant." We are thrilled that you found the paper to be one of the best you have read this year. Below we have endeavored to address your questions and concerns.
>
> ## Questions
>
> > How do you choose the $\lambda$ schedule?
>
> In practice we find that a linear $w(t) = t$ schedule works well, with $\Lambda_+ = \Lambda_-$. We choose a moderate noise level, usually $~32$. We have added an analysis across noise levels in the appendix.
>
> **Table: Count Bridge (Energy Score) Results Across Different λ₊ = λ₋ Values**
>
> | λ₊ = λ₋ | MMD                    | W₂                     | Energy                 |
> |--------|-------------------------|-------------------------|-------------------------|
> | 0      | **0.0038 ± 0.0012**     | 0.0046 ± 0.0002         | **0.0075 ± 0.0011**     |
> | 8      | 0.0039 ± 0.0015         | **0.0045 ± 0.0006**     | 0.0080 ± 0.0022         |
> | 16     | 0.0049 ± 0.0003         | 0.0049 ± 0.0003         | 0.0095 ± 0.0004         |
> | 32     | 0.0052 ± 0.0024         | 0.0055 ± 0.0015         | 0.011 ± 0.006           |
> | 256    | 0.0064 ± 0.0020         | 0.0063 ± 0.0009         | 0.015 ± 0.004           |
>
> Our analysis actually demonstrates that our noise level was not fully optimized, and a lower noise level actually leads to better performance on the 8-Gaussians to 2-Moons task, possibly because the target distribution in this task is "low noise". If it would be of interest to the reviewer, we can rerun our experiments using this lower noise level.
>
> > Could CBs be used to find a bridge between pairs of bulk data distributions?
>
> If we have many "source" bulk distributions and many "target" bulk distributions, for example, bulk sequencing from unpaired patients in healthy and disease states, then yes, we could just model the counts at the bulk level using CBs. If instead we want to model the unit level at both source and target, we cannot do this directly with count bridges. The source must be observed at the unit level, and is assumed to be the "true" source. We could train a count bridge model on observable single-cell data and fine-tune this to deconvolve bulk. We did not focus on this "fine-tuning" or "mixing unit level" data into deconvolution in the manuscript for lack of space and time, but it is likely a highly effective strategy.
>
> > Could CBs be used to find a bridge between pairs of unit-level data distributions?
>
> Yes, we could deconvolve a pair of distributions and then train a separate unit-level bridge between these distributions.

---

> ### Author Response · Authors · 2025-11-21
> **Response (II)**
>
> ## Concerns
>
> > I found it quite frustrating that $\lambda$ were not defined...
> > "Let $P_{ref}^K$ be the joint induced by the birth-death kernel."
>
> We apologize for the lack of clarity in Section 3. We have rewritten the section to ensure these terms and the birth-death kernel are defined prominently in the main text before they are used.
>
> > Why should Count Bridges not be considered as an instance of Schrödinger Bridges?...
>
> We agree with the reviewer. Count Bridges indeed solve a Schrödinger Bridge problem on the integers using a specific reference process. Our previous phrasing ("connection between") was imprecise and accidentally obfuscatory. We have rewritten this section to explicitly frame CBs as an instance of SBs to avoid distancing the methods.
>
> > line 200: "We employ the standard"...
>
> We have clarified this section in two ways: first the "weighting" scheme here can really be suppresed as in our work we use a uniform weighting scheme. Second we have written out the U-statistic estimator so it is clearer how we estimate the loss.
>
> > The distributions and their parameters pop up in Proposition 3.1 without explanation...
>
> We have added a note here pointing to the appendix and have tried to give some idea of where these come from in the main text as well.
>
> > Are the results averaged over multiple runs?
>
> All synthetic experiments are averaged over multiple runs already (see the appendix for the error bars). We have now additionally run inference over the larger scale applications the variance is quite small since we have large evaluations sets and average over them for our distributional metrics, for example the three distributional metrics over the three:
>
> | Comparison            | Energy Distance | Wasserstein Distance | MMD (RBF) |
> |-----------------------|------------------|-----------------------|-----------|
> | Pred 1 vs True        | 8.891            | 0.017                 | 0.203     |
> | Pred 1 vs Spot Mean   | 42.903           | 0.034                 | 0.419     |
> | Pred 2 vs True        | 8.919            | 0.017                 | 0.203     |
> | Pred 2 vs Spot Mean   | 42.767           | 0.034                 | 0.418     |
> | Pred 3 vs True        | 8.898            | 0.017                 | 0.204     |
> | Pred 3 vs Spot Mean   | 42.856           | 0.034                 | 0.419     |
> | True vs Spot Mean     | 41.717           | 0.030                 | 0.409     |
>
> We will update the manuscript to present means and standard errors.
>
> ## Minor Corrections
>
> > Related...
>
> We thank the reviewer for the DestVI reference, which we have added to the context of ST deconvolution. We have also corrected the citation formatting issues (parentheses) in the Related Work section.
>
> > Typos...
>
> We have fixed all listed typos (e.g., "adresss", "STDeconvolve"). We have also standardized the usage of "Count Bridges" vs "CBs" throughout the manuscript and added formal definitions for JDS and RMSE in the main text.

---

> ### Author Response · Authors · 2025-11-25
> **A Brief Update**
>
> We would like to give a short update:
> 1. We have now added standard errors over inference seeds to all tables in the main applications section. We comment on this at the beginning of the applications section.
> 2. In response to reviewer TMxD we have substantially clarified the sequence-level deconvolution application and unified our presentation of the applications substantially.

---

### Author Response · Authors · 2025-11-21
**Summary of Response to Reviewers**

We thank all reviewers for their thoughtful, constructive, and often extremely positive assessments. We are grateful that multiple reviewers highlighted the novelty, mathematical rigor, elegance, and biological significance of *Count Bridges*, with comments such as:

* *"One of the best papers I have read this year."*
* *"Novel, non-trivial and elegant."*
* *"Mathematically sound and potentially impactful."*
* *"A principled integer-native alternative that fills a real gap in generative modeling."*

There are two major dimensions to our updates.

# **1. Writing Clarifications**

We have used the additional space to significantly rewrite and improve the clarity of the manuscript. We have rewritten Section 3 to be much clearer about our introduction to diffusion models and the core methodological contributions. We give a deeper exposition of the birth–death process, rate parameters $\lambda$, and the kernel ($K_{t|0}$). We have expanded the OT/SB discussion so that it is easier to follow and clarified the relationship to the Gaussian case. We have also significantly revised the proof in the appendix to make it smoother to read. Throughout we have cleaned up the notation.

We have additionally expanded our discussion and explanation of the main biological applications so our approach to both nucleotide level sequence modeling for bulk deconvolution and direct spatial transcriptomic deconvolution are as clear as possible.

# **2. Expanded Experiments, Ablations, and Analysis**

- We added a new experiment on a pure generative task to generate two moons, allowing us to compare Count Bridges to Gaussian Diffusion (Song et al., DDIM) and Discrete Diffusion (Shi et al., 2024). We found Count Bridges were the most performant across all three metrics.

- We added a full appendix documenting projection operators and compared three methods for rounding our projection approach. We found our preferred method is the best but a simpler alternative is a close second.

- We added a table comparing the performance of Count Bridges across many noise levels to the appendix.

- We have significantly deepened the biological analyses in the appendices, providing pathway and cell-type analysis alongside UMAPs to qualitatively validate the deconvolved transcriptomic profiles. These analyses show that CBs recover realistic single-cell manifolds and cell-type structure.

We thank all reviewers again for their time and thoughtful feedback, and we hope these additions strengthen the clarity, rigor, and impact of the paper.

**References**

Shi, J., Han, K., Wang, Z., Doucet, A. and Titsias, M., 2024. Simplified and generalized masked diffusion for discrete data. Advances in neural information processing systems, 37, pp.103131-103167.

Song, J., Meng, C. and Ermon, S., 2020. Denoising diffusion implicit models. arXiv preprint arXiv:2010.02502.

---

### Author Response · Authors · 2025-12-03
**A Final Summary for the AC**

We thank the reviewers for their helpful commentary and kind words about our manuscript:

- _"One of the best papers I have read this year."_
- _"Novel, non-trivial and elegant."_
- _"Mathematically sound and potentially impactful."_
- _"A principled integer-native alternative that fills a real gap in generative modeling."_

In our rebuttal, we have addressed every comment from every reviewer point by point. Reviewers had several major categories of requests:
1. Clarity: We have extensively rewritten large sections of the theory and applications text for improved clarity. **This back and forth led Reviewer TMxD to raise their score from 4 -> 8.**
2. Baselines: We added two additional diffusion baselines and an additional bulk deconvolution baseline. These experiments required dozens of new model runs.
3. Ablations: We ran ablations across added noise in the count process and ablations for our projection operation.
4. Biological relevance: We added pathway and cell-type analyses alongside UMAPs, which all validate that deconvolved transcriptomic profiles are realistic. We additionally extended our results to real Visium data in the appendix, which we validate similarly.
5. Standard errors: We have now included standard errors for all the tables in the main text.
6. Algorithmic appendix: We have added a full appendix describing all core algorithms in detail, along with a computational and memory complexity analysis.

We would also like to call attention to the fact **Reviewer S8Cu made the false claim that we do not analyze deconvolution**: the reviewer appears to have missed Figure 4 in the main text, the consistent discussion of the limits of deconvolution throughout the paper, and Appendix B, where we give a theoretical treatment. Other elements of this review also make it seem LLM-generated. Nonetheless, we responded to all the reviewer's requests and had hoped to resolve this during the discussion, but the reviewer did not respond before the discussion was frozen. **Since the concerns of Reviewer TMxD have been resolved (see discussion), Reviewer S8Cu is the only reviewer to not recommend acceptance.**

---

### Meta-Review · Area_Chair_5TQh · 2026-01-08

**Summary:**

The authors propose Count Bridges, a novel generative framework specifically designed for ordinal/count data. Traditional diffusion models rely on Gaussian noise (continuous). While discrete diffusion exists, it often struggles with the specific structure of biological counts. Count Bridges use a birth–death process (based on Poisson dynamics) to create a "bridge" between probability distributions.

The discussion was thorough, I agree with reviewers.

**Reviewer Concerns:**

Several reviewers (K7ei, xPQ5) found Section 3 difficult to follow, noting that key variables ($\lambda$, $\beta$, etc.) and the "birth-death kernel" were not properly defined in the main text.

Reviewer S8Cu (the most critical reviewer) requested sensitivity analyses on group size. They argued that deconvolving aggregates becomes "ill-posed" as the number of cells in a group increases.

Reviewers wanted to see the model tested on "genuine" spatial data (like 10x Visium) rather than just synthetic simulations.

**Reviewer Scores:**

TMxD 4 -> 8

---

### Decision · Program_Chairs · 2026-01-26

Accept (Poster)